# YTHDC1 m⁶A-dependent and m⁶A-independent functions converge to preserve the DNA damage response

Daniel Elvira-Blázquez [1,2], José Miguel Fernández-Justel [1,2], Aida Arcas [1,2,3], Luisa Statello [1,2], Enrique Goñi[1,2], Jovanna González[1,2], Benedetta Ricci [4], Sara Zaccara [4], Ivan Raimondi [5,6✉] & Maite Huarte [1,2✉]

## Abstract

**Cells have evolved a robust and highly regulated DNA damage response to preserve their genomic integrity. Although increasing evidence highlights the relevance of RNA regulation, our understanding of its impact on a fully efficient DNA damage response remains limited. Here, through a targeted CRISPR-knockout screen, we identify RNA-binding proteins and modifiers that participate in the p53 response. Among the top hits, we find the m⁶A reader YTHDC1 as a master regulator of p53 expression. YTHDC1 binds to the transcription start sites of *TP53* and other genes involved in the DNA damage response, promoting their transcriptional elongation. YTHDC1 deficiency also causes the retention of introns and therefore aberrant protein production of key DNA damage factors. While YTHDC1-mediated intron retention requires m⁶A, *TP53* transcriptional pause-release is promoted by YTHDC1 independently of m⁶A. Depletion of YTHDC1 causes genomic instability and aberrant cancer cell proliferation mediated by genes regulated by YTHDC1. Our results uncover YTHDC1 as an orchestrator of the DNA damage response through distinct mechanisms of co-transcriptional mRNA regulation.**

**Keywords** YTHDC1; p53; DDR; RNAPII Pausing; Splicing
**Subject Categories** Chromatin, Transcription & Genomics; RNA Biology

## Introduction

Cells are continuously exposed to extrinsic and intrinsic DNA-damaging agents, including ionising radiation, oxidative free radicals, and replication stress, among many others (de Almeida et al, 2021; Lieber, 2016). These agents can cause persistent and cumulative damage, necessitating the evolution of pathways that are capable of detecting and repairing DNA damage sites to maintain genome integrity (Arcas et al, 2014). Since the diverse sources of damage cause different types of lesions, cells have developed a comprehensive set of mechanisms to sense and repair the genome, which collectively constitutes the DNA damage response (DDR) (Ceccaldi et al, 2016; Ciccia and Elledge, 2010; Huang and Zhou, 2021; Groelly et al, 2022). The proper integration and regulation of this response is crucial for the maintenance of cellular homoeostasis and, ultimately, the life of complex organisms. The DDR strongly relies on the pivotal role of p53, a key transcription factor, and tumour suppressor. In response to stress, p53 orchestrates a gene expression programme that enforces cell cycle arrest to allow DNA repair or triggers apoptosis when the damage is too extensive (Kastenhuber and Lowe, 2017; Riley et al, 2008). When DNA damage occurs, primary sensors activate ATR and ATM kinases, initiating a phosphorylation cascade involving CHK1, CHK2, and ATM, which phosphorylate p53 (Banin et al, 1998; Cimprich and Cortez, 2008; Derheimer et al, 2007; Maréchal and Zou, 2013; Tibbetts et al, 1999). Phosphorylation of p53 hinders its interaction with the E3 ubiquitin ligase Mdm2, leading to its stabilisation and nuclear translocation (Chène, 2001; Gaglia et al, 2013; Haupt et al, 1997; Kubbutat et al, 1997), followed by binding and activation of its direct transcriptional targets, encompassing protein-coding genes and noncoding RNAs (El-Deiry et al, 1993; Huarte et al, 2010; Riley et al, 2008). This concerted action plays a crucial role in the DNA damage protective response of p53, safeguarding cell homoeostasis.

Although posttranslational control of DNA damage factors such as p53 is widely recognised as critical for articulating the DNA damage response (Bode and Dong, 2004; Huen and Chen, 2007; Liu et al, 2019), the direct involvement of mRNA regulation has been more challenging to establish. The introduction of RNA post-transcriptional modification as an additional layer of gene expression control has heightened the need for a comprehensive understanding of how molecular and biochemical processes affecting RNA can influence genomic integrity. In particular, N(6)-methyladenosine (m⁶A) deposition on RNA has emerged as a

[1]Center for Applied Medical Research (CIMA), University of Navarra, Pamplona, Spain. [2]Institute of Health Research of Navarra (IdiSNA), Pamplona, Spain. [3]Clarivate, Barcelona, Spain. [4]Department of Systems Biology, Columbia University Irving Medical Center, New York, NY, USA. [5]New York Genome Center, New York, NY, USA. [6]Weill Cornell Medicine, New York, NY, USA. ✉E-mail: iraimondi@nygenome.org; maitehuarte@unav.es

major epigenetic mark impacting RNA fate in the context of stress (Qi et al, 2022; Zhou et al, 2015). The regulatory potential of m⁶A is largely mediated through the crosstalk between m⁶A modification and RNA-binding proteins of the YTH family, which act as specific readers of this biochemical mark (Hsu et al, 2017; Xiao et al, 2016; Xu et al, 2021; Zaccara and Jaffrey, 2020). YTHDC1 is the only member of the family with a localisation exclusive to the nucleus, and has been implicated in the co-transcriptional regulation of coding and noncoding RNAs (Dattilo et al, 2023; Widagdo et al, 2022; Xiao et al, 2016). Significantly, YTHDC1 has been linked to various facets of RNA expression, encompassing the regulation of splicing, mRNA stability, mRNA nuclear export and transcriptional control (Akhtar et al, 2021; Roundtree et al, 2017; Su et al, 2023; Timcheva et al, 2022; Widagdo et al, 2022; Xiao et al, 2016; Zhang et al, 2020). Nevertheless, the interdependence of diverse regulatory processes and the selective recognition of m⁶A-modified RNAs by YTHDC1 warrant further elucidation. Furthermore, while YTHDC1 has been associated with the initiation and progression of diverse cancer types (Su et al, 2023; Yan et al, 2023; Yan et al, 2022), a direct mechanistic correlation between its gene regulatory activities and the consequential biological impact on crucial cancer pathways has yet to be firmly established. Here we investigated the role of post-transcriptional RNA regulation in the response to DNA damage and genomic stability. By conducting a targeted genetic screen centred around proteins involved in RNA modification pathways, we successfully identified key factors that activate the p53 response. Notably, our top hit was YTHDC1, revealing a positive regulatory circuit between *TP53* and *YTHDC1* expression, and indicating that YTHDC1 functions as a general regulator of the p53 response. Additionally, we discovered that YTHDC1 controls p53 by directly regulating the transcriptional pausing-release of RNA polymerase II (RNAPII). Intriguingly, this mechanism extends to other DNA damage factors, leading to intron retention and abnormal protein expression of genes central to the DNA damage response, such as *ATR*. We found that intron retention is m⁶A-dependent, while regulation of pausing-release is not specific to genes with m⁶A-modified mRNAs. Importantly, our study demonstrates that YTHDC1-mediated mRNA regulation significantly impacts the genomic stability of cancer cells. Collectively, our findings unveil distinct aspects of YTHDC1's role in gene regulation and highlight its critical influence on genomic stability.

## Results

### Identification of RNA modifiers that affect p53 response through reporter-based CRISPR screening

We aimed to investigate how RNA metabolism might influence the DNA damage response and play a role in preserving genomic stability. In particular, we focused our efforts on understanding how RNA modifiers could impact the activation of the p53 pathway, which is central to this response. To unbiasedly determine the role of known and putative RNA modifiers in modulating p53 response dynamics, we designed a CRISPR–Cas9 knockout screen using a reporter system that is able to detect the activation of the p53 transcriptional programme (Hsu et al, 2019). Briefly, the lung cancer cell line A549 was engineered to express the fusion product of the endogenous *CDKN1A* (p21) gene with mVenus (Fig. 1A),

enabling the measurement of levels of p53 activation using fluorescence as readout (Fig. EV1A). We further modified p21-reporter cells to stably express SpCas9 endonuclease (Fig. EV1B), and transduced these cells with human lentiviral-based CRISPR pooled library in CRISPseq-BFP-backbone vector (Addgene). The library, containing 1628 sgRNAs (Dataset EV1), was designed to target 407 genes encoding proteins known or predicted to be involved in the catalysis or recognition of RNA modifications, based on the presence of RNA-binding domains and/or structural homology with previously known RNA modifiers (Barbieri et al, 2017). In addition, the library included 200 non-target sgRNAs as negative controls.

To isolate the effects directly related to p53 signalling, we performed our CRISPR screening in cells treated with Nutlin-3a, a potent inhibitor of Mdm2-mediated p53 degradation (Fig. 1A). This treatment results in a strong accumulation of functional p53 protein (Fig. EV1C). In order to identify positive and negative regulators of the p53 pathway, cells were treated with 10 μM Nutlin-3a for 16 h, which is the minimal dose necessary to sufficiently stabilise p53 to activate its cognate targets and induce the reporter gene expression in 95% of the cells, reflected by an increase of p21-mVenus signal (Fig. EV1A). Then, from the cell population that incorporated a lentiviral-sgRNA (BFP positive), we isolated those cells showing either higher or lower p53 activation compared to the average activation of the total population, followed by sequencing to identify enriched sgRNAs (Figs. 1B and EV1D). To exclude unspecific effects, we performed in parallel the screening on untreated cells (MOCK), representing the entire number of guide RNAs in the screen in basal conditions. As we expected, p21-mVenus signal was close to background levels in MOCK condition. We used the MOCK population as reference to determine enrichment scores of p53-enhanced-response and p53-attenuated-response populations, defined as the group of genes whose disruption promotes increased or decreased activation of the p53 pathway, without affecting cellular homoeostasis in untreated condition. We defined Log₂FC values for each candidate gene in both populations as the average log₂ fold change in the abundance of all sgRNAs targeting the same gene normalised by the 200 non-target-sgRNAs and compared with the total fraction (Figs. 1C and EV1E; Dataset EV1).

To gain understanding of the type of RNA regulations that are enhancing and attenuating the p53 response, we globally examined the top 50 candidates from each population to identify which molecular substrates are implicated (Figs. 1D and EV1F; Dataset EV1). This analysis indicated that most of the hits in the p53-enhanced-response population were RNA modifiers, or protein modifiers containing domains potentially related to RNA modifications. However, in the p53-attenuated-response population, there was a significantly higher proportion of mRNA-targeting RNA modifiers, with a much lower proportion of protein-modifying enzymes. Notably, within N-methylation modifications, m⁶A, the most widely studied modification in RNA (Pan, 2013), was the most predominant. The individual analysis of the top gene hits showed that our reporter-based CRISPR screen was able to effectively detect previously described p53 positive regulators, such as SETD2 and PRDM1 (Xie et al, 2008; Yan et al, 2007), which had enriched depletion in the p53-attenuated-response population. In addition, established p53 negative regulators, including methyltransferases PRMT5 and PRMT1 (Jackson-Weaver et al, 2020; Scoumanne et al, 2009), were detected as being more enriched in the p53-enhanced-response population (Fig. EV1G). Furthermore,

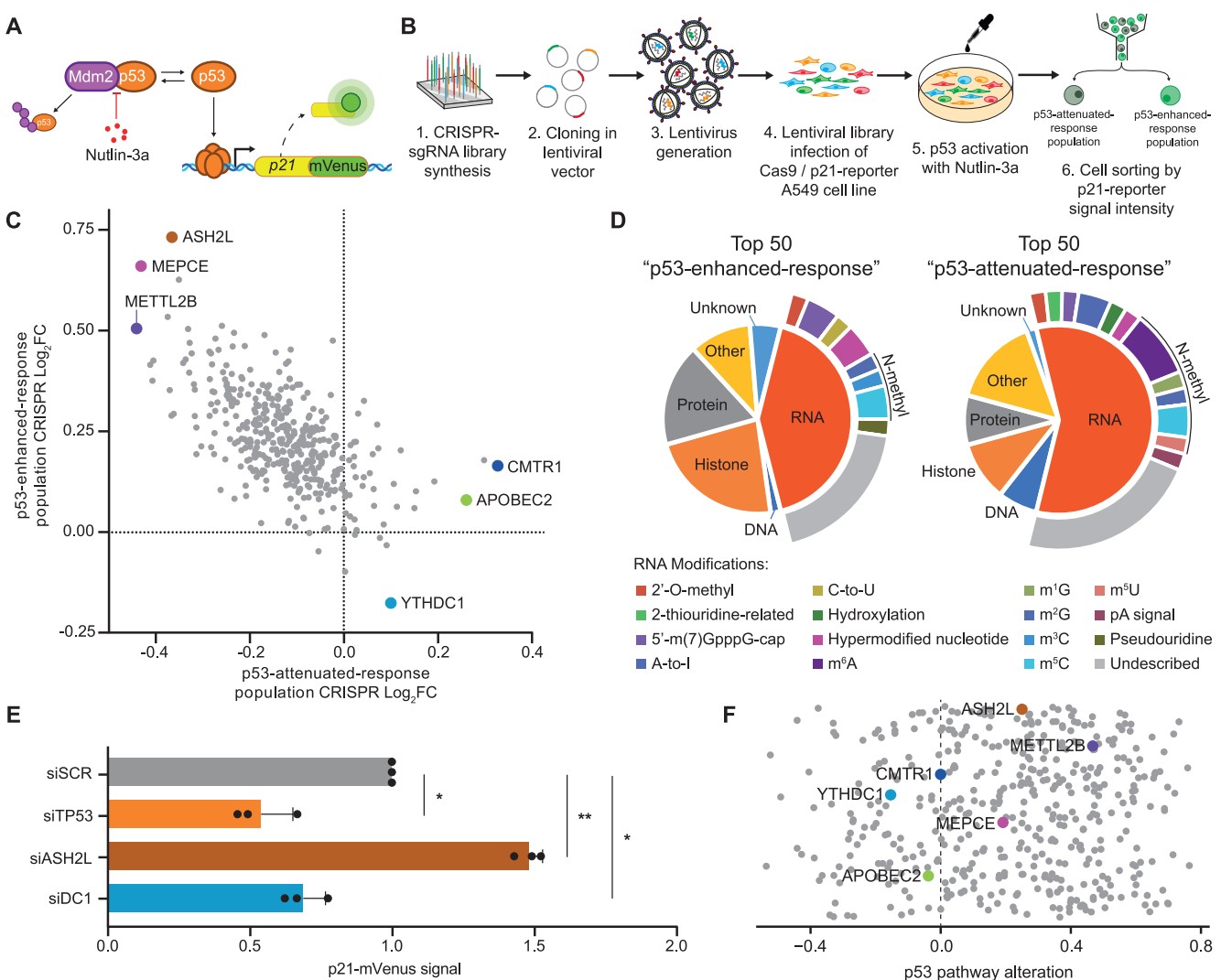

**Figure 1. CRISPR screening enables unbiased identification of RNA modifiers involved in p53 response.**

(A) Schematic representation of p21 endogenously tagged reporter system in A549 cell line. (B) Overview of the CRISPR-screening design. A library containing 1680 guide RNAs was cloned into a lentiviral system and used to infect A549 reporter cells. After 4 days of infection, cells were exposed to Nutlin-3a to allow for p53 protein stabilisation and subsequent reporter gene activation. Cells were sorted by FACS based on the level of reporter gene activation. (C) Scatter plot of $Log_2$ Fold Change for sgRNA enrichment in p53-enhanced-response ($Y$ axis) and p53-attenuated-response ($X$ axis) populations. The three top candidates for each population are labelled and highlighted in the plot. (D) Functional annotation of the top 50 candidates for p53-enhanced and p53-attenuated populations based on their molecular activity. (E) In vitro validation of the top candidate for p53-enhanced and p53-attenuated populations. Each gene was independently silenced by siRNA. After silencing, mVenus signal was taken as readout of p21-reporter gene activation. Statistical analysis was performed by paired two-tailed Student's $t$ test KD versus scramble (SCR) control. (F) Pearson correlation score between p53-downstream-effector misregulation (p53 pathway) compared to the expression of the 407 CRISPR-Screening candidates in Lung carcinoma patients from TCGA (LUAD + LUSC patients) with p53 *wild-type* genotype versus healthy patients. Data information: All data are shown are representative of at least three independent experiments. Data are presented as mean ± s.d. *$P \leq 0.05$, **$P \leq 0.01$, paired two-tailed Student's $t$ test was performed in (E). Source data are available online for this figure.

it identified ASH2L, MEPCE and METTL2B as top hits from the p53-enhanced-response population, where ASH2L is a histone methyltransferase with putative activity on RNA (Barsoum et al, 2022), MEPCE is both an RNA binding and RNA methyltransferase of 5' capping 7SK snRNA (Shelton et al, 2018) and METTL2B is involved in N3-methylcytidine ($m^3C$) on tRNAs (Xu et al, 2017). In the opposite direction, it identified CMTR1, APOBEC2 and YTHDC1 as top hits from the p53-attenuated-response population, with CMTR1 mediating the O-2'-methylation that forms the predominant cap structure 5'-m(7)GpppG (Liang et al, 2022),

APOBEC2 performing cytidine deaminase activity on mRNA (Okuyama et al, 2012) and YTHDC1 acting as a canonical nuclear $m^6A$-reader protein (Xu et al, 2014) (Figs. 1C and EV1E).

To orthogonally validate the screening results, we identified the top candidates where gene knockout resulted in the largest p21-reporter fold change signal for each direction, selecting the negative regulator ASH2L and the positive regulator YTHDC1. We depleted the mRNA coding for these proteins through small-interfering RNAs (siRNA) transfection in p21-reporter A549 cell line. Each target was silenced with two independent siRNAs, resulting in a

decreased expression (69.1% and 83.7% for *YTHDC1*; 81.3% and 80.4% for *ASH2L*) (Fig. EV1H) compared to control scrambled siRNA (siSCR). As expected, the depletion of ASH2L resulted in an increase of p21-reporter signal, while the depletion of YTHDC1 resulted in a decreased level of response, recapitulating what we previously observed (Fig. 1E). Together, these data corroborate our screen results, confirming that we were able to identify bona-fide p53 response regulators.

For further functional validation, we sought to investigate the relation between our CRISPR screen hits and p53 pathway activation in a physio-pathological context. Since the screen was performed in a lung cancer cell line, we analysed RNA-seq from 326 primary lung carcinoma (LUAD and LUSC) *TP53* wild-type tumour samples and 108 healthy *TP53* wild-type lung samples from TCGA database, in order to correlate the expression of candidate genes with alterations in the p53-downstream effectors (p53 pathway). For each individual tumour sample, we calculated a p53 pathway alteration score based on the projection of the gene expression level onto a low-dimensional manifold in the subspace spanned by genes that canonically define this pathway ("Methods"). Then, we correlated the alteration score with the expression of the genes used in the screen within the same tumour sample. Interestingly, even considering the inherent heterogenicity between patient samples, the top three candidates from the p53-enhanced-response population (*ASH2L*, *MEPCE* and *METTL2B*), showed a clustered positive Pearson correlation with p53 alteration score, indicating that the higher the expression of these genes in the tumour, the more altered the p53 response. The same pattern but in the opposite direction was observed for the top three hits from the p53-attenuated-response cluster (*APOBEC2*, *CMTR1* and *YTHDC1*), suggesting that p53 pathway alteration anti-correlates with the expression of these genes. Therefore, the primary tumour analysis indicates that the in vitro screening recapitulates the cancer cell behaviour observed in vivo (Fig. 1F).

In summary, this high-throughput genetic screening has uncovered several RNA regulators that exert control over the gene expression outcomes mediated by the tumour suppressor p53.

## YTHDC1 regulates *TP53* transcription

The CRISPR screen revealed a significant enrichment of RNA metabolism proteins among the top candidates whose gene knockout resulted in reduced p53-reporter activation (Fig. 1D). Notably, proteins involved in m6A-mRNA metabolism emerged as key targets, such as METTL14, ALKBH5 and METTL5, with YTHDC1 as the top candidate. Consequently, we decided to investigate the relationship between YTHDC1 and p53 in greater depth. YTHDC1 is a nuclear protein, whose depletion resulted in the strongest reduction in p21 expression. Our screen was designed to assess p21-reporter induction through p53 stabilisation by Nutlin-3a treatment. Therefore, to determine whether the downregulation of p21 detected upon YTHDC1 depletion is a direct consequence of p53 protein dysregulation, or if it is the downstream components of the p53 cascade that are affected, we performed siRNA-mediated YTHDC1 knockdown in A549 cells followed by treatment with Nutlin-3a, and quantified p53 protein levels. YTHDC1 depletion (Figs. EV1H and EV2A) led to a significant reduction of p53 protein levels (Fig. 2A), suggesting that the observed reduction of p21 expression following YTHDC1 inactivation is the consequence of a decrease in p53 protein. Given that YTHDC1 is

a nuclear protein, we excluded the possibility that the lack of p53 protein was due to a decrease in *TP53* messenger RNA translation. However, several YTH family proteins have been reported to control m6A-marked RNA, modifying their stability, resulting in dramatic changes to RNA half-life (Wang et al, 2014; Zaccara and Jaffrey, 2020; Zhang et al, 2020). To investigate whether the control of mRNA stability through m6A might be a possible mechanism by which YTHDC1 regulates p53 levels, we analysed *TP53* mRNA stability, and the short half-life *c-Myc* mRNA as control, upon siRNA-mediated depletion of YTHDC1. This analysis showed no changes in *TP53* mRNA half-life after 8 h of treatment with the transcription inhibitor actinomycin D (ActD), suggesting that a different mechanism is controlling p53 levels (Figs. 2B and EV2B). We hypothesised that YTHDC1 could instead be regulating the *TP53* gene directly at the transcription level. To explore this possibility, we performed a series of RT-qPCR analyses to identify potential differences in nascent *TP53* RNA levels upon depletion of YTHDC1. We performed siRNA-mediated knockdown of YTHDC1 in A549 cells, and used siRNA knockdown of p53 as control. By using a primer pair designed to amplify mature RNA, we observed a significant reduction of *TP53* mRNA level in YTHDC1-knockdown cells, suggesting that the observed lack of p53 protein product is a direct consequence of the downregulation of *TP53* mRNA (Fig. 2C). We additionally performed RT-qPCR to target intron 10 of the p53 gene as a proxy for nascent RNA production, finding a 60% reduction in newly synthesised RNA compared to cells transfected with control scramble siRNA, which strongly suggests that YTHDC1 is directly involved in regulating transcription of the p53 gene (Fig. 2C). In contrast, as expected, p53-knockdown cells had a significant reduction of the mature *TP53* RNA, but no significant change in its pre-mRNA levels. Importantly, similar results were observed when YTHDC1 was abolished by CRISPR–Cas9 in a stable knockout A549 cell line (Fig. EV2C–E).

To characterise the transcriptome-wide effect of YTHDC1 depletion, we performed total RNA-seq in siRNA-treated A549 cells. In agreement with our previous observations, the RNA-seq analysis revealed that *TP53* gene expression was downregulated in YTHDC1-knockdown cells (Fig. 2D,E; Dataset EV2). Pathway-level analysis of the differentially expressed genes indicated that p53 signalling was the most affected pathway, suggesting that YTHDC1 serves as a critical regulator controlling the proper execution of the p53 response (Fig. 2F). To further explore this relationship between *YTHDC1* and *TP53* that we observed in the A549 lung carcinoma cell line in other tumour types, we conducted a correlation analysis for differential gene expression in different tumours from TCGA database, including lung adenocarcinoma (LUAD), lung squamous cell carcinoma (LUSC) and colorectal adenocarcinoma (COAD) patients. Notably, we discovered a positive correlation between *TP53* and *YTHDC1* gene expression in all of the tumour types, showing LUAD the highest and most significant correlation (Fig. EV2F–H). We further confirmed this relationship by examining public Dependency Map data from the Broad Institute (Tsherniak et al, 2017), analysing *YTHDC1* and *TP53* expression levels in over 1000 different cell lines grouped by lineage (Fig. EV2I). Again, we found a positive correlation across multiple cancer lineages. Finally, we performed YTHDC1 depletion in three different cell lines from different tissues: MCF7 (breast), HeLa (uterus/cervix) and HCT116 (colon). As observed in the A549 cell line, the knockdown of YTHDC1 promoted a significant decrease of mature and pre-mRNA levels of *TP53*, confirming that

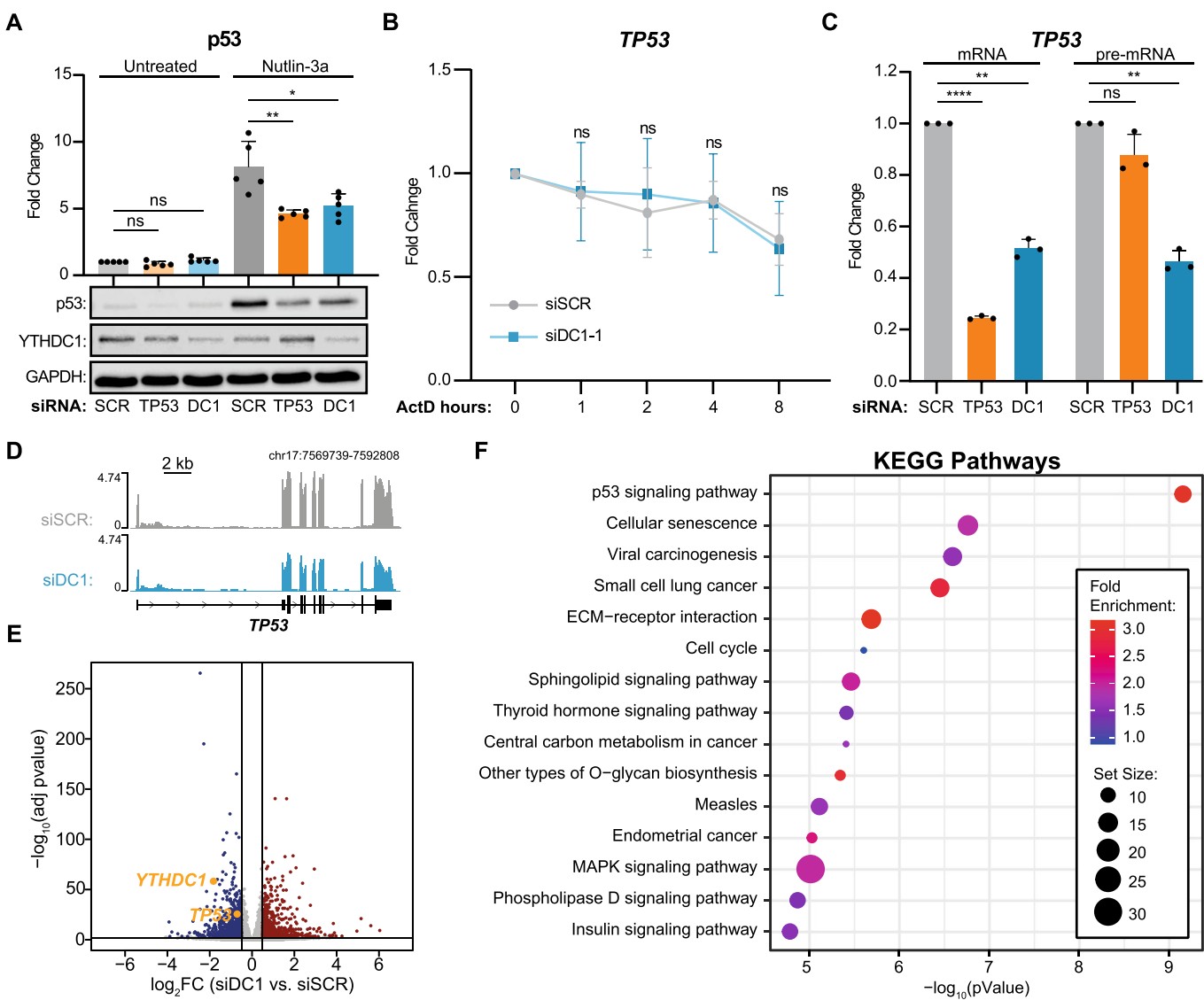

**Figure 2. YTHDC1 Directly regulates *TP53* transcription.**

(A) Representative immunoblot of YTHDC1 and p53, with GAPDH as loading control. Cells were transfected with siRNA against *TP53*, *YTHDC1* or Scramble (SCR) as negative control. After silencing cells were treated to Nutlin-3a, DMSO was used for the untreated condition as a negative control. The bar plot shows protein quantification relative to GAPDH level of $n = 5$ biologically independent experiments. (B) *TP53* RNA stability assay. Cells were transfected with siRNA, YTHDC1, or Scramble (SCR) as negative control. After silencing cells were treated with Actinomycin D to stop the transcription. Cells were collected at different time points indicated on the X axis to assess *TP53* RNA level by quantitative RT-qPCR. In vitro transcribed *Luciferase* RNA was used as spike-in to normalise the signal. (C) RNA-level quantification of mature mRNA and pre-mRNA by RT-qPCR. Cells were transfected with siRNA against *TP53* (orange), *YTHDC1* (cyan) or Scramble (light grey) as negative control. (D) Genome browser tracks for total RNA-seq showing reads coverage over *TP53* gene. Sequencing data were normalised as Fragments Per Kilobase of transcript per Million mapped reads (FPKM). (E) Volcano plot showing differentially expressed genes (DEGs) identified in YTHDC1-silenced cells versus scramble (SCR) cells. Genes with $\log_2(FC) > 0.5$ and Adjusted $P$ value (Benjamin–Hochberg correction) <0.001, resulting from the DESeq2 analysis, are considered DEGs. Significantly upregulated, downregulated, or not changed genes in the YTHDC1 knocked-down cells are labelled with blue, red, or grey colour, respectively. (F) The Kyoto Encyclopedia of Genes and Genomes (KEGG) enrichment analysis showing pathways affected by DEGs identified in (E). Data information: All data are shown are representative of at least three independent experiments. Data are presented as mean ± s.d. ns not significant $P > 0.05$, *$P \leq 0.05$, **$P \leq 0.01$, ****$P \leq 0.0001$, paired two-tailed Student's $t$ test was performed in (A–C). DESeq2 Wald test $P$ value was calculated in (E). Statistical test was intrinsically applied as explained in pathfinder pipeline (Ulgen et al, 2019) in (F). Source data are available online for this figure.

the positive correlation between these genes is not dependent on the origin of the cell line (Fig. EV2J,K).

Together, these findings strongly suggest that there is a positive regulatory circuit between *YTHDC1* and *TP53* transcription, implicating YTHDC1 as a general regulator of p53 response both in vitro and in vivo.

## YTHDC1 modulates transcription through promoter-proximal pausing-release of RNA polymerase II

Our results thus far demonstrated that YTHDC1 acts as a master regulator of *TP53* transcription, which is required by cells to overcome DNA damage. Transcription is a complex stepwise

process that involves the recruitment and assembly of the machinery responsible for initiation, pause-release, elongation, and termination. Notably, polymerase pausing represents a critical, tightly regulated juncture of the transcription process that upon disruption can lead to cellular dysfunction, cancer or aging (Gyenis et al, 2023; Shao and Zeitlinger, 2017; Wang et al, 2022; Zhang et al, 2017). As we showed that YTHDC1 regulates *TP53* at the transcriptional level, we hypothesised that YTHDC1 depletion may specifically alter transcription dynamics. To investigate this, we performed RNAPII ChIP-seq in both YTHDC1-knockdown and scramble siRNA control A549 cells (Fig. 3A). We additionally performed YTHDC1 ChIP-seq and chromatin CLIP-seq (enriched in nascent RNA), to further dissect the relationship between transcription and YTHDC1 DNA and RNA binding genome-wide, respectively.

We used the RNAPII ChIP-seq data to calculate RNAPII pausing index values based on $\log_2$ RNAPII promoter density/ RNAPII gene body density ("Methods"), and identified 6026 out of 21727 genes with an increased value of Pausing Index upon YTHDC1 depletion, indicating that YTHDC1 is widely needed to release paused RNAPII for continued transcript elongation. The extent of RNAPII pausing variation was comparable to that observed when comparing wild-type and knockout cells of the elongation factor NELF (Wu et al, 2022) (Fig. EV3A), highlighting the impact of YTHDC1 in transcriptional regulation. To evaluate the direct role of YTHDC1 in the regulation of promoter-proximal RNAPII pausing, we integrated the pausing index analysis data with YTHDC1 ChIP-seq and CLIP-seq binding around TSS. Our analysis confirmed a direct relationship between YTHDC1 binding and promoter-proximal RNAPII pausing, as it was primarily the genes bound by YTHDC1 ChIP that showed the most altered pausing after YTHDC1 silencing (Fig. 3B; Dataset EV3). However, this correlation was not observed when analysing the binding of YTHDC1 to the chromatin-associated RNA by CLIP (Fig. EV3B), suggesting that YTHDC1 transcriptional regulation is occurring at the chromatin/DNA level.

More importantly, we observed significant changes in RNAPII occupancy after YTHDC1 knockdown in *TP53* gene locus, being one of the most affected genes with altered RNAPII binding at promoter regions, further validated by ChIP-qPCR analysis (Fig. 3C,D). The quantification of RNAPII ChIP showed increased presence of RNAPII around the 5' and decrease towards the 3' of *TP53* gene in conditions of YTHDC1 depletion (Fig. 3D), indicating that RNAPII pausing is linked to the observed down-regulation of *TP53* expression. We then concluded that YTHDC1, by binding to their TSSs, regulates the transcriptional elongation of multiple genes, *TP53* included.

## YTHDC1 affects the correct splicing of factors involved in the DNA damage response

ChIP-seq data revealed a widespread YTHDC1 positive effect on the transcriptional machinery to promote the release of RNAPII from the paused state, which leads to the productive transcription of the pivotal tumour suppressor *TP53*. However, not all the genes with high transcriptional pausing index upon YTHDC1 depletion showed obvious changes in their mRNA expression levels. Most interestingly, these genes included several DNA damage factors, with the DNA repair as the most enriched functional pathway

(Fig. EV3C; Dataset EV3), suggesting that YTHDC1 has a broader role as an orchestrator of the cellular DDR. We hypothesised that the altered transcription process occurring upon YTHDC1 depletion could lead to other mechanisms of aberrant mRNA production of DNA damage factors, such as defects in splicing, as previously reported (Akhtar et al, 2019; Caizzi et al, 2021; Saldi et al, 2016). Indeed, we found several alternative splicing events, such as exon inclusion or intron retention upon silencing of YTHDC1 (Fig. 4A). We decided to apply more accurate pipelines to detect different splicing aberrations. In particular, the IRFinder pipeline is specifically designed to detect retained introns (Middleton et al, 2017), and identified a total of 52 genes containing differentially retained introns (74 differentially retained introns, adjusted *P* value < 0.1). Interestingly, we identified three key players in the DNA damage response, *ATR*, *BIRC6* and *SETX*, among the most affected genes (Fig. 4B, Dataset EV2). YTHDC1 knockdown promoted a clear retention of intron 3 in *ATR*, intron 10 in *BIRC6*, and intron 10 in *SETX* (Fig. 4C). ATR has a major role as one of the primary effectors of the DNA repair pathway in the presence of single-stranded breaks (SSBs) (Maréchal and Zou, 2013). BIRC6/ Bruce/Apollon functions together with ATM and ATR to potentiate DNA repair pathway signalling (Ge et al, 2015, 2019), and SETX is a RNA:DNA helicase involved in the resolution of R-loop structures formed while RNAPII is transcribing (Hasanova et al, 2023), which is necessary to avoid the formation of DNA breaks (Gan et al, 2011). While we did not observe a significant reduction in total mRNA level for *ATR*, *BIRC6* and *SETX* (Fig. EV4A; Dataset EV2), the aberrant splicing led to the predicted appearance of truncated open- reading frames (Fig. EV4B).

To further confirm YTHDC1 direct activity in regulating mRNA splicing, we analysed relative location of YTHDC1-CLIP peaks to the retained introns and identified the preferential binding of YTHDC1 to retained introns and their flanking exons globally (Fig. EV4D), which emphasizes the direct regulation of the splicing over the affected introns upon YTHDC1 depletion. Moreover, we further compared the relative distribution for YTHDC1 ChIP (genome) and CLIP (transcriptome) peak (Fig. EV4E). YTHDC1 localised more frequently around splicing sites through the transcriptome, while it was preferentially located at TSS when it binds to the genome. It highlights the differential function that YTHDC1 exerts over splicing regulation or transcriptional regulation.

To investigate the functional consequence of YTHDC1 depletion at the level of protein production of these genes, we measured their protein levels in A549 cells treated with two independent siRNAs to knockdown YTHDC1. Western blot analysis confirmed that YTHDC1 depletion led to a significant reduction of all three proteins analysed (Fig. 4D–F). To validate that the reduction in protein observed is a direct consequence of aberrant splicing described above, we performed splicing-aware RT-qPCR on A549 cells depleted for YTHDC1 with two independent siRNAs. The primers were specifically designed to selectively amplify mature mRNA or the relevant mis-spliced isoform for each gene (Fig. EV4C). YTHDC1 depletion led to a significant reduction of fully spliced mRNA and significantly increased levels of the intron-retained forms of all three genes (Fig. 4G–I). These results show that splicing regulation by YTHDC1 has a strong impact in the levels of these DNA damage proteins. YTHDC1-KO cells showed the same phenotype on intron retention over *ATR*, *BIRC6* and *SETX* transcripts, further confirming YTHDC1 activity (Fig. EV4F).

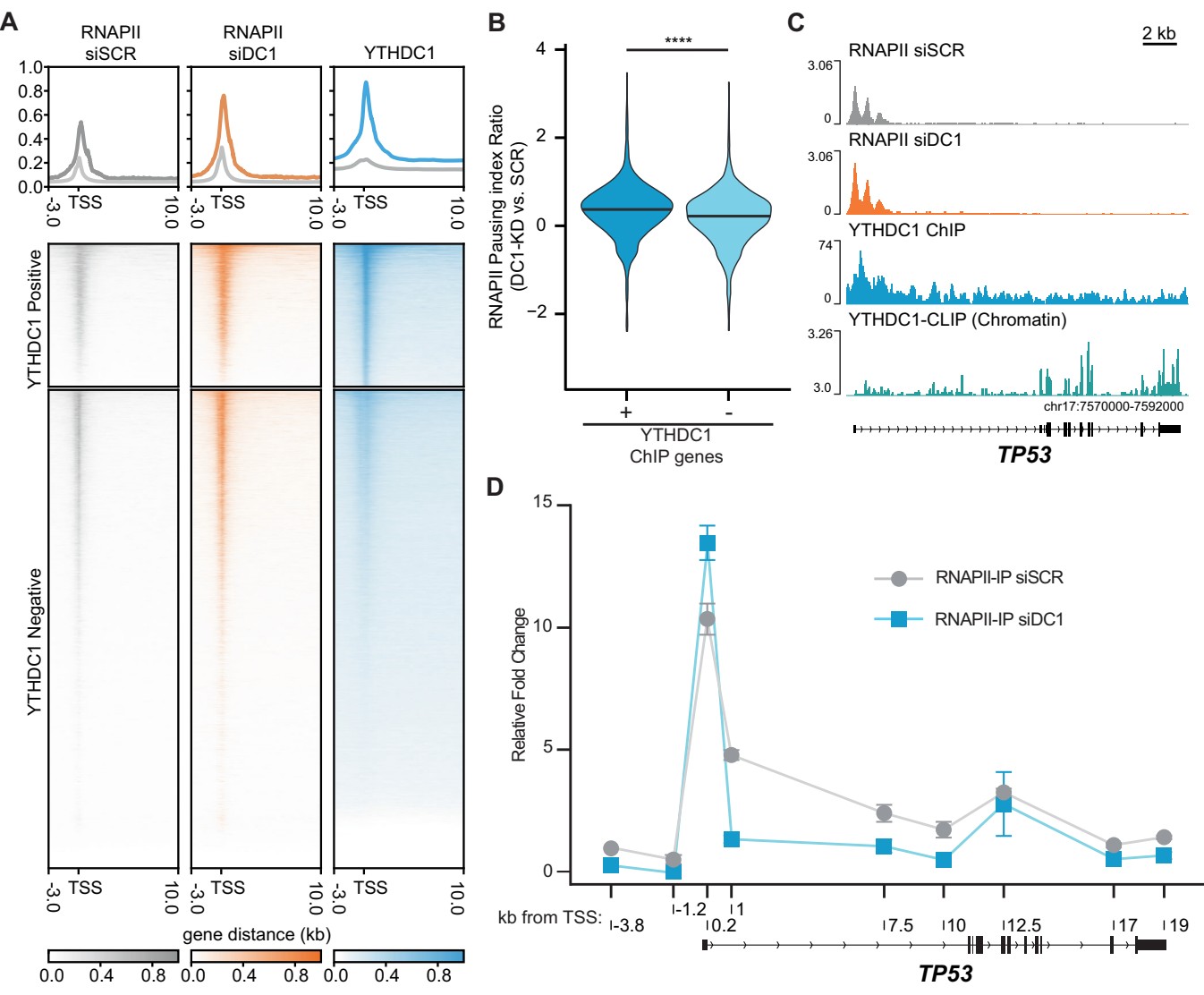

**Figure 3.  YTHDC1 modulates transcription through promoter-proximal pausing-release of RNA polymerase II.**

(**A**) Heatmap displaying coverage within 6368 YTHDC1 peaks identified using MACS2, for RNAPII and YTHDC1 ChIP-seq experiments. A total of 29435 transcription start sites (TSS) were divided into two groups based on the presence (top) or absence (bottom) of YTHDC1 in a region spanning + and – 200 bp from the TSS. RNAPII ChIP-seq experiment was performed in cells transfected with siRNA, YTHDC1 (orange), or Scramble (light grey) as a negative control, and YTHDC1 ChIP-seq experiment (blue) was performed in parallel in native conditions. (**B**) Violin plots of the distribution of expression corrected RNAPII Pausing Index ratio calculated on the ChIP-seq experiment shown in (**A**). All the actively transcribed genes were divided into two different groups based on the presence or absence of YTHDC1 peaks in the TSS (cyan and blue, respectively). Pausing index was calculated as log2 RNAPII promoter density/RNAPII gene body density for both cells transfected with scramble (SCR) siRNA and YTHDC1 siRNA. Data are presented as the distribution of Pausing Index ratio for YTHDC1 knockdown versus SCR. (**C**) Genome browser tracks for a representative region of the human *TP53* gene for the ChIP-seq experiment shown in (**A**). ChIP-seq was performed on cells transfected with scramble siRNA (light grey), and YTHDC1 siRNA (orange). YTHDC1 ChIP (cyan) and YTHDC1-CLIP (blue bondi) occupancy of the same region is shown in the bottom panel (cyan). Sequencing data were normalised as bins per million mapped reads (BPM). (**D**) Validation of ChIP-seq experiments performed in (**A–C**) by RNAPII ChIP-qPCR. Crosslinked fragmented chromatin was immunoprecipitated with an antibody against RNAPII. After precipitation, genomic DNA was extracted and analysed by qPCR with primers covering the entire *TP53* gene. Data are presented as relative fold change enrichment normalised against an intergenic region downstream from *HSP90AA1* gene of one representative replicate. The *X* axis indicates the genomic distance of the region analysed relative to the TSS of the *TP53* gene. ChIP-qPCR was performed on cells transfected with scramble siRNA (light grey), and YTHDC1 siRNA (cyan). Data information: All data are shown are representative of at least three independent experiments, except for (**D**). Data are presented as mean ± s.d. ****$P \leq 0.00001$, paired two-tailed Student's *t* test was performed in (**B**). Source data are available online for this figure.

## YTHDC1 has m⁶A-dependent and independent functions

Given the known role of YTHDC1 as a nuclear m⁶A-binding protein (Xu et al, 2014), we speculated that RNA modifications could play a pivotal role in regulating transcriptional dynamics and splicing. To address m⁶A involvement in both processes, we first investigated whether m⁶A potentially mediates the differences in RNAPII pausing observed in YTHDC1-knockdown cells (Fig. 3A–C). To that end, we analysed MeRIP data on chromatin-associated RNA (ChrMeRIP) (Xu et al, 2022) and

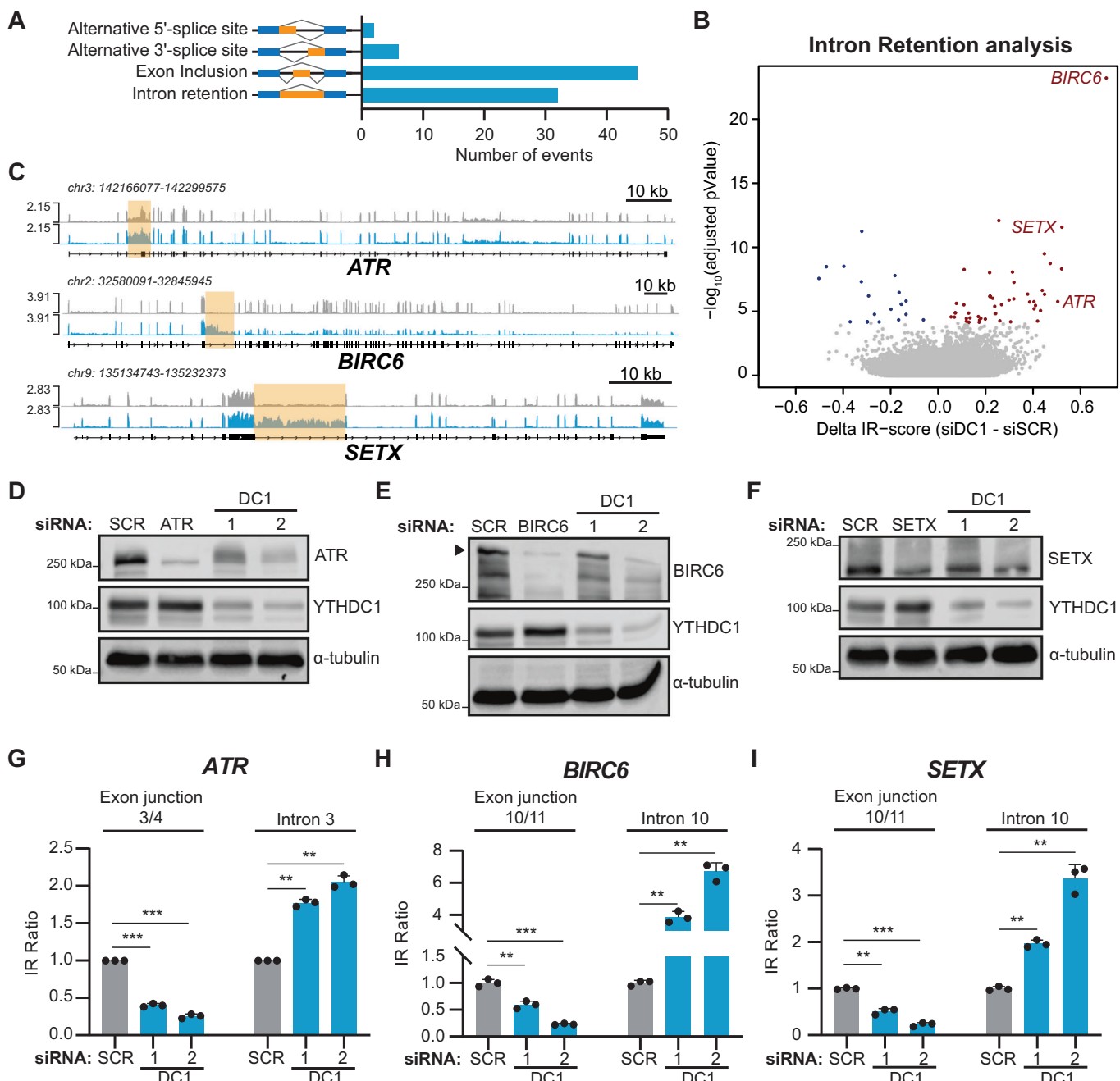

**Figure 4. YTHDC1 affects alternate splicing of factors involved in the DNA damage response.**

(A) Alternative splicing analysis. The bar plot represents the number of aberrant splicing events identified by analysing total RNA-seq of cells depleted for YTHDC1 by siRNA. (B) Volcano plot showing the Δ-IRScore versus the -log10(adjusted p value), as calculated by IRFinder, for all the introns in the Gencode v19 annotation in YTHDC1-silenced cells versus scramble (SCR) cells. Genes with introns retained with an Adjusted P value < 0.1 are considered differentially retained. Significantly upregulated or downregulated intron-retained genes in the YTHDC1 knocked-down cells labelled in blue or red colour, respectively. (C) Genome browser tracks for total RNA-seq showing reads coverage over *ATR*, *BIRC6* and *SETX* genes. Sequencing data were normalised as Fragments Per Kilobase of transcript per Million mapped reads (FPKM). Yellow boxes highlight differentially retained introns identified in (B). (D–F) Representative immunoblot of ATR (D), BIRC6 (E) or SETX (F) and YTHDC1, with α-tubulin as loading control. Cells were transfected with siRNA against target, two independent siRNA against YTHDC1 (DC1-1 and -2) or Scramble (SCR) as negative control. (G–I) RNA-level quantification of spliced or unspliced for *ATR* (G), *BIRC6* (H) or *SETX* (I) mRNA by RT-qPCR. Cells were transfected with two independent siRNA against YTHDC1 (DC1-1 and -2, cyan), or Scramble (light grey) as negative control. qPCR primers were designed to target exon junction to evaluate correct splicing, or to target the retained intron. Data information: All data are shown are representative of at least three independent experiments. Intron Retention analysis and statistics in (B) has been applied through IRFinder pipeline (Middleton et al, 2017). Data are presented as mean ± s.d. **P ≤ 0.01), ***P ≤ 0.001, paired two-tailed Student's *t* test was performed in (G–I). Source data are available online for this figure.

divided the transcriptome into m⁶A genes and non-m⁶A genes, based on the presence of ChrMeRIP peaks (Dataset EV3). Surprisingly, we did not observe a difference in the pausing index between these two categories (Fig. 5A), suggesting that the presence of YTHDC1, rather than the presence of m⁶A, is more critical for the control of elongation dynamics. Specifically, when we focused in *TP53* transcript, we identified an m⁶A peak in the beginning of its first intron (Fig. EV5A), that colocalised with the intronic long noncoding RNA (lncRNA) *ENST00000571370.1*, with very low expression and it has been poorly described by literature (Pang et al, 2019; Zhang et al, 2022). Although it remains to be determined whether this noncoding transcript could be related to *TP53* expression, overall we did not observe a relationship between m6A modification and transcriptional regulation in YTHDC1-bound genes.

Moreover, since YTHDC1 has been shown to affect splicing by selectively engaging m⁶A on RNAs (Qiao et al, 2023; Xiao et al, 2016), we hypothesised that splicing regulation should be a m⁶A-dependent process. Specifically, METTL3 knockdown, or depletion of ALKBH5 or FTO demethylases was shown to impact the alternative splicing of numerous cellular mRNAs (Achour et al, 2023; Tang et al, 2017; Zhao et al, 2014). For the purpose of assessing whether the observed splicing aberrations were due to the presence of m⁶A, we mapped the m⁶A signal from ChrMeRIP data, and analysed their relative location to retained introns regions. We observed a significantly higher m⁶A signal on the retained introns and their flanking exons compared to the rest of the gene body (Fig. 5B), similarly to the observed found for YTHDC1-CLIP peaks (Figure EV4D), suggesting this regulation is in a m⁶A-dependent manner. Moreover, when we focused on introns affected in *YTHDC1* knockdown, YTHDC1-CLIP peaks of total and chromatin-associated RNA, overlapped with ChrMeRIP peaks, asserting the m⁶A-dependency of YTHDC1 splicing regulation over *ATR*, *BIRC6* and *SETX* (Fig. EV5B).

To further investigate m⁶A dependency, we tested *TP53* transcriptional regulation and *ATR, BIRC6* and *SETX* introns processing using other approaches. Firstly, we performed siRNA-mediated knockdown of METTL3 (Fig. EV5C), the primary enzyme involved in catalysing the modification of adenosine (A) to methyladenosine (mA) (Wang et al, 2022), and quantified *TP53* mRNA and pre-mRNA levels by RT-qPCR. As we hypothesised, unlike YTHDC1 knockdown, METTL3 depletion did not cause a decrease of either mature or pre-mRNA *TP53* RNA levels (Figs. 5C and 2C), suggesting that *TP53* gene transcriptional dynamics were modulated by YTHDC1 in an m⁶A-independent manner. Then, we analysed the retention behaviour of the introns in *ATR, BIRC6* and *SETX* genes (Figs. 5D and EV5D), detecting an increase in the retention of the introns of the three DNA damage response factors.

To confirm this observation in conditions of efficient depletion of m⁶A levels, we decided to use the METTL3 inhibitor drug STM2457 (Yankova et al, 2021). We treated A549 cells with different concentrations of STM2457 molecule, confirming the reduction of m⁶A by measuring m⁶A/A ratio through thin-layer chromatography (Fig. EV5E). Notably, increasing concentrations of METTL3 inhibitor did not affect *TP53* pre-mRNA levels (Fig. 5E), suggesting that TP53 transcriptional rate remained intact even when m⁶A has been substantially reduced. However, introns of the three DDR factors were significantly retained upon m⁶A depletion (Fig. 5F). We concluded that depletion of m⁶A does not affect

RNAPII stalling/elongation properties, at least for *TP53*, but it affects splicing dynamics as observed in YTHDC1 knockdown (Fig. 4G–I), further supporting that the presence of m⁶A is essential for the RNA maturation process of these genes.

Finally, to further test the dependency of m⁶A, we expressed YTHDC1 *wild-type* and YTHDC1 *mutant* (W377A / W428A) versions of the protein in A549 cells under the control of a doxycycline-inducible system (see "Methods") (Fig. EV5F), while we knocked down the endogenous YTHDC1 (Fig. EV5G). To ensure that the observed changes are not caused by doxycycline drug itself, we treated parental A549 cells with similar doxycycline concentrations, which had no detectable effects. (Fig. EV5H,I). Then, we measured *TP53* mRNA and pre-mRNA levels, in addition to the intron retention ratio to test whether the re-introduction of YTHDC1 is able to recover the phenotype caused by YTHDC1 KD. While the re-expression of neither *wild-type* YTHDC1 nor *mutant* YTHDC1 were able to affect *TP53* transcription levels (Fig. 5G), the overexpression of *wild-type* YTHDC1 but not *mutant* YTHDC1 was able to decrease the intron retention in *ATR*, *BIRC6* and *SETX* observed upon depletion of endogenous YTHDC1 (Fig. 5H), confirming the hypothesis that splicing dynamics are regulated by YTHDC1 in m⁶A-dependent manner.

In conclusion, the results of the experiments performed with the wild-type and mutant forms of YTHDC1, together with the effect observed with the highly efficient METTL3 inhibitor, confirm the m⁶A dependency of YTHDC1 in regulating the *ATR*, *BIRC6* and *SETX* splicing events under study. While the expression of *TP53* could not be rescued by the re-introduction of YTHDC1, the experiments performed in m⁶A depletion clearly uncouple the observed phenotype from the presence of m⁶A.

## YTHDC1-dependent gene regulation ensures an effective cellular response to DNA damage

To investigate how the molecular mechanisms uncovered here translate to cellular behaviours, we performed a series of functional experiments with the aim of understanding the impact of YTHDC1 depletion on DDR. We speculated that the lack of proper activation of DNA damage responders such as p53 and ATR upon YTHDC1 depletion would lead to an increased level of double-stranded DNA breaks (DSBs) when cells are exposed to genotoxic stress. To test this hypothesis, we silenced YTHDC1, p53 and ATR using siRNAs, and subsequently treated the cells with cisplatin, a chemotherapy drug that introduces DNA breaks, followed by quantification of DNA damage using the COMET assay. As expected, YTHDC1-knockdown cells showed a significant increase in DNA damage, resulting in a higher proportion of DNA in tails than in the nucleus compared to cells transfected with scrambled siRNA (Fig. 6A,B). The independent knockdown of p53 and ATR also resulted in increased damage, albeit to a lesser degree than YTHDC1 knockdown. These results confirm that YTHDC1 is critical for the control of the DNA damage response, affecting multiple targets and necessary for preventing the accumulation of DNA damage. To evaluate defects in the DNA damage signalling cascade, we quantified γH2AX levels in YTHDC1-knockdown cells treated with cisplatin. Phosphorylation of H2AX occurs predominantly upon DSB formation and is the first step in recruiting and localising DNA repair proteins (Lowndes and Toh, 2005; Paull et al, 2000). This signalling process requires ATM, DNA-PKcs, or ATR

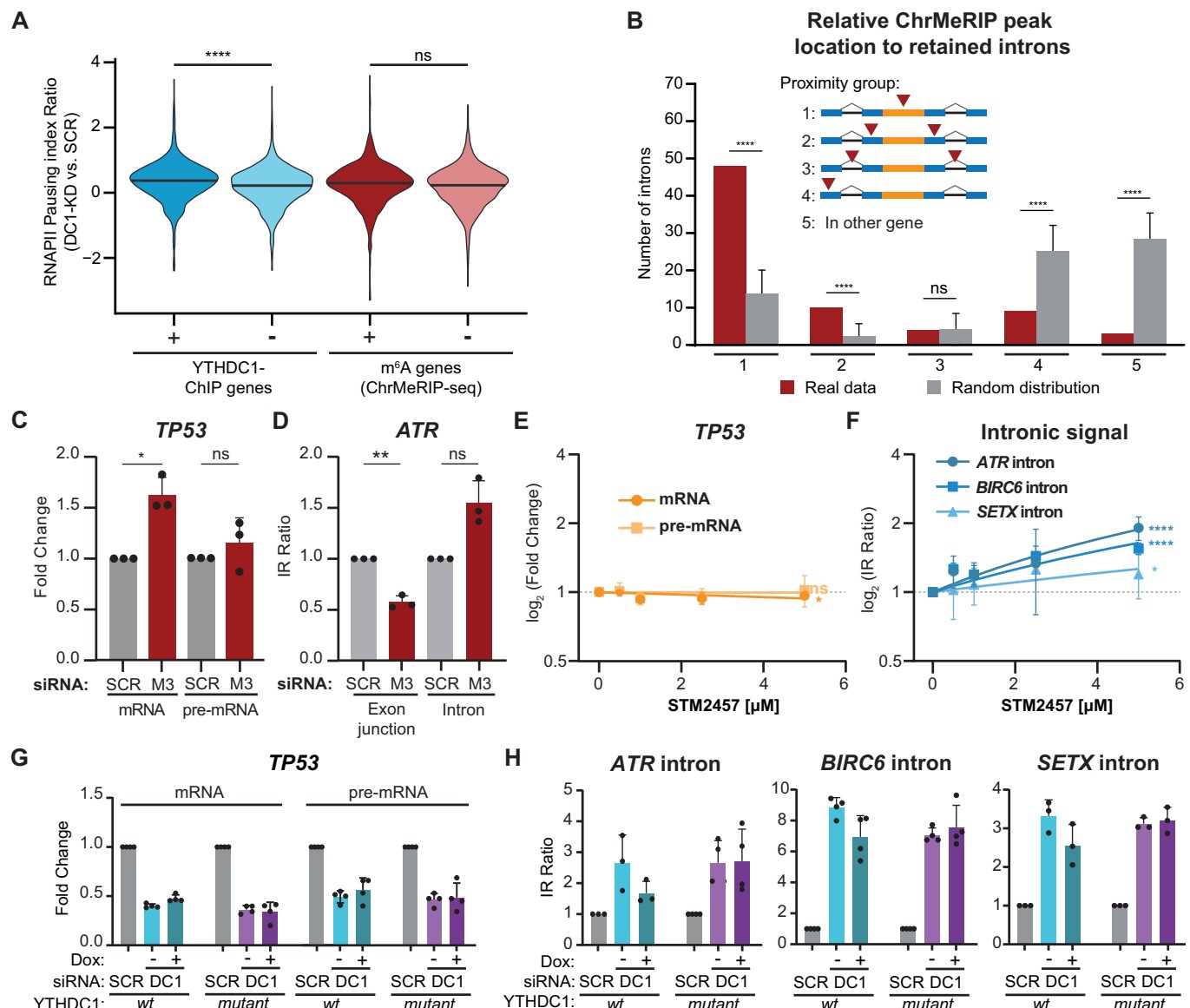

**Figure 5. YTHDC1 modulates RNAPII dynamics and splicing in a m6A-independent and dependent manner, respectively.**

(A) Violin plots of the distribution of expression corrected Pausing Index ratio calculated on the ChIP-seq experiment shown in (A). All the actively transcribed genes were divided into four different groups based on the presence or absence of YTHDC1 peaks in the TSS (cyan and blue, respectively) and based on the presence or absence of m6A in the mRNA (light and dark red, respectively). Pausing index was calculated as log2 RNAPII promoter density/RNAPII gene body density for both cells transfected with scramble (SCR) siRNA and YTHDC1 siRNA. Data are presented as distribution of Pausing Index ratio for YTHDC1 knockdown versus SCR for each category. (B) Plot showing the number of differentially retained introns (dark red bars) and 1000 random selections of GENCODE v19 introns (light bars), classified in five different groups depending on the proximity to a ChrMeRIP peak (group 1: peak inside the intron; group 2: peak in adjacent exons; group 3: peak in adjacent introns; group 4: peak anywhere inside the gene; group 5: no peak inside the gene). Error bars represent twice the standard deviation. (C, D) RNA-level quantification of mature mRNA and pre-mRNA of *TP53* (C) and intron retention of *ATR* (D) by RT-qPCR. Cells were transfected with siRNA against METTL3 (dark red), or Scramble (light grey) as a negative control. Statistical analysis was performed by paired two-tailed Student's *t* test METTL3-KD versus scramble (SCR) control. ns, not significant $P \geq 0.05$, *$P \leq 0.05$. Data are presented as mean ± s.d. (E) RNA-level quantification of mature mRNA (orange) and pre-mRNA (light orange) by RT-qPCR for *TP53* in A549 cells treated with increasing concentrations of STM2457. (F) Intron retention measurement for *ATR*, *BIRC6* and *SETX* by RT-qPCR, normalising signal from their respective upstream exon junction in A549 cells treated with increasing concentrations of STM2457. (G) RNA-level quantification of *TP53* mRNA and pre-mRNA levels in A549 cells that has been transfected with siRNA for endogenous YTHDC1 and induced expression of wt and mutant. (H) RNA-level quantification of intron retention for *ATR*, *BIRC6* and *SETX* genes in A549 cells that has been transfected with siRNA for endogenous YTHDC1 and induced expression of wt and mutant. Data information: All data are shown are representative of at least three independent experiments. For (A), data are presented as mean ± s.d. ns, not significant $P > 0.01$, ****$P \leq 0.00001$, paired two-tailed Student's *t* test was performed. Empirical *P* value for (B) was obtained by randomising 100 times the selected introns to compared. For (C–H), data are presented as mean ± s.d. ns, not significant $P > 0.05$, *$P \leq 0.05$, **$P \leq 0.01$, ****$P \leq 0.0001$, paired two-tailed Student's *t* test was performed. Source data are available online for this figure.

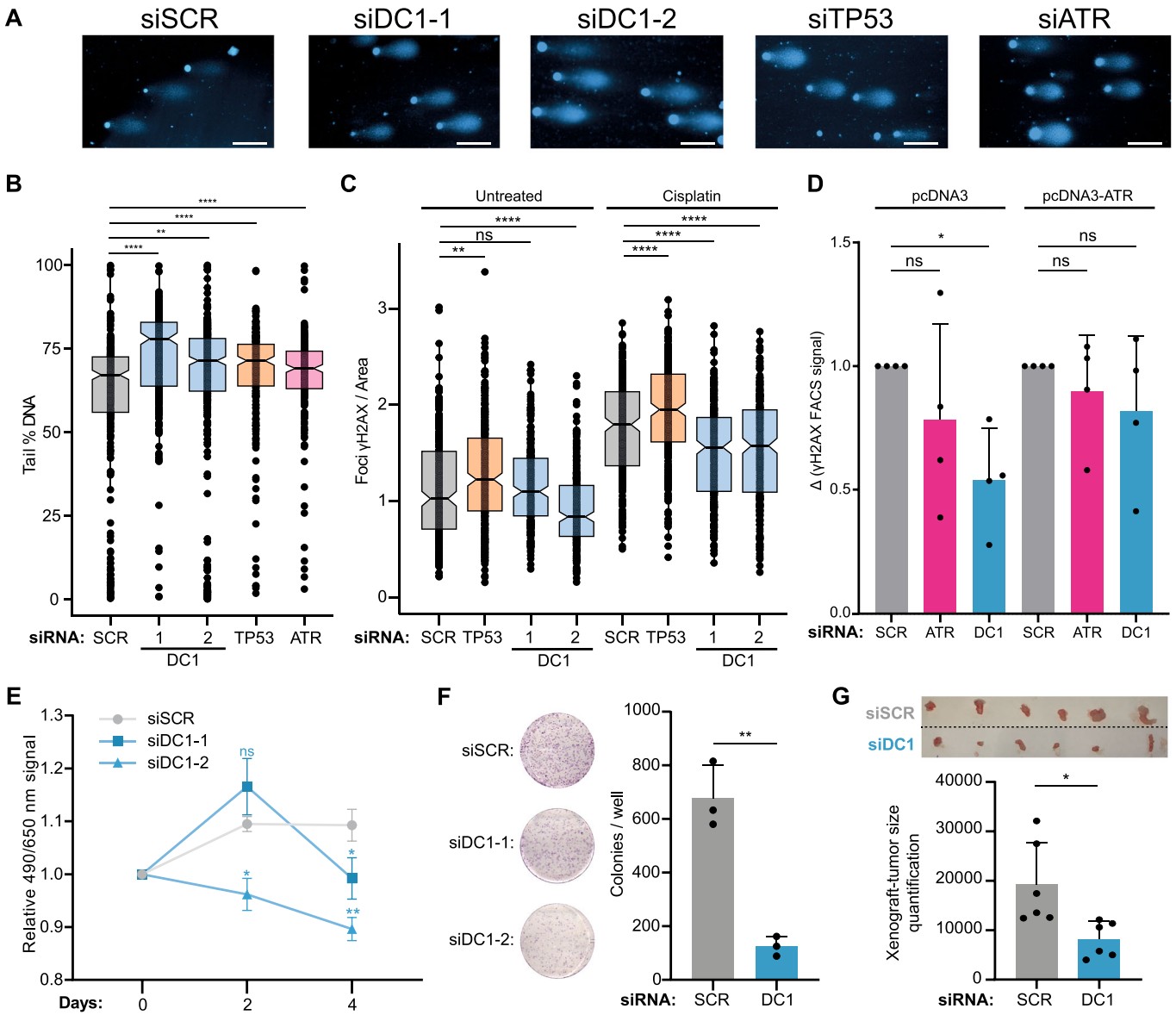

**Figure 6. YTHDC1-dependent gene regulation ensures an effective cellular response to DNA damage.**

(A) Representative images of COMET assay. Cells were transfected with two independent siRNA against *YTHDC1* (DC1-1 and -2), siRNA against *TP53*, siRNA against *ATR* or Scramble (SCR) as negative control. After silencing cells were treated with cisplatin to induce DNA damage. One cell per condition was selected as representative of the total cells analysed. 100 μm are represented in scale bars. (B) Quantification of COMET assay experiments. Boxplot showing the percentage of DNA in the tail for the conditions described in (A). (C) Immunofluorescence quantification of γ-H2AX. Cells were transfected with two independent siRNA against *YTHDC1* (DC1-1 and -2), siRNA against *TP53*, or Scramble (SCR) as negative control. After silencing cells were treated with cisplatin to induce DNA damage. After γ-H2AX immunostaining the cells were imaged. The boxplots represent the number of foci per area obtained analysing 100 images per replicate. (D) Flow cytometry quantification of γ-H2AX. Cells were transfected with siRNA against *YTHDC1*, siRNA against *ATR*, or Scramble (SCR) as negative control. After silencing cells were transfected with a plasmid expressing ATR to rescue the phenotype, or an empty plasmid as a control, then treated with cisplatin to induce DNA damage. After γ-H2AX immunostaining the cells were analysed by flow cytometry. The bar plot represents the average delta signal of γ-H2AX of treated versus cisplatin-treated cells. (E) Cell proliferation determined by MTS assay for cells transfected with two independent siRNA against YTHDC1 (DC1-1 and -2), or Scramble (SCR) as negative control. After silencing, viability was measured by absorbance at days indicated on the *X* axis. (F) Colony formation assay in cells transfected with siRNA against *YTHDC1*, or Scramble (SCR) as negative control. The bar plot represents the average number of colonies obtained. (G) Volume of tumours obtained after subcutaneous injection of A549 transfected with siRNA against *YTHDC1*, or Scramble (SCR) as negative control in immune-compromised mice. Twenty-seven days post injection the mice were sacrificed and the relative volume was quantified by imaging, image of tumours above of graph. The bar plot represents the average relative size of $n = 6$ samples for each condition. Data information: All data are shown are representative of at least three independent experiments, except for (C). Data are presented as mean ± s.d. ns, not significant $P > 0.05$, *$P \leq 0.05$, **$P \leq 0.01$, ****$P \leq 0.0001$, paired two-tailed Student's *t* test was performed in (B–G). Boxplots represent the 25th and 75th percentiles, with mean as wider line, and outer line encompass values not considered as outlier, in (B, C). Source data are available online for this figure.

activation, depending on the context or type of DNA lesion (An et al, 2010; Burma et al, 2001; Ward and Chen, 2001). We quantified γH2AX levels by immunofluorescence in cells treated with cisplatin and silenced with two independent siRNA for YTHDC1, using p53 knockdown as a positive and scramble siRNA as a negative control. As expected, p53-knockdown cells showed increased levels of γH2AX, indicating that these cells accumulate more damage, as we previously observed with the comet assay, and that this damage sufficiently activates the signalling kinase cascade. In contrast, both YTHDC1-knockdown samples showed a significant decrease in fluorescence signals, suggesting that the cells were not able to properly activate the signalling cascade of kinases that ultimately result in the phosphorylation of the histone variant (Fig. 6C). We speculated that this lack of signalling was, at least in part, due to the ATR defects that we observed upon YTHDC1 depletion. To validate this hypothesis, we performed a rescue experiment in which we restored ATR expression in YTHDC1-silenced, cisplatin-treated cells by transfecting a plasmid coding for ATR, resulting in overexpression of the protein. As a control, we knocked down ATR using siRNA. Interestingly, the ATR-knockdown cells had a decreased level of H2AX phosphorylation, measured by fluorescent activated cell sorting, mimicking the phenotype we observed in YTHDC1-knockdown cells. The overexpression of ATR, in both ATR- and YTHDC1-silenced cells, rescued the phenotype, restoring normal levels of γH2AX fluorescence indicating DNA damage signalling, supporting our hypothesis (Fig. 6D). When lesions accumulate and are unrepaired or there are defects in the DNA repair system, DNA damage is often coupled with apoptosis and cell growth defects. Therefore, we investigated the proliferative capacity of YTHDC1-knockdown A549 cells in different experimental contexts. In vitro experiments confirmed that YTHDC1 is necessary to sustain cell growth and colony formation (Fig. 6E,F). To validate this finding in a more physiological environment, we injected YTHDC1-knockdown or scrambled siRNA control A549 cells into immunosuppressed mice to allow tumour formation. After 27 days post injection, we sacrificed the animals and surgically removed the tumours to evaluate their size. We observed tumour formation in all the animals ($n = 12$), with a significant decrease in size for those tumours deriving from YTHDC1-deficient cells (Fig. 6G), corroborating the in vitro experiments. Thus, our findings reveal that YTHDC1 impacts the DNA damage response through both m⁶A-dependent and m⁶A-independent mechanisms, regulating different steps of transcriptional and mRNA maturation processes for genes encoding crucial proteins that are responsible for maintaining genome stability.

## Discussion

The complex relationship between RNA metabolism and the DNA damage response is a major challenge to resolve due to the lack of a comprehensive understanding of how the molecular and biochemical processes affecting RNA can impact DNA response. Advances in CRISPR–Cas9 loss-of-function screening have allowed the systematic discovery of the intrinsic gene requirements for important cellular functions, including DDR (Cuella-Martin et al, 2021; Zhao et al, 2023). Here, our screening strategy enabled the high-throughput discovery of RNA modifiers required for initiating

p53 activation and controlling the DDR. To our knowledge, this is the first screening specifically designed to address specifically this question. In the present study, we identified YTHDC1 as a pivotal regulator of DDR through its ability to control the proper transcription of several DNA damage response factors, regulating different aspects of this intricate, multi-step process. First, we demonstrated that YTHDC1 is essential for maintaining physiological levels of *TP53* mRNA, through regulation of the transcription machinery. In particular, we found that YTHDC1 promotes promoter-proximal RNAPII pausing-release in *TP53* and many other genes, allowing the cell to productively transcribe them. Second, we detected widespread alternative splicing defects in YTHDC1-deficient cells, with genes related to the DNA damage response having the most prevalent levels of intron retention. Interestingly, our data suggest that these two mechanisms are differentially impacted by the presence of m⁶A in the processed RNAs. We showed that the transcription repression of *TP53* is an m⁶A-independent process. Indeed, cells depleted for this modification through METTL3 knockdown or inhibition did not phenocopy the effect of YTHDC1 depletion on *TP53* mRNA level (Figs. 2C, 5C and 5E). Providing wild-type or mutant YTHDC1 protein was not sufficient to recover *TP53*. We speculate that the disrupted chromatin environment around *TP53* TSS cannot be restored by YTHDC1 expression alone. On the other hand, the splicing aberrations that we observed in *ATR*, *BIRC6* and *SETX* transcripts upon YTHDC1 depletion could be mimicked by METTL3 depletion (Figs. 5D and EV5D) and METTL3 inhibition (Fig. 5E). Note that when we looked at other m⁶A factors in CRISPR screening (Fig. EV1G), no trend was observed for either m⁶A writers or erasers, consistent with the m⁶A-independency of this process. Moreover, YTHDF proteins have been widely described to be involved in mRNA stability (Zaccara and Jaffrey, 2020), but they did not lead to any change in the p53-reporter signal, suggesting that these proteins are not involved in the p53 response. Thus, while other m⁶A regulators may be acting at multiple levels of the p53 response (Raj et al, 2022), our results show the distinct YTHDC1 functions in this pathway are not only dependent on m⁶A.

Several studies have investigated the connection between YTHDC1 and alternative splicing. While it has been reported that YTHDC1 can directly regulate mRNA splicing of mRNAs by binding through m⁶A (Achour et al, 2023; Haussmann et al, 2016; Xiao et al, 2016), it has also been shown that YTHDC1 sequestration by m⁶A-modified noncoding RNAs indirectly results on mRNA splicing alterations (Ninomiya et al, 2021; Timcheva et al, 2022) (Lee et al, 2021; Li et al, 2022). Among these, it has been shown that the lncRNA *MALAT1* acts as a scaffold that recruits YTHDC1 to nuclear speckles, regulating the expression of several genes (Wang et al, 2021). However, we did not observe this dependency for phenotypes described in own unpublished work. Our study shows that the retention of introns of the DNA damage genes *ATR*, *BIR6* and *SETX* is linked not only to the presence of m⁶A but also to the binding of YTHDC1 determined by CLIP, suggesting a direct role of YTHDC1 in the processing of these mRNAs.

Dependence on m⁶A has been shown for YTHDF proteins, which promote m⁶A-mediated degradation in proportion to the number of m⁶A sites in the transcript (Boo et al, 2022; Zaccara and Jaffrey, 2020; Zou et al, 2023). In contrast, our data suggest an

m[6]A-independent effect of YTHDC1 in mediating the RNAPII pausing/release process, as pause-release did not show a clear correlation with the presence of m[6]A sites despite the binding of YTHDC1 to the regulated genes (Fig. 5A). While it has been proposed that m[6]A modification is linked to transcriptional elongation by avoiding the binding of INTS11 to the nascent transcripts, the direct implication of YTHDC1 in the described mechanism remains to be clarified (Xu et al, 2022). On the other hand, it has been shown that YTHDC1 has the capacity to bind to RNA even in the albescence of m[6]A mark (Cheng et al, 2021; Lee et al, 2021; Roundtree et al, 2017), potentially binding to nascent RNAs regardless of their methylated status to favour transcriptional elongation. Nevertheless, although the presence of m[6]A may be dispensable, it likely favours YTHDC1-RNA binding in some contexts. Beyond YTHDC1-RNA interaction, the transcriptional role of YTHDC1 is dependent on specific protein interactions established in the environment of RNAPII (Dou et al, 2023). Moreover, transcriptional elongation and splicing are functionally coupled (Caizzi et al, 2021), and also linked to nuclear architecture. Thus, the role of YTHDC1 in these processes may not be uncoupled. However, we were able to find differential distribution in the binding of YTHDC1 to the genome and chromatin-associated RNA (Fig. EV4E), which could be a clue of spatial separation of both processes (Tammer et al, 2022). The full elucidation of such interplay will shed light on the function of YTHDC1 in transcriptional control.

By analysing the dysregulation of transcriptional dynamics, we demonstrate that YTHDC1 is critical for the full induction of DDR. YTHDC1 knockdown in cells exposed to the genotoxic compound cisplatin supports this conclusion, showing that the YTHDC1 depletion results in the accumulation of double-strand breaks (Fig. 6A,B). While some of the DNA damage may be attributed to the increased presence of R-loops (Liu et al, 2020), here we unveil key defects in the signalling cascade. This cascade plays an essential role in establishing the first step in detecting the damage and subsequently activating checkpoints to delay cell cycle progression, allowing the cell to recruit the proper repair machinery. In summary, our findings highlight the importance of YTHDC1 in regulating the DDR to guarantee cell survival through controlled and efficient DNA repair. However, the nature of the DNA damage and p53 responses is highly dependent on the cellular and genetic context, showing key differences between primary and tumour cells (Kastenhuber and Lowe, 2017), which may also explain the differences observed between tumour types (Fig. EV2F–H). In addition, the recognition that there are both m[6]A-dependent and m[6]A-independent mechanisms involved in DDR and further understanding of these pathways may be helpful for the development of potential novel therapies. Aberrant m[6]A modification is prevalent in various types of cancer and is associated with patient prognosis (Hu et al, 2021; Li et al, 2022; Meng et al, 2020). The dysregulation of m[6]A modification also critically regulates malignant processes, including proliferation, metastasis, tumour stemness, and drug resistance (Li et al, 2022; Liu et al, 2018; Meng et al, 2020; Shriwas et al, 2021). For these reasons, several pharmaceutical companies are focused on moving m[6]A modulators to the clinic. We therefore believe that the identification of m[6]A-direct effects of important mRNA regulation processes in critical functional pathways will promote the development of novel, specific, and effective m[6]A modification inhibitors and activators for potential clinical use in the near future.

## Methods

### Cell culture and treatment

The following human cell lines were employed for this study: A549-p21-Reporter (kindly provided by Dr. Lani F. Wu laboratory), A549 (purchased from ATCC), HeLa (kindly provided by Dr. Tomás Aragon laboratory), HEK293T (purchased from ATCC) and MCF7 (RRID: CVCL_0031) cell lines, which were cultured in DMEM medium (GIBCO), and HCT116 (kindly provided by Dr. Vogelstein's laboratory) cell line was cultured in RPMI-1640 medium (GIBCO). All mediums were supplemented with 10% foetal bovine serum (GIBCO) and 1× penicillin/streptomycin (Lonza). Cells were maintained at 37 °C in the presence of 5% $CO_2$.

Stable expression SpCas9 was achieved by lentiviral infection of p21-reporter A549 cell line. Cells were transduced 48 h with a lentivirus carrying lentiCas9-Blast vector (Addgene) in complete DMEM medium supplemented with 10 μg/mL polybrene (Santa Cruz). After 48 h, cells were washed with PBS (GIBCO) and selected with complete DMEM medium supplemented with 10 μg/mL Blasticidin (InvivoGen) for 10 days and later maintained with the same medium. 7 days before performing CRISPR Screening, p21-reporter-SpCas9 A549 cells were cultured in complete DMEM without Blasticidin.

Same procedure was followed to generate YTHDC1 *wild-type* and *mutant* A549 cell lines. In this case, we generated lentivirus using Lenti-X™ Tet-One™ system (Takara). To induce over-expression of YTHDC1 *wild-type* and *mutant*, 20 ng/mL and 200 ng/mL Doxycycline were used for 2 days, respectively, to equal protein levels.

To generate the YTHDC1-KO clone, A549 cells were transfected with px459-Cas9 plasmid (Addgene), carrying sgRNA for YTHDC1 "ATTCTTATAAGGTTCTCTGG". Clones were checked by Western blot of YTHDC1 and later Sanger sequencing to confirm frameshifting events.

For p53 activation, cells were treated with 10 μM Nutlin-3a (Sigma-Aldrich) for 16 h prior to RNA or protein extraction. To induce DNA damage, cells were treated with 10 μM Cisplatin (Merck KGaA, Darmstadt, Germany) for 12 h or 16.6 μM Cisplatin for 2 h plus 12 h of recovery, prior to protein extraction or cell fixation to immunofluorescence.

For transcription inhibition, A549 cells were treated with Actinomycin D (Sigma) at a final concentration of 10 μg/mL at different time points. To analyse RNA stability in YTHDC1-depleted condition, A549 cells were transfected with corresponding siRNAs 64 h before Actinomycin D (ActD) treatment. RNA levels were normalised to in vitro transcribed Luciferase RNA (LUC), used as an added spike-in in samples before starting RNA extraction. To inhibit METTL3, A549 cells were treated with different concentrations of STM2457 molecule (Sigma-Aldrich) for 3 days. Negative control conditions were treated with DMSO.

### Cellular transfection

Transfection reaction was carried out in Opti-MEM medium (GIBCO) using RNAiMax Lipofectamine Transfection Reagent (Invitrogen) or Lipofectamine 3000 Transfection Reagent (Invitrogen) to transfect siRNA or plasmid, respectively, following the manufacturer instructions.

For RNA knockdown, siRNAs were transfected 48–72 h at a final concentration of 40 nM. siRNA sequences were obtained from previous studies or designed using the "i-Score Designer" and "siDirect v.2" designing tools, and purchased from Sigma (Dataset EV4).

For exogenous ATR overexpression, pcDNA3-ATR WT (Addgene) was transfected with 1 μg of plasmid during 48 h.

For rescue experiments, 40 nM siRNAs were transfected for 72 h with RNAiMax Lipofectamine (Invitrogen), and later 1 μg plasmids were transfected for 48 h using Lipofectamine 3000 (Invitrogen), prior to cell fixation for flow cytometry.

## Lentivirus production and cellular transduction

Lentivirus were produced by co-transfecting the transfer vector of interest with psPAX2 (Addgene) and pMD2.G (Addgene) packaging vectors into HEK293T cells using Lipofectamine 2000 Transfection Reagent (Invitrogen). After 6 h of transfection, culture medium was changed and lentivirus-containing supernatant was collected 48 h after transfection and filtered through a 0.45 μm low-protein-binding filter (VWR).

Target cells were plated in six-well culture plates with a 60–80% confluency, lentivirus were mixed in a ratio 1:1 with fresh complete culture medium and supplemented with 10 μg/mL polybrene (Santa Cruz) to transduce them. 48 h post-infection, cells were pelleted to remove excess lentivirus and plated again in fresh complete medium. When cells began to grow normally, cells were selected with corresponding antibiotics. In case of CRISPR-Screening, each lentiviral library batch was previously tested to determine the amount of lentivirus to obtain a multiplicity of infection (MOI) between 0.3 and 0.6; and also, there were not any selection with antibiotics.

## CRISPR–Cas9 screen and data analysis

P21-reporter A549 cells were transduced with a lentivirus construct expressing SpCas9 as previously described. After blasticidin selection, we confirmed SpCas9 expression through Western blot (Fig. EV1B). Cells were maintained with DMEM medium supplemented with 10% foetal bovine serum, 1× penicillin/streptomycin and 10 μg/mL blasticidin.

CRISPR-sgRNA library was based on Human CRISPR Brunello Knockout Library sequences (Doench et al, 2016). This pooled-sgRNA library contains 1828 sgRNAs, comprising 200 non-target sgRNAs as control and 1628 sgRNAs targeting 407 protein-coding genes, 4 sgRNAs per target (Dataset EV1). It was synthesised and cloned into CRISPRseq-BFP-backbone (Addgene). The quality of the pooled-CRISPR-sgRNA library was verified by Illumina sequencing, taking into account depth (more than 100 reads per sgRNA), overall representation (less than 0.5% of sgRNAs have no reads) and uniformity (less than 10-fold difference between the 90th and 10th percentile of sgRNAs).

Reporter-based CRISPR–Cas9 screening was performed as it was previously described in (Shalem et al, 2014) with few modifications. We performed this screening twice to assess reproducibility. We transduced a minimum of 6.1 million of cells to obtain a coverage of 1000 copies per sgRNAs with a multiplicity of infection (MOI) between 0.3 and 0.6, which was measured by EBPF2 fluorescent signal in FACSAria Ilu sorter. 1 million p21-

Reporter/SpCas9 A549 cells per 100-mm plate were plated to have a 60–80% of confluency. Cells were transduced with a previously tested pooled-CRISPR-sgRNA library lentivirus batch. 48 h later, cells were pelleted to remove excess of lentivirus and splitted up in 150 mm plate to avoid confluency problems. We allowed the cells to be incubated for a total of 4 days after infection to ensure that SpCas9 had performed its activity and target-protein levels had been depleted. 4 days post-transduction, cells were treated with 10 μM Nutlin-3a for 16 h. Cells were harvested and pelleted in ice-cold Sorting Buffer (1× PBS, 5% foetal bovine serum, 5 mM EDTA) to a final concentration of 1 million cells per millilitre and sorted using FACSAria Ilu (EBFP2 signal was recognised with $\lambda$Ex/Em = 405/450–440 nm and mVenus signal was recognised $\lambda$Ex/Em = 488/530 nm). A representative fraction of EBFP2+ was selected from Mock population (~1 million of cells), as well as 25% of the population with EBFP2 + /p21-Reportermore signal and 25% of the population with EBFP2 + /p21-Reporterless signal (~1 million of cells per condition), defined as +25% and −25% population, respectively.

Genomic DNA was extracted with phenol:chloroform:isoamyl alcohol (25:24:1) (Sigma-Aldrich, #516726), according to the manufacturer's instructions. Genome-integrated-CRISPR guides were amplified with custom primers (Dataset EV4), followed by a standard sequencing library preparation protocol. Samples were sequenced at Novogene (150 bp paired-end, sgRNA).

We used the Model-based Analysis of Genome-wide CRISPR–Cas9 Knockout (MAGeCK) (version 0.5.9.2) (Li et al, 2014) for prioritising single-guide RNAs, genes and pathways in the genome-scale CRISPR–Cas9 knockout screen designed to detect proteins involved in RNA modifications which modulate p53 response through the use of p21-reporter A549 cell line.

In the MAGeCK algorithm, the raw read counts corresponding to single-guided RNAs (sgRNAs) from different experiments are first normalised using median normalisation and mean-variance modelling is used to capture the relationship of mean and variance in replicates. The MAGeCK count command was run to evaluate the quality of the data and obtain QC measurements of the fastq files. The statistical significance of each sgRNA is calculated using the learned mean-variance model. The essential genes (both positively and negatively selected) are then identified by looking for genes whose sgRNAs are ranked significantly higher using robust rank aggregation (RRA) algorithm. Finally, RRA is applied to the ranked list of genes to identify enriched pathways.

We also used CASPR (Bergadà-Pijuan et al, 2020) to obtain the consensus prediction generated by MAGeCK and the PBNPA algorithm (Jia et al, 2017), which uses an empirical model to identify significantly enriched or depleted targets.

## Pathway alteration analysis

We used Pathtracer (v. 0.1.0) (Nygård et al, 2019) to detect pathway activity alteration in tumour vs healthy tissue. The Pathtracer algorithm projects the samples onto a low-dimensional manifold in the subspace spanned by the genes belonging to a given pathway. For each sample, a score is next found by calculating the distance between each projected sample and the projection of a subgroup of reference samples.

PathTracer was applied to Lung Carcinoma samples with a *TP53* wild-type genotype. Lung Carcinoma samples were formed by

grouping TCGA-LUAD (244 cancer and 59 healthy tissue samples) and TCGA-LUSC (82 cancer and 49 healthy tissue samples) patients. Pathways for *Homo sapiens* were downloaded from the Reactome database (Gillespie et al, 2022) using the reactome.db R/Bioconductor package (v. 1.68.0) (Ligtenberg 2019). From these, the Transcriptional Regulation by TP53 (R-HSA-3700989) pathway was used to obtain pathway alteration scores for the aforementioned TCGA-LUAD samples. Pearson correlation analysis was performed between the pathways alteration scores obtained with Pathtracer and the expression of the 407 screening candidate genes plus p53 targets in the KEGG p53 signalling pathway.

As we are interested in genes downstream of p53, we selected those genes from the following pathways included within R-HSA-3700989: TP53 Regulates Metabolic Genes (R-HSA-5628897), TP53 Regulates Transcription of Genes Involved in G1 Cell Cycle Arrest (R-HSA-6804116), TP53 Regulates Transcription of Genes Involved in G2 Cell Cycle Arrest (R-HSA-6804114), TP53 Regulates Transcription of Caspase Activators and Caspases (R-HSA-6803207), TP53 Regulates Transcription of Genes Involved in Cytochrome C Release (R-HSA-6803204),TP53 Regulates Transcription of Death Receptors and Ligands (R-HSA-6803211) and TP53 Regulates Transcription of DNA Repair Genes (R-HSA-6796648). We also selected p53-Dependent G1/S DNA damage checkpoint (R-HSA-69580), ending up with a total of 264 unique genes that were used to subset the expression matrices from TCGA.

## Phenotypical assays

In the proliferation assay, we seeded 1000 cells/well in 96-well plates. At 0, 2 and 4 days since cells were re-plated, we supplemented culture medium with 10 μL of CellTiter96 Aqueous Non-Radioactive Cell Proliferation Assay (MTS) kit (Promega) reagent for 1 h. Subsequently, we used a SPECTROStar Nano 96-well plate reader (BMG Labtech) to measure the absorbance ratio at $\lambda = 490/650$ nm by spectrophotometry.

For colony-formation assay, we plated 2000 cell/well in six-well plates the day after siRNA transfection. We maintained them in complete medium for 10 days. Later, cells were fixed with 0.5% glutaraldehyde in 1× PBS for 20 min and stained with 0.1% Crystal violet solution (Sigma) for 30 min. We performed multiple washes with MilliQ H2O to remove excess of crystal violet. After air-drying plates, colonies were manually counted to assess colony-formation capacity.

The animal experiments were carried out at the animal facility at CIMA, registered as a centre that uses animals for experimentation purposes in accordance with RD 53/2013. All experimental procedures has approval by the Ethics Committee for Animal Testing of the University of Navarra (Ref. number: 006-20). For cell line-derived xenograft (CDX) mice, $2.5 \times 10^6$. A549 cells were transfected for each condition and later resuspended in 100 μL of complete DMEM medium mixed with Matrigel Matrix (Corning) in a ratio 1:1. This cell-Matrigel mixture was injected subcutaneously in the flank of 6–12 weeks old male and female BALB/cA-Rag2 − /−γc − /− immunodeficient mice (RRID: IMSR_JAX:014593). 6 mice were used per condition. To monitor tumour growth, the size of the tumours was measured on specified days over a period of 24 days at the indicated days using a electronic precision caliper. The tumour volume (V) was calculated using the following formula: $V = \pi/6 \times width^2 \times length$. The mice

were sacrificed 27 days after injection, and xenograft tumours were extracted. Images of the tumours were captured for further analysis. The tumour size was quantified by analysis of the pixel area with ImageJ in the captured pictures.

To perform alkaline comet assay, we followed the previously reported protocol (REF) with a few modifications. After transfection and treatment incubations, we resuspended 25,000 A549 cells per condition in 300 μL of 0.5% Agarose LM-GQT (Conda) in 1× PBS. This cell suspension was quickly deposited onto pre-coated slides with a 1% agarose base layer. A coverslip was placed over the cell suspension to create a thin layer of embedded A549 cells in 0.5% agarose. Once the agarose solution solidified, the coverslips were carefully removed.

Next, we immersed slides in ice-cold Lysis Buffer (2.5 M NaCl, 10 mM Tris pH 8, 100 mM EDTA, 1% Triton X-100, [pH 10]) for at least 1 h. After lysis step, the slides were transferred to an electrophoretic chamber containing Alkaline Buffer (300 mM NaOH, 1 mM EDTA, [pH >13]) and incubated for 30 min. Subsequently, electrophoresis was performed under the following conditions: 0.7 V/cm (distance between electrodes) and 300 mA for 1 h.

To inactivate the alkaline buffer, the slides were washed three times with Inactivation Buffer (400 mM Tris [pH 7.5]), followed by a rinse with MiliQ Water and another rinse with absolute ethanol. Slides were rehydrated and mounted with ProLong® Gold Antifade Reagent with DAPI (Cell Signalling). Pictures were collected with a Zeiss Axio Imager M1 automated optical microscope running Zen 2 core imaging software (Zeiss). The collected images were analysed with OpenComet software tool in ImageJ (Gyori et al, 2014).

## Immunofluorescence

A549 cells were cultured on glass coverslips and transfected with the respective siRNAs. After 72 h of transfection, the cells were fixed using 3% paraformaldehyde (Electron Microscopy Sciences) in 1× PBS for 10 min. Next, slides were washed and permeabilized twice with Wash Buffer (0.5% IGEPAL, 0.01% Na-Azide in PBS 1×) for 10 min at room temperature. Then, we proceed to block fixed cells with 10% Foetal Bovine Serum (FBS) in PBS for 20 min. Subsequently, cells were incubated with the anti-Phospho-Histone H2A.X (Ser139) antibody (Cell Signalling #2577) for 30 min at room temperature. Once incubation has finished, we washed twice with Wash Buffer. Afterwards, we incubated cells with a secondary donkey anti-Rabbit antibody conjugated to Alexa-Fluor® 488 (Thermo Fisher) for an additional 30 min at room temperature. The coverslips were then washed three times with a washing buffer, briefly air-dried, and mounted using ProLong® Gold Antifade Reagent with DAPI (Cell Signalling).

Cellular imaging was performed using Zeiss Axio Imager M1 automated microscope with either a ×20 or ×40 objective, and images were captured using the ZEN microscopy software. Quantification of foci was carried out using ImageJ.

## RNA extraction and real-time quantitative PCR (RT-qPCR)

For RNA extraction, two different methods were employed based on the intended use of the RNA. The TRI Reagent (Sigma) or the Maxwell® RSC simplyRNA Tissue kit (Promega) were utilised,

depending on whether the RNA was intended exclusively for RT-qPCR or sequencing library preparation, respectively.

RNA extraction by TRI Reagent was performed following the manufacturer's instructions, followed by RNA precipitation using 2-propanol. For RT-qPCR, 1 µg of RNA was treated with DNAse I (Invitrogen) for 15 min at 25 °C, and later, DNAse I activity inhibited with ~2.5 mM EDTA at 65 °C for 10 min. Conversely, for the Maxwell® kit, the extraction protocol provided by the manufacturer was followed.

Once we have purified RNA, we conducted reverse transcription of 1 µg RNA using the High-Capacity cDNA Reverse Transcription kit (Applied Biosystem), containing random hexamer primers, following the manufacturer's indications. The resulting complementary DNA (cDNA) was analysed by quantitative PCR using iTaq Universal SYBR Green Supermix (Bio-Rad) in a ViiATM 7 Real-Time PCR System (Thermo Fisher). The qPCR programme consisted in an intial step at 95 °C for 15 min at 95 °C, followed by 40 cycles of 95 °C for 30 s and 60 °C for 30 s. Every reaction was performed in quadruplicates using the primers listed in Dataset EV4. The RNA levels of HPRT1, GAPDH and U6 were used for normalisation, depending on the depleted target. For intron retention analysis, a downstream region of the target itself was used to normalise in those experiments. Statistical analysis of relative RNA levels was performed by two-tailed unpaired $t$ test.

## Protein extraction and immunoblot (western blot)

Isolated cells were lysed for 30 min at 4 °C with RIPA Buffer (150 mM NaCl, 50 mM Tris-HCl [pH 7.5], 0.1% SDS, 0.5% Na-Deoxycholate [Na-DOC], 1% Triton X-100) supplemented with 1X cOmplete Protease Inhibitor Cocktail (Roche), 1× PhosSTOP (Roche) and 0.5 mM Dithiothreitol (DTT) (REF). To improve lysis efficiency, a Bioruptor sonication device was used for a 30-s cycle. Lysed cells were centrifuged at maximum speed for 20 min at 4 °C and we collected supernatant. Protein concentration was determined by Pierce BCA Protein Assay kit (Thermo Fisher), according to the manufacturer's instructions.

Equal amounts of quantified proteins were loaded onto denaturing SDS-PAGE gels, and blotted onto nitrocellulose membranes (Bio-Rad) following standard conditions. Membranes were blocked with 5% dry-milk or 3% BSA in PBST Buffer (0.1% Tween-20 in 1× PBS) for 1 h at room temperature. Subsequently, membranes were probed first with primary antibody (Dataset EV4) overnight at 4 °C. Next day, membranes were washed in PBST Buffer and incubated for 1 h at room temperature with HRP-conjugated secondary antibodies. Chemiluminescence detection of the proteins was performed using Western Lightning ECL-Plus (Perkin Elmer) and an Odyssey CLx device (LI-COR). The relative protein levels were determined based on the intensity of the western blot bands, quantified with Image Studio Lite software. Normalisation was performed based on loading reference proteins, such as GAPDH or α-tubulin. Statistical differences between western blot bands intensities were calculated by two-tailed paired $t$ test.

## FACS gamma-H2AX experiments

The following protocol, with a few modifications, was generously provided by Dr. Mitxelena. After appropriate siRNA and treatment incubations, $3 \times 10^5$ A549 cells were collected in PBS and fixed by adding 1 mL of ice-cold 70% ethanol dropwise into FACS tubes. The cells could be stored at 4 °C for several weeks until further use. The cells were pelleted and washed twice with PBST. Next, the cells were incubated with the primary antibody for γH2AX (Dataset EV4) at room temperature for 2 h. After two washes with PBST, AF488-conjugated donkey anti-rabbit secondary antibody (Dataset EV4) was added and incubated for 1 h at room temperature. Following another round of washing, the cells were incubated at 37 °C for 30 min with PI Buffer (38 mM Na-Citrate, 150 Propidium Iodide, 0.01% Triton X-100, 5 µg RNAse A).

For sample analysis, a CytoFLEX LX cytometer (Beckman Coulter) was used. The Propidium Iodide signal was detected through the Y585-PE-A channel to distinguish between live and dead cells, while Alexa-Fluor 488 was detected through the B525-FITC-A channel. The CytExpert software was employed for gating, population distribution, and determination of median intensity for analysis purposes.

## Chromatin immunoprecipitation (ChIP) and data analysis

In all, $30 \times 10^6$ A549 per condition were crosslinked with 1% formaldehyde in 1× PBS for 10 min in agitation. Crosslinking was quenched with final concentration of 125 mM Glycine (Bio-Rad). Cells were collected and incubated in 7 mL of Cell Lysis Buffer (5 mM Tris [pH 8], 85 mM KCl, 0.5% IGEPAL) for 10 min at 4 °C in rotation. Nuclear samples was collected by centrifugation for 10 min at $3000 \times g$ and 4 °C. Pellet was washed in Cell Lysis Buffer without IGEPAL (5 mM Tris [pH 8], 85 mM KCl) and pelleted again. Pellet was resuspended in RIPA Buffer (150 mM NaCl, 50 mM Tris-HCl [pH 7.5], 0.1% SDS, 0.5% Na-Deoxycholate [Na-DOC], 1% Triton X-100) supplemented with 0.1 mM DTT, 1× cOmplete Protease Inhibitor Cocktail (Roche) and 20 U/mL RNAsin Ribonuclease Inhibitors (Promega) for 10 min in ice. Lysate was dounced and chromatin was sheared by sonication in a Bioruptor device for 45 cycles (30"ON-30"OFF). We cleared nuclear extract by centrifugating 30 min at max speed and incubation with 50 µL Protein A/G Dynabeads for 1 h at 4 °C. Nuclear extract was incubated with primary antibodies (Dataset EV4) overnight at 4 °C in rotation. The following day, 50 µL Protein A/G Dynabeads where washed and mixed to antibody-hybridised nuclear extract for 6 h at 4 °C. Beads were washed following the next order: 1 wash of Low-Salt Buffer (150 mM NaCl, 20 mM Tris [pH 8], 2 mM EDTA, 0.1% SDS, 1% Triton x-100), 1 wash of High-Buffer (500 mM NaCl, 20 mM Tris [pH 8], 2 mM EDTA, 0.1% SDS, 1% Triton x-100), 1 wash of LiCl Buffer (0.25 M LiCl, 1% IGEPAL, 1% Na-Deoxycholate [Na-DOC], 1 mM EDTA, 10 mM Tris [pH 8]) and 2 washes with TE Buffer (10 mM Tris, 1 mM EDTA [pH 8]). Beads are resuspended in 250 µL of Elution Buffer (25 mM Tris [pH 7.5], 5 mM EDTA, 0.5% SDS) supplemented with 10 µL of 5 M NaCl, 10 µL 1 M Tris pH 7, 5 uL 0.5 M EDTA and 20 µg Proteinase K for 4 h at 65 °C to promote proper decrosslinking of chromatin from immunoprecipitated proteins. DNA was isolated with Phenol:Chloroform:isoamyl (25:24:1) alcohol (Sigma-Aldrich) isolation and quantified by Qubit4 fluorometer (Invitrogen).

ChIP library preparation was performed following a custom standard protocol with specific primers (Dataset EV4).

In case of YTHDC1 ChIP-seq, reads from three different replicates were aligned to hg19 genome using the BOWTIE2

algorithm (v. 2.3.4.2) (Langmead and Salzberg, 2012). Bigwig files for IGV browser images were generated with deepTools (v. 3.2.0) bamCoverage (Ramírez et al, 2016), including a normalisation by CPM. The aligned reads coming from the three replicates were pooled together and used as input for MACS2 (Zhang et al, 2008) peak-calling software (parameters --broad --broad-cutoff 0.25 -q 0.1).

Pausing index was calculated for the Gencode v19 genes, as the ratio of the RPKMs in 1 kb windows around the TSS divided by the RPKMs inside the gene body, from TSS + 1 kb to TTS-1kb. Genes shorter than 4 kb, or containing the promoter of an internal transcript, were excluded from this analysis. The pausing index ratio was defined as the log2 ratio between the pausing index in siYTHDC1 and siSCR conditions. This pausing index ratio measuring was further corrected by generating a linear model, substracting the effect of siSCR RPKMs quantile division on pausing index ratio.

GSEA analysis was done as described in Subramanian et al, 2005, using the clusterProfiler Bioconductor library.

For RNAPII pausing index Ratio of NELF factor, RNAPII and NELF ChIP-seq public data were extracted from GSE182862 (Wu et al, 2022), and aligned to mm9 genome. We processed data as described above.

## Chromatin-associated RNA crosslinked immunoprecipitation (CLIP) and data analysis

In total, $30 \times 10^6$ A549 per condition were crosslinked with 1% formaldehyde in 1× PBS for 10 min in agitation. Crosslinking was quenched with final concentration of 125 mM Glycine (Bio-Rad). Cells were fractionated as described by Neugebauer's lab (Brugiolo et al, 2017). After obtaining chromatin extracts, pellet was sheared by sonication in a Bioruptor device for 5 cycles (30"ON-30"OFF). We cleared nuclear extract by centrifuging 30 min at max speed and incubation with 50 µL Protein A/G Dynabeads for 1 h at 4 °C. Chromatin extract was incubated with primary antibodies (Dataset EV4) overnight at 4 °C in rotation. The following day, 50 µL Protein A/G Dynabeads where washed and mixed to antibody-hybridised nuclear extract for 6 h at 4 °C. Beads were washed following the next order: 1 wash of Low-Salt Buffer (150 mM NaCl, 20 mM Tris [pH 8], 2 mM EDTA, 0.1% SDS, 1% Triton x-100), 1 wash of High-Buffer (500 mM NaCl, 20 mM Tris [pH 8], 2 mM EDTA, 0.1% SDS, 1% Triton x-100), 1 wash of LiCl Buffer (0.25 M LiCl, 1% IGEPAL, 1% Na-Deoxycholate [Na-DOC], 1 mM EDTA, 10 mM Tris [pH 8]) and 2 washes with TE Buffer (10 mM Tris, 1 mM EDTA [pH 8]). Beads are resuspended in 250 µL of Elution Buffer (25 mM Tris [pH 7.5], 5 mM EDTA, 0.5% SDS) supplemented with 10 µL of 5 M NaCl, 10 µL 1 M Tris pH 7, 5 uL 0.5 M EDTA and 20 µg Proteinase K for 4 h at 65 °C to promote proper decrosslinking of chromatin from immunoprecipitated proteins. RNA was isolated with TRI Reagent (Sigma-Aldrich) isolation and resuspended in 50 µL of water. Samples were treated with TURBO DNAse I (Invitrogen) for 45 min at 37 °C. After that time, a second RNA extraction with TRI Reagent was performed to removed TURBO DNAse I and final quantification by Qubit4 fluorometer (Invitrogen).

RNA was used for library preparation with SMARTer® Stranded Total RNA-Seq Kit v3 (Takara Bio) with rRNA removal and

sequenced on Illumina Novaseq 6000 (50 bp paired-end mode, 40 $\times 10^6$ reads/sample).

After filtering out reads with a quality lower than 20 in the first 30 nt, exact duplicates were collapsed, and UMIs were stripped. The reads were aligned to hg19 genome using bwa mem with standard parameters, PCR duplicates were collapsed, and the final aligned reads coming from two replicates were pooled. For the peak-calling, the software Piranha was used (Uren et al, 2012), with a binning of 50 bp (-b 50).

To show Total RNA YTHDC1 iCLIP tracks we used public from GSE78030 (Patil et al, 2016).

## RNA-sequencing and data analysis

For RNA-seq of YTHDC1-depleted- and non-depleted-A549 cells, total RNA was isolated and DNAse I-treated using Maxwell® RSC simplyRNA Tissue kit (Promega), following manufacturer's instructions. In all, 1 µg of RNA was used for library preparation with Truseq Stranded Total RNA-seq kit (Illumina) with rRNA removal and sequenced on Illumina Novaseq 6000 (150 bp paired-end mode, $30 \times 10^6$ reads/sample).

Reads were aligned with STAR (v. 2.7.0) (Dobin et al, 2013) using standard parameters against hg19 genomic version. Bigwig files for IGV browser images were generated with deepTools (v. 3.2.0) (Ramírez et al, 2016) bamCoverage, including a normalisation by CPM. The counts of reads per gene was done with HTseq (Anders et al, 2015), using Gencode v19 annotation, and differential expression analysis was carried out with the Bioconductor library DESeq2 (v. 1.34.0) (Love et al, 2014) (an extra variable was included in the DESeq2 generalised linear model to account for two different batches of sequencing). Genes with adjusted $P$ value < 0.001 and a log2FC > 0.5 were considered differentially expressed.

For pathway enrichment analysis, PathFindR pipeline (v. 2.1.0) (Ulgen et al, 2019) was used, applying the default thresholds.

For the general differential splicing analysis, the vast-tools pipeline (Tapial et al, 2017) for more than 5 replicates per condition was used: after the vast-tools alignment of the reads and the quantification of the PSIs, splicing events were selected as differential if they had a dPSI>8 and a Mann–Whitney test $P$ value < 0.05.

Retained introns were found using IRFinder (v. 1.3.1) (Middleton et al, 2017) with standard parameters. The retention was considered statistically significant for those cases where adjusted $P$ value < 0.1.

## ChrMeRIP-seq data analysis

ChrMeRIP reads were aligned to hg19 genome using STAR (v. 2.7.0) (Dobin et al, 2013) with standard parameters. Bigwig files for IGV browser images were generated with deepTools (v. 3.2.0) bamCoverage (Ramírez et al, 2016), including a normalisation by CPM. Peak-calling analysis was performed with exomePeak2.

The classification of Gencode v19 introns in five proximity groups depending on the relative position to a meRIP peak was performed with homemade shell and R scripts.

Public data were used to performed this analysis: GSE144404 (Xu et al, 2022).

## Data availability

A reporting summary for this article is provided. RNA-seq, ChIP-seq and CLIP data reported in this study have been deposited in Gene Expression Omnibus (GEO) repository under the accession number: GSE239333.

The source data of this paper are collected in the following database record: biostudies:S-SCDT-10_1038-S44318-024-00153-x.

## Peer review information

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

## Acknowledgements

This work was supported by grants to MH: PID2020-113683GB-I00 and ED2022-134792-T funded by MICIU/AEI /10.13039/501100011033, European Research Council Consolidator grant 771425 and Worldwide Cancer Research grant 20-0204. DE was supported by Ph.D. fellowship from Ministerio de Ciencia e Innovación REF: PRE2018-085992. We thank Lani F Wu's laboratory for donating the A549-p21-mVenus cell line and Catherine Potenski for her feedback on the manuscript.

## Author contributions

**Daniel Elvira-Blázquez**: Conceptualisation; Formal analysis; Investigation; Methodology; Writing—original draft; Writing—review and editing. **José Miguel Fernández-Justel**: Data curation; Formal analysis; Investigation; Methodology. **Aida Arcas**: Data curation; Formal analysis; Methodology. **Luisa Statello**: Investigation; Methodology. **Enrique Goñi**: Data curation; Formal analysis. **Jovanna González**: Investigation; Methodology. **Benedetta Ricci**: Investigation. **Sara Zaccara**: Investigation. **Ivan Raimondi**: Conceptualisation; Supervision; Investigation; Methodology; Writing—original draft; Project administration; Writing—review and editing. **Maite Huarte**: Conceptualisation; Supervision; Funding acquisition; Investigation; Methodology; Writing—original draft; Project administration; Writing—review and editing.

Source data underlying figure panels in this paper may have individual authorship assigned. Where available, figure panel/source data authorship is listed in the following database record: biostudies:S-SCDT-10_1038-S44318-024-00153-x.

## Disclosure and competing interests statement

The authors declare no competing interests.

# Expanded View Figures

**Figure EV1.  CRISPR screening enables unbiased identification of RNA modifiers involved in p53 response.** ▶

(A) Representative Immunoblot of p53, with GAPDH as loading control. Cells were treated with Nutlin-3a or untreated as negative control. (B) Nutlin-3a dose response treatment. P21-Reporter cells were treated with increasing doses of Nutlin-3a or untreated as negative control. After treatment, reporter gene activation was measured by flow cytometry. The percentage of cells showing reporter gene activation was calculated and presented in the plot of the signal distribution. (C) Representative immunoblot of spCas9, with a-tubulin as loading control. Cells were transduced with a lentivirus carrying spCas9 or an empty vector as negative control. (D) Flow cytometry quantification of P21-Reporter gene activation and CRISPR library infection for the two replicates used in the screening. (E) Scatter plot of $\log_2$ fold change for sgRNA enrichment in p53-enhanced and p53-attenuated populations. Each individual sgRNA for the three top candidates for each population is labelled and highlighted in the plot. Non-targeting sgRNAs used as negative controls in black. (F) Scatter plot of $\log_2$ fold change for gene enrichment in p53-enhanced and p53-attenuated populations. Previously reported negative or positive regulators of p53 activity are labelled in the plot in red and green, respectively. (G) Functional annotation of the top 50 candidates for p53-enhanced and p53-attenuated populations based on their molecular substrate. (H) RNA-level quantification of mature mRNA by RT-qPCR for *YTHDC1* and *ASH2L*. Cells were transfected with two independent siRNA against *YTHDC1* (DC1-1 and -2), two independent siRNA against *ASH2L* (ASH2L-1 and -2) or Scramble (SCR) as negative control. Data information: All data are shown are representative of at least three independent experiments, except for (C). Data are presented as mean ± s.d. ***$P \le 0.001$, ****$P \le 0.0001$, paired two-tailed Student's *t* test was performed in (H).

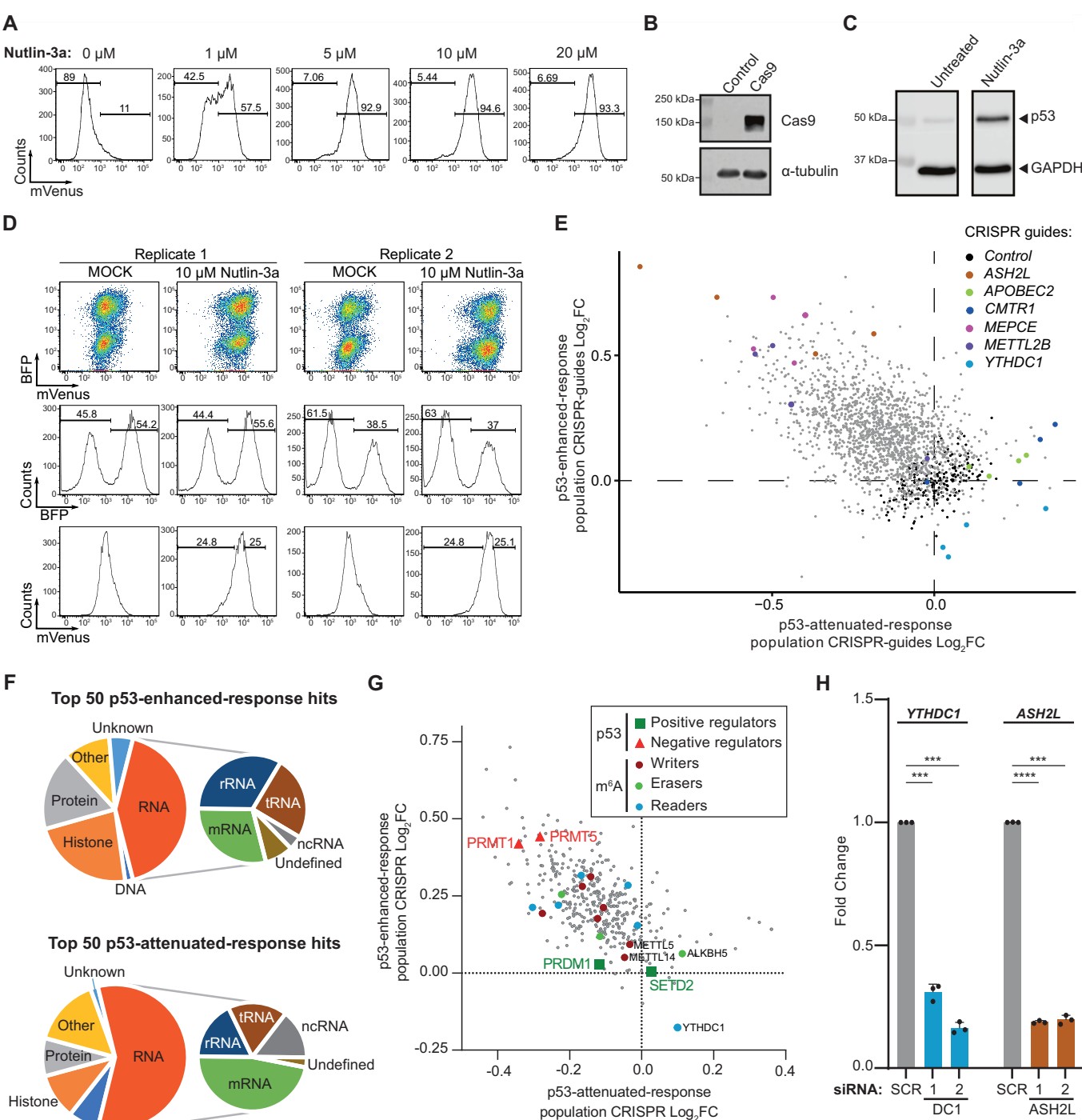

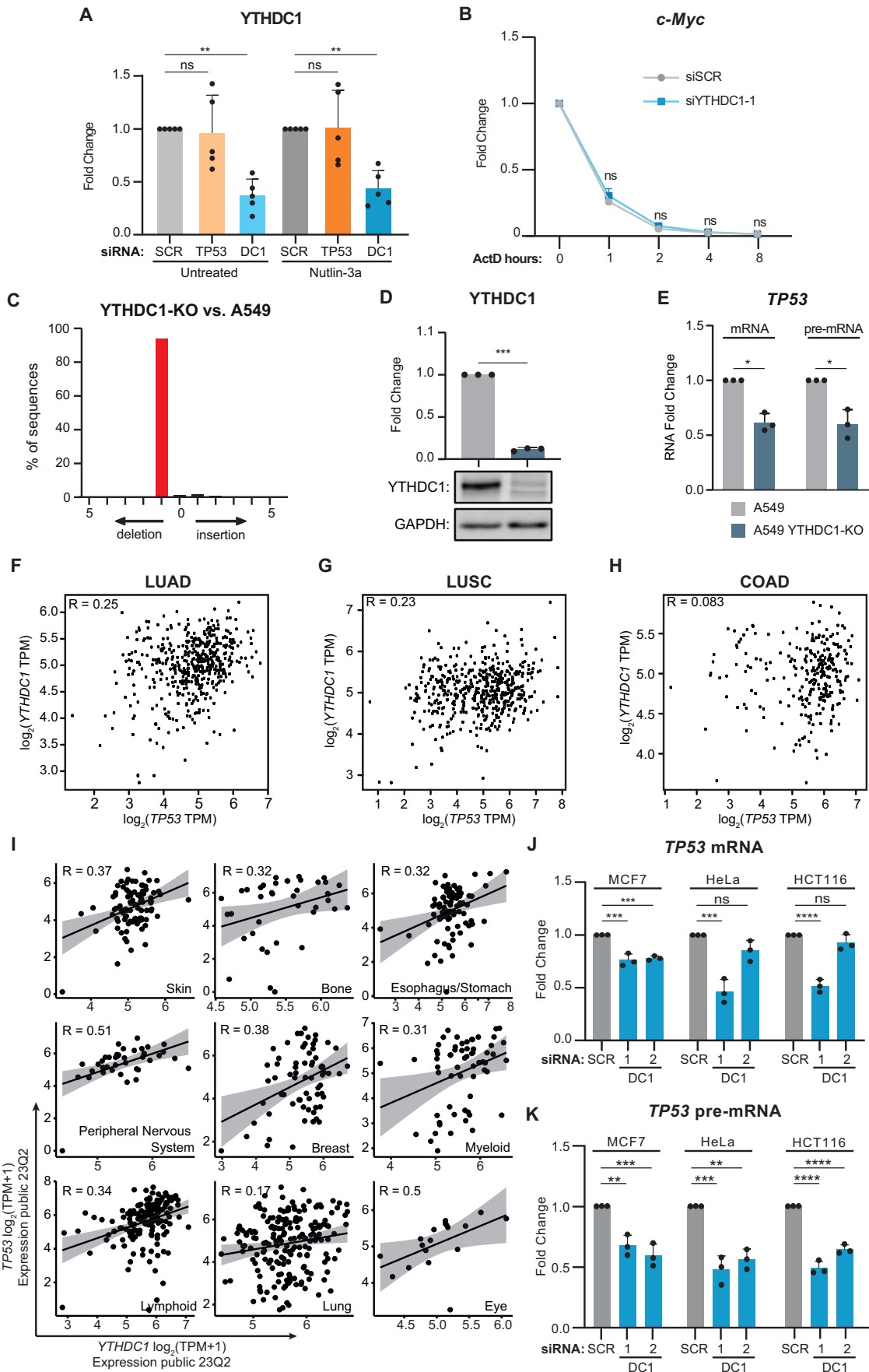

**Figure EV2.   YTHDC1 directly regulated TP53 transcription.**

(A) YTHDC1 protein quantification. Cells were transfected with siRNA against *TP53*, *YTHDC1* or Scramble (SCR) as negative control. After silencing cells were treated to Nutlin-3a, DMSO was used for the untreated condition as negative control. The bar plot shows protein quantification relative to GAPDH level of $n = 5$ biologically independent experiments. (B) *c-Myc* RNA stability assay. Cells were transfected with siRNA, YTHDC1, or Scramble (SCR) as negative control. After silencing cells were treated with Actinomycin D to stop the transcription. Cells were collected at different time points indicated on the X axis to assess *c-Myc* RNA level by quantitative RT-qPCR. In vitro transcribed *Luciferase* RNA was used as spike-in to normalise the signal. (C) Comprehensive profile of insertions and deletions (indels) in YTHDC1-KO A549 clone compared to a control A549 cells. (D) Representative picture of a Western blot of normal A549 cell line and the YTHDC1-KO A549 clone, together with relative quantification of 3 independent protein extractions from the same clone. (E) RNA-level quantification of mature mRNA and pre-mRNA by RT-qPCR for *TP53* in A549 cells (grey) and YTHDC1-KO (dark blue). (F–H) Correlation plot showing *YTHDC1* and *TP53* expression (as log2 TPM + 1 pseudocounts) in LUAD samples ($n = 505$) (B), in LUSC samples ($n = 479$) (C), and COAD samples ($n = 427$) (D) from TCGA database (gdc-portal.nci.nih.gov). Correlation *P* value is calculated using a t-distribution. (I) Correlation plot showing *YTHDC1* and *TP53* expression (as log2 TPM + 1 pseudocounts) in 824 cell lines grouped based on the tissue of origin from DepMap database (Tsherniak et al, 2017, https://depmap.org/portal). Correlation *P* value is calculated using a t-distribution. (J) RNA-level quantification of mature mRNA by RT-qPCR for *TP53* in multiple cell lines. MCF7, HeLa and HCT116 cell lines were transfected with two independent siRNA against *YTHDC1* (DC1-1 and -2), or Scramble (SCR) as negative control. (K) RNA-level quantification of pre-mRNA by RT-qPCR for *TP53* in multiple cell lines. MCF7, HeLa and HCT116 cell lines were transfected with two independent siRNA against *YTHDC1* (DC1-1 and -2), or Scramble (SCR) as negative control. Data information: All data are shown are representative of at least three independent experiments. Data are presented as mean ± s.d. ns, not significant $P > 0.05$, *$P \leq 0.05$, **$P \leq 0.01$, ***$P \leq 0.001$, ****$P \leq 0.0001$, paired two-tailed Student's *t* test was performed in (A, B, D, E, J, K).

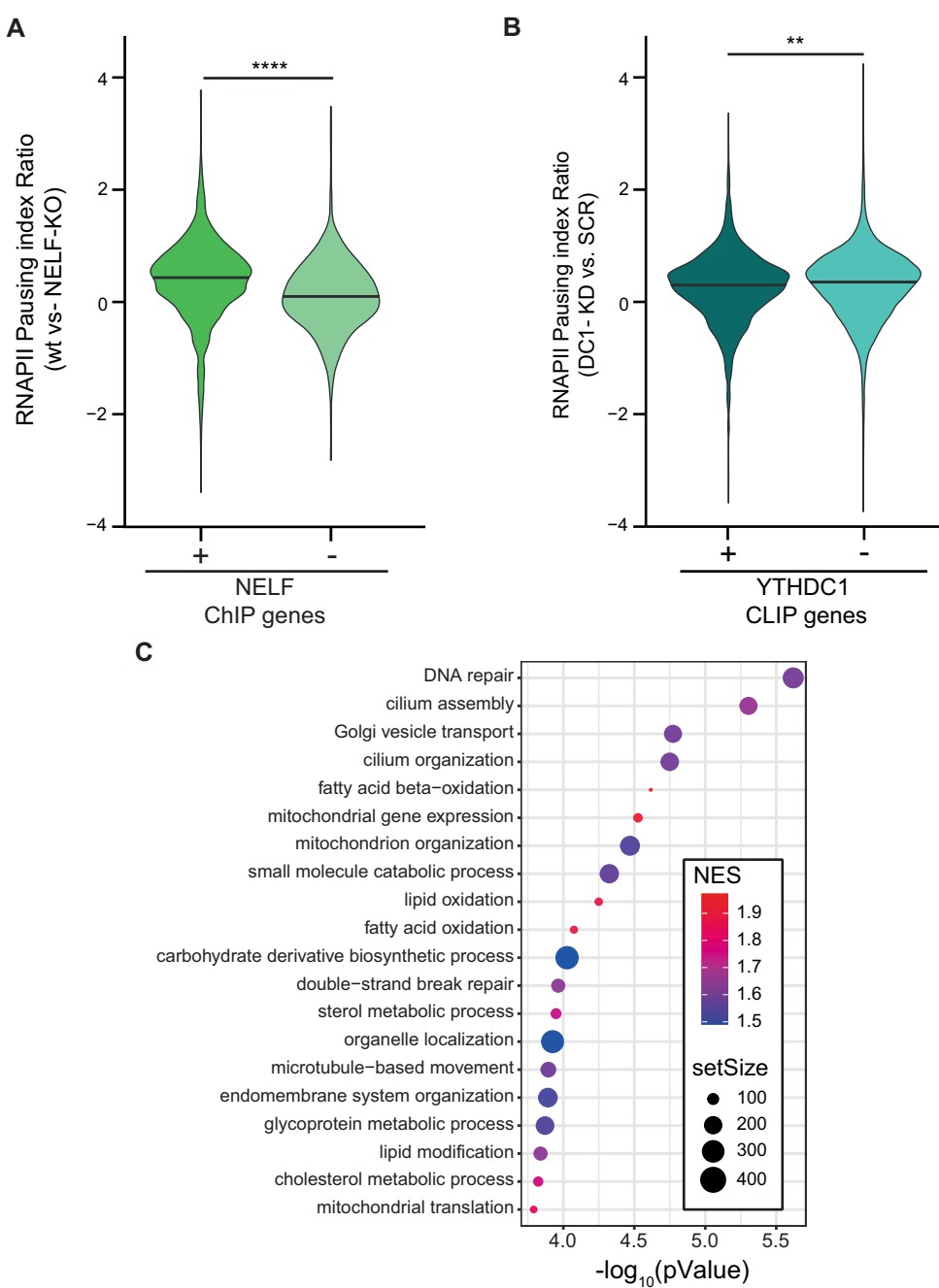

**Figure EV3.  RNAPII pausing upon YTHDC1 depletion mainly affects DNA repair.**

(A) Violin plots of the distribution of expression corrected RNAPII Pausing Index ratio calculated for public dataset NELF ChIP-seq experiment. All the actively transcribed genes were divided into two different groups based on the presence or absence of NELF peaks in the TSS (green and light green blue, respectively). Pausing index was calculated as log2 RNAPII promoter density/RNAPII gene body density for both wild-type and NELK-KO cells. Data are presented as distribution of Pausing Index ratio for wild-type versus KO. (B) Violin plots of the distribution of expression corrected RNAPII Pausing Index ratio calculated for YTHDC1-CLIP-seq dataset. All the actively transcribed genes were divided into two different groups based on the presence or absence of YTHDC1 along the transcript (dark and light blue bondi). Pausing index was calculated as log2 RNAPII promoter density/RNAPII gene body density for both cells transfected with scramble (SCR) siRNA and YTHDC1 siRNA. Data are presented as distribution of Pausing Index ratio for YTHDC1 knockdown versus SCR. (C) Gene Set Enrichment Analysis (GSEA) of genes with an increased RNAPII Pausing Index upon YTHDC1 depletion. Data information: All data are shown are representative of at least three independent experiments. Data are presented as mean ± s.d. **$P \leq 0.001$, ****$P \leq 0.00001$, paired two-tailed Student's $t$ test was performed in (A, B). GSEA has been performed following Subramanian algorithm (Subramanian et al, 2005).

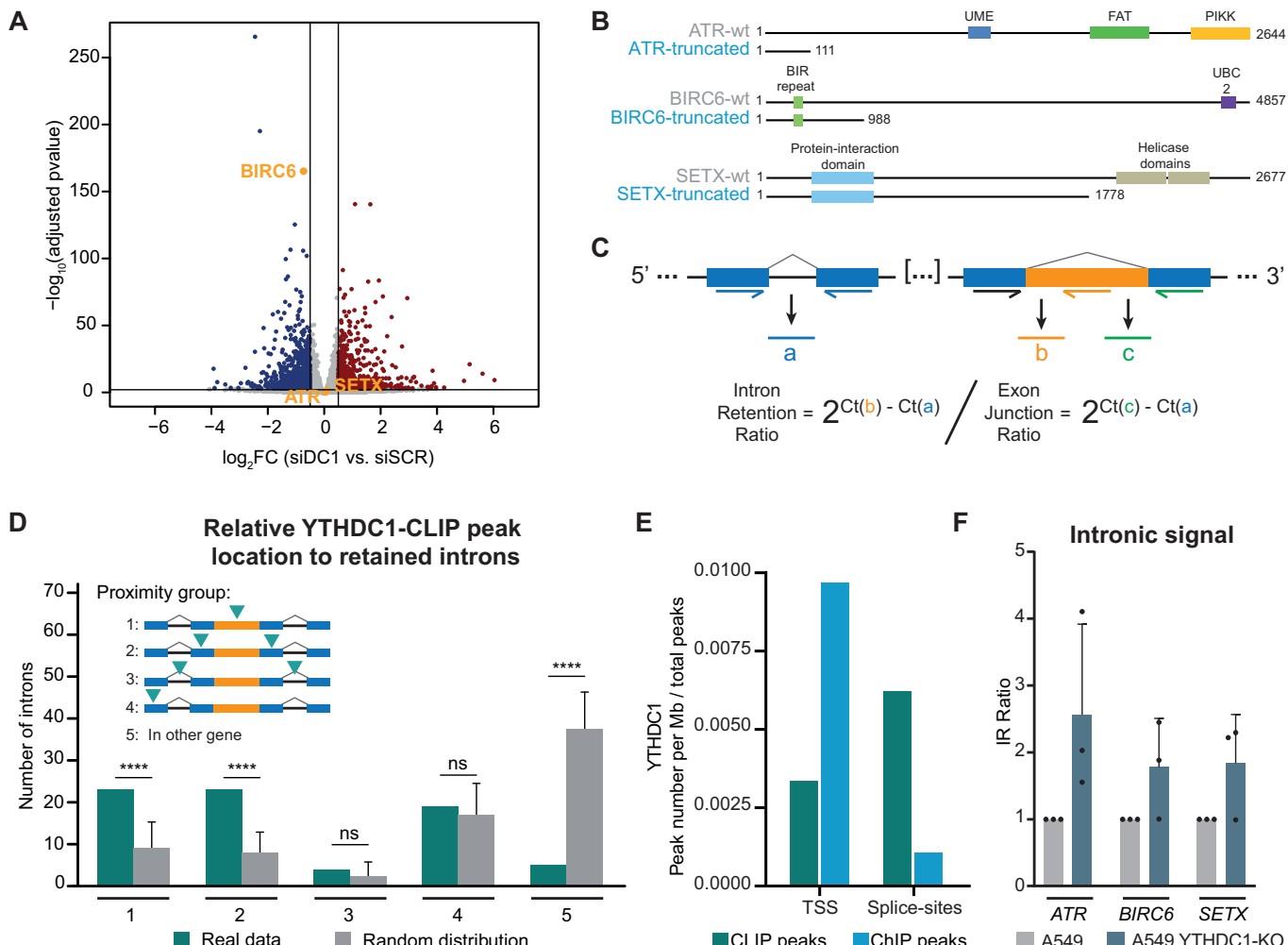

**Figure EV4.  YTHDC1 directly promotes correct splicing.**

(A) Volcano plot showing differentially expressed genes (DEGs) identified in YTHDC1-silenced cells versus scramble (SCR) cells. Genes with $\log_2$(FC) > 0.5 and Adjusted *P* value < 0.001, resulting from the DESeq2 DGA, are considered DEGs. Significantly upregulated, downregulated or not changed genes in the YTHDC1 knocked-down cells are labelled in blue, red or grey colour, respectively. (B) Schematic representation of truncated-version proteins for ATR, BIRC6 and SETX in negative control condition (grey) and YTHDC1-depleted condition (cyan), due to the emergence of premature stop-codons. (C) Schematic representation of primer design for quantitative PCR to detect spliced and unspliced isoforms of *ATR, BIRC6* and *SETX*. An upstream exon-junction region we selected to normalise signal. (D) Plot showing the number of differentially retained introns (blue bondi bars) and 1000 random selections of GENCODE v19 introns (grey bars), classified in 5 different groups depending on the proximity to a YTHDC1-CLIP peak (group 1: peak inside the intron; group 2: peak in adjacent exons; group 3: peak in adjacent introns; group 4: peak anywhere inside the gene; group 5: no peak inside the gene). Error bars represent twice the standard deviation. (E) Relative distribution of YTHDC1-CLIP (blue bondi) and YTHDC1 ChIP (cyan) peaks over TSS (defined from TSS to 500 bp downstream) and splicing sites (5' and 3' splicing sites) along genome. (F) RNA-level quantification of intronic retention ratio by RT-qPCR for *ATR, BIRC6* and *SETX* mRNA in A549 cells (grey) and YTHDC1-KO (dark blue). Data information: DESeq2 Wald test *P* value was calculated in (A). Empirical *P* value for (D) was obtained by randomising 100 times the selected introns to compared. Data are presented as mean ± s.d. ns, not significant *P* > 0.05, ****$P \leq 0.0001$, paired two-tailed Student's *t* test was performed in (D, F).

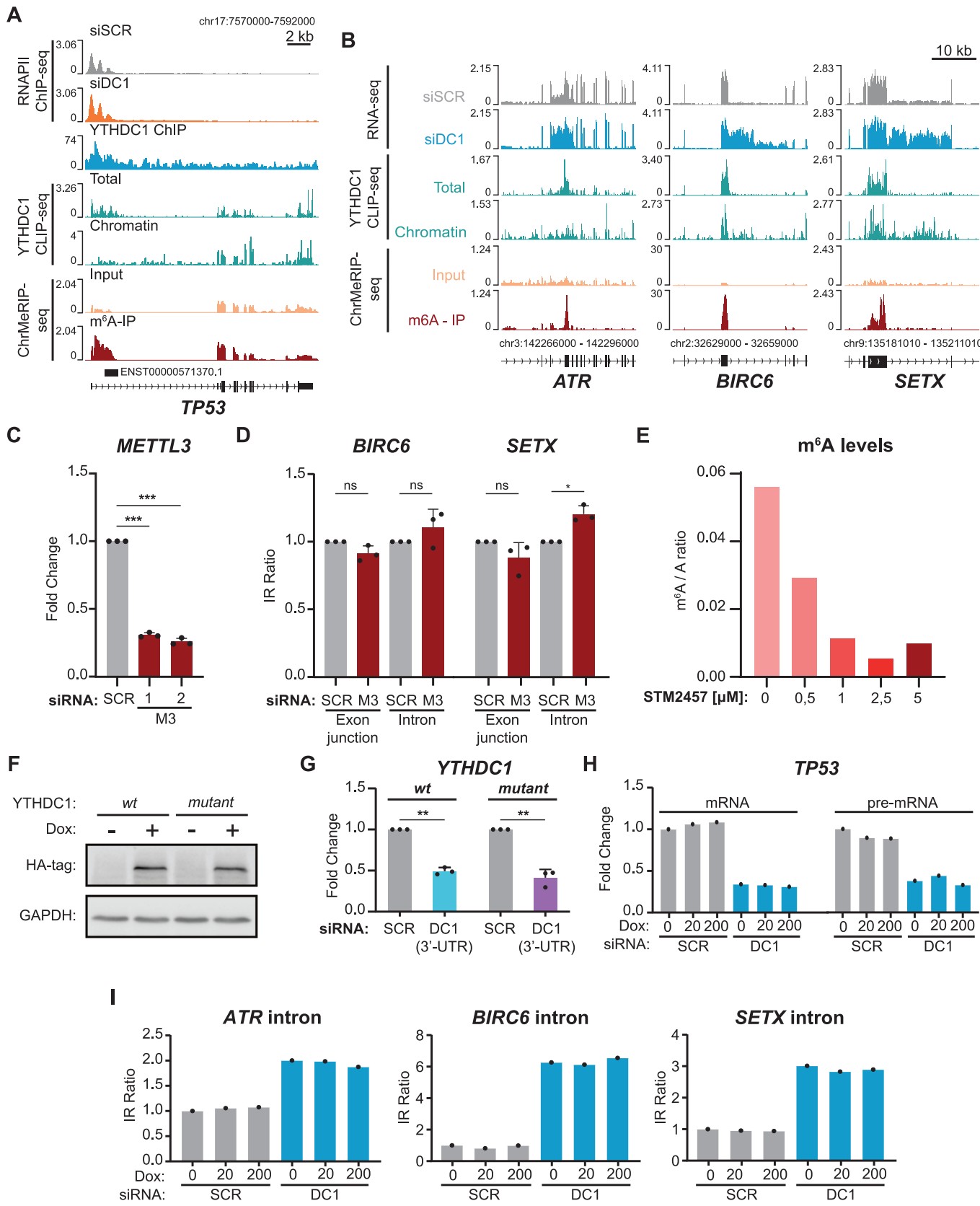

◄ **Figure EV5. YTHDC1 modulates *TP53* transcription independently of m⁶A and regulates the correct splicing of *ATR, BIRC6* and *SETX* intron in a m⁶A-dependent manner.**

(A) Genome browser tracks for RNAPII ChIP-seq of cell transfected with scramble siRNA (light grey) or siRNA against YTHDC1 (orange), YTHDC1 ChIP (cyan) YTHDC1-CLIP (blue bondi) and ChrMeRIP input (yellow) and m⁶A specific peaks (dark red), showing reads coverage over *TP53* locus. Sequencing data were normalised as Fragments Per Kilobase of transcript per Million mapped reads (FPKM). (B) Genome browser tracks for total RNA-seq of cell transfected with scramble siRNA (light grey) or siRNA against YTHDC1 (cyan), YTHDC1-CLIP (blue bondi) and ChrMeRIP input (yellow) and m⁶A specific peaks (dark red), showing reads coverage over *ATR, BIRC6* and *SETX* retained introns. Sequencing data were normalised as Fragments Per Kilobase of transcript per Million mapped reads (FPKM). (C) RNA-level quantification of mature mRNA by RT-qPCR for *METTL3*. Cells were transfected with two independent siRNA against *METTL3* (M3-1 and -2) or Scramble (SCR) as negative control. (D) RNA-level quantification of spliced or unspliced for *BIRC6* and *SETX* mRNA by RT-qPCR. Cells were transfected with siRNA against *METTL3* (dark red), or Scramble (light grey) as negative control. (E) TLC quantification of the ratio between total amount of Adenosine nucleotides (A) and N6-methyladenosine nucleotides (m⁶A) of one representative replicate. (F) Representative western blot of HA-YTHDC1 *wt* and *mutant* versions upon doxycycline treatment with 20 and 200 ng/mL, respectively, from one of the experiments performed in Fig. 5G, H. (G) RNA-level quantification of endogenous YTHDC1 upon 3′-UTR designed siRNA transfection. (H, I) Parental A549 cells treated with concentrations of Doxycycline applied for YTHDC1 inducible system. Data information: All data are shown are representative of at least three independent experiments. For (C, D, G), data are presented as mean ± s.d. ns, not significant $P > 0.05$, *$P \leq 0.05$, **$P \leq 0.01$, ***$P \leq 0.001$, paired two-tailed Student's *t* test was performed.

