## [Peer Review File · The EMBO Journal]

YTHDC1 m6A-dependent and m6A-independent functions converge to preserve the DNA damage response

Daniel Elvira-Blázquez, José Fernández-Justel, Aida Arcas, Luisa Statello, Enrique Goñi, Jovanna González, Benedetta Ricci, Sara Zaccara, Ivan Raimondi, and Maite Huarte

Corresponding author(s): Maite Huarte (maitehuarte@unav.es), Ivan Raimondi (iraimondi@nygenome.org)

Review Timeline:

Submission Date:	1st Sep 23
Editorial Decision:	7th Nov 23
Revision Received:	24th Mar 24
Editorial Decision:	29th Apr 24
Revision Received:	7th May 24
Accepted:	5th Jun 24

Editor: Cornelius Schneider

Transaction Report:

Dear Dr. Huarte,

It was good meeting you and discussing the manuscript at Heidelberg. I have just realized that I have not yet officially invited the revision.

Therefore I would like to invite you officially now to submit a revised version of the manuscript, addressing the comments of all three reviewers. I should add that it is EMBO Journal policy to allow only a single round of revision, and acceptance of your manuscript will therefore depend on the completeness of your responses in this revised version.

Thank you for the opportunity to consider your work for publication. I look forward to your revision.

Yours sincerely,

Cornelius Schneider, PhD
Editor
The EMBO Journal
c.schneider@embojournal.org

We realize that it is difficult to revise to a specific deadline. In the interest of protecting the conceptual advance provided by the work, we recommend a revision within 3 months (5th Feb 2024). Please discuss the revision progress ahead of this time with the editor if you require more time to complete the revisions.

Referee #1:

In this manuscript, the authors used A549 derived p21-mVenus reporter cell line and performed CRISPR screening with a sgRNA library targeting 407 genes involved in RNA modifications. Several hits were verified using the same reporter cell line and YTHDC1 was chosen for further analysis. The authors proposed that YTHDC1 may directly regulate p53 transcription via modulating the release of paused RNA POL II near the promoters. Based on the role of YTHDC1 in splicing, the authors also examined alternative splicing in YTHDC1 depleted cells and confirmed ATR, BIRC6 and SETX as genes regulated by YTHDC1. Additional data were provided to show modest increase of tail DNA and reduction of pH2AX signal in YTHDC1 depleted cells, which may result from its effect on ATR expression. However, these effects may be due to reduced cell proliferation upon YTHDC1 depletion as shown in Fig. 5E.

Overall, the data presented did not support a specific role of YTHDC1 in p53 and/or DNA damage responses. Given the pleiotropic effects of YTHDC1 in gene regulation, the impact of this study is limited.

Major issues:

1. The screen was conducted only in the presence of Nutlin-3a, but not in cells without Nutlin treatment. Therefore, the genes identified may participate in general transcriptional/post-transcriptional regulation, but not specifically involved in p53-dependent response.
2. Fig 1F showed Pearson correlation scores of 407 genes presented in the sgRNA library. Unfortunately there is no correlation between these scores and the scores presented in Fig. 1C.
3. The data presented in Fig. 2 were not convincing. siRNA targeting YTHDC1 only showed a modest reduction in YTHDC1 protein level. Similarly, Fig. 2E showed that siRNA targeting YTHDC1 did not result in significant downregulation of YTHDC1 expression. The authors need to use YTHDC1 knockout cells and reconstituted cells to further validate their results.
4. The effect on RNAPII pausing shown in Fig. 3 was modest and not specific for p53 gene.

Referee #2:

This is a very interesting manuscript where the authors have identified YTHDC1 as a critical mediator of p53 and its transcriptional response. A major driver for publication is the identification of a completely new regulator of p53, which is very surprising since this is such a highly studied protein. The other major driver is that this study links a very important m6A pathway protein, YTHDC1 with the p53 pathway.

This was a very interesting paper to read. The authors use a very good and high quality CRISPR screening approach. The connection between YTHDC1 and p53 is compelling.

However, there are problems with the mechanisms that need to be addressed. I'm not fully convinced of the mechanistic connections due to some technical problems and other issues with some of the experiments.

1. First, I think most readers will be surprised that YTHDC1 has not come up in prior CRISPR or other screens for regulators of p53 function. Why is that? Perhaps YTHDC1 did come up, but it was never studied or addressed. The author should, in the supplement, list all of the screens that have been performed for p53 regulators, and indicate where YTHDC1 was ranked amongst the regulators. Perhaps, some of these earlier screens did not include YTHDC1. Alternatively, if there are screens that are very similar to the ones performed here, and YTHDC1 did not appear as a among the topics, the net is somewhat concerning. It should at least be mentioned.
2. The authors bring up the very interesting finding that the DepMap shows a strong codependency between p53 and YTHDC1. However, there are some pretty serious problems with this line of reasoning. The authors claim that the mechanism of YTHDC1 is m6A independent. However, the authors may not realize that METTL3 shows very strong codependency with p53 as well.

other components of the m6A writer complex also show high codependency. This seems to argue against the core idea that the authors present which is that the actions of YTHDC1, at least on p53, are independent of m6A.

A major method that the authors used to demonstrate that the m6A deposition is not needed for the actions of YTHDC1 is that they use siRNA to knock down METTL3, and failed to see p53 regulation similar to what they see with YTHDC1 knockdown. The authors may not know this, but METTL3 knocked down often does not deplete m6A levels. This is because METTL3 is generally knocked off the rate limiting for m6A production. Schwartz was the first to show that knockdown of METTL3 by 90% causes almost no change in m6A levels. Indeed, the authors did not measure m6A levels after their METTL3 knockdown to the best of my understanding. Typically one would expect that the authors would knock out YTHDC1 and replace it with a tryptophan mutant, or similar, that cannot bind m6A.

Note that some studies show that YTHDC1 is a major component of speckles :<https://doi.org/10.1016/j.devcel.2021.01.015> - this activity involves its ability to bind m6A and the MALAT1 noncoding RNA, target transcripts do not necessarily need to contain m6A, similar to what the authors find here. I would suggest that they reconsider their proposed mechanism, at least in the context of m6A independent functions on p53.

3. I'm somewhat suspicious of the proposed mechanism of YTHDC1 regulating splicing. The authors should look carefully at the work of Brockdorff (<https://www.ncbi.nlm.nih.gov/pmc/articles/PMC8327914/>) these authors did a very careful analysis of m6A and its role in splicing using acute depletion. This is important because m6A knock down and other m6A inhibition mechanisms can result in secondary and tertiary effects which ultimately affect splicing. However, this work using acute m6A depletion, I believe, did not show any overlap with the proposed alternative splicing mechanisms shown here. I think that the mechanism that they've identified is an indirect effect - m6A probably affects a certain regulator, which ultimately affects other pathways that lead to splicing defects that are ascribed to YTHDC1 in this study. More documentation of the direct effect of YTHDC1, in light of Brockdorff needs to be described, or the authors need to change their story so that they encompass secondary effects of m6A/YTHDC1 depletion.

4. Attardi proposed a mechanism of p53 stabilization via Mett13 which seems to provide an alternative model for m53 regulation - <https://doi.org/10.1016/j.molcel.2022.04.010> I cannot reproduce this study but nevertheless, the authors should address whether this can account for p53 expression in their hands or rebut this study, especially if they find Mett13 dependence for their pathway.

Referee #3:

Through a targeted CRISPR-knockout screen, these authors identified a number of genes involved in p53 regulation. In this list a nuclear m6A-binding protein YTHDC1 appears to be a master regulator of p53 expression. The authors showed two pathways underlying this regulation. YTHDC1 can bind to the transcription start sites of TP53 and other genes involved in DNA damage response, promoting their transcriptional elongation. This protein also recognizes m6A and mediates correct splicing. Depletion of YTHDC1 leads to intron retention and reduced TP53 expression. These results are very interesting, particularly the splicing regulation part, which to my knowledge is novel and a rare example showing connection of intron methylation with splicing and functional outcome. I do have several questions that would need to be addressed.

Main:

1. The authors uncovered two effects of YTHDC1, transcription regulation and splicing regulation. The authors will really need to separate these two effects on selected genes. They showed that YTHDC1 occupies promoter site through ChIP. Can the authors perform CLIP-seq and analyze proportions of YTHDC1 on promoter RNA or nascent RNA versus across splicing junctions. This is critical.

2. A previous work from Yang Shi and coworkers has shown a mechanism of m6A to promote nascent RNA synthesis ([https://www.cell.com/molecular-cell/pdfExtended/S1097-2765\(22\)00111-3](https://www.cell.com/molecular-cell/pdfExtended/S1097-2765(22)00111-3)). It looks to me this pathway might be the main pathway functioning here instead of splicing regulation. Again, the authors need to carefully analyze both pathways.

3. "to investigate whether m6A potentially mediates the differences in RNAPII pausing that we observed in YTHDC1-knockdown cells, we divided the transcriptome into m6A genes and non-m6A genes,... suggesting that it is the presence of YTHDC1, rather than the presence of m6A, is the most critical for the control of elongation dynamics in our system" makes no sense. The authors are studying nascent RNA and introns, yet they were trying to correlate m6A on mature mRNA with transcription and splicing changes. They really need to map m6A (MeRIP) on nascent RNA or chromatin RNA. This is key to the current work.

4. The novelty of the work lies on splicing regulation. The authors can deliver dCas13-FTO or ALKBH5 to intron m6A sites or nascent RNA m6A sites and monitor productive mRNA synthesis.

Again, effects of splicing changes versus transcription are critical. A few clear examples that splicing but not transcription affects outcome of gene expression and DNA damage response are required.

Minor:

Fig. 5F not clear.

"However, several YTH family proteins have been reported to control m6 A-marked RNA, modifying their stability, resulting dramatic changes to RNA half-life (Zaccara & Jaffrey, 2020; Z. Zhang et al., 2020)"

Wrong citation. First paper was published in 2013 by Wang et al. Important to cite original literature. Please go through the manuscript and cite original literature.

Point-by-point response to referees

Elvira-Blázquez et al.

Referee #1 (Report for Author)

In this manuscript, the authors used A549 derived p21-mVenus reporter cell line and performed CRISPR screening with a sgRNA library targeting 407 genes involved in RNA modifications. Several hits were verified using the same reporter cell line and YTHDC1 was chosen for further analysis. The authors proposed that YTHDC1 may directly regulate p53 transcription via modulating the release of paused RNA POL II near the promoters. Based on the role of YTHDC1 in splicing, the authors also examined alternative splicing in YTHDC1 depleted cells and confirmed ATR, BIRC6 and SETX as genes regulated by YTHDC1. Additional data were provided to show modest increase of tail DNA and reduction of pH2AX signal in YTHDC1 depleted cells, which may result from its effect on ATR expression. However, these effects may be due to reduced cell proliferation upon YTHDC1 depletion as shown in Fig. 5E.

Overall, the data presented did not support a specific role of YTHDC1 in p53 and/or DNA damage responses. Given the pleiotropic effects of YTHDC1 in gene regulation, the impact of this study is limited.

We thank the referee for the suggestions and comments. We took into consideration every point as explained below.

Major issues:

1. The screen was conducted only in the presence of Nutlin-3a, but not in cells without Nutlin treatment. Therefore, the genes identified may participate in general transcriptional/post-transcriptional regulation, but not specifically involved in p53-dependent response.

We acknowledge the reviewer's concern and apologise for the lack of clarity. Our main goal was to identify RNA metabolism-related novel regulators of the p53 response. To address this, we performed a fluorescence-based screen with a p53 reporter gene (p21) as phenotypic readout and Nutlin-3a treatment to activate p53, by doing so we could easily identify cells with a high or low response. To exclude unspecific effects, we performed the same screening in untreated conditions (MOCK), which is used to evaluate the representation of the CRISPR-sgRNA library in the overall population in basal conditions. As expected, the untreated condition presents very low p21-reporter signal, close to background levels (Figure EV1D, low panels). We now included a paragraph in the results section to better explain our experimental design, highlighting the use of the untreated condition as a control.

Moreover, after identifying YTHDC1 as a hit of the screen we extensively validated its direct implication in p53 response by performing numerous experiments and analyses of the p53 pathway in different cell systems and tumour and healthy patients with a p53-wt background and p53 expression in different tissues (Figure 1F, Figure 2F and Figure EV2F-I). These corroborate that YTHDC1 is a positive regulator of p53. We are aware that YTHDC1 could be involved in many cellular pathways, for this reason we carefully planned our experiments to specifically evaluate the role of this protein in p53 response, including controls with the aim to avoid confounding effects derived from the global alteration of the RNA metabolism.

Although reviewer concerns are more than legitimate, we believe we did our best to isolate p53-specific phenotypic changes. Taking into consideration the technological limitations that still exist in the m6A field, our data clearly demonstrates that YTHDC1 acts at multiple levels to fine tuning p53 response and DNA damage.

2. Fig 1F showed Pearson correlation scores of 407 genes presented in the sgRNA library. Unfortunately there is no correlation between these scores and the scores presented in Fig. 1C.

We apologise again for the lack of clarity. Figure 1F represents the correlation between the expression of the genes of interest and the alteration score of the p53 pathway. The score is calculated based on the expression of 264 downstream genes of the p53 pathway across hundreds of tumour samples compared to other hundreds of healthy samples with *wild-type TP53* background. This p53 alteration score was related with the expression of each candidate included in the CRISPR Screening. Positive and negative values of p53 alteration score means that higher or lower expression of CRISPR candidate, respectively, the more altered is p53 pathway. Given the nature of the samples, it would be unrealistic to expect a perfect correlation between the screen results (generated in one single cell line) and a score value that integrates expression of hundreds of genes in hundreds of patient-derived tumours with inherent heterogeneity. In Figure R1, we showed two individual examples of a candidate with positive (*PRMT1*) and negative (*YTHDC1*) p53 pathway alteration score. Each dot represents a single LUAD patient compared to the normalised expression of healthy patients.

Remarkably, despite this heterogeneity, we believe that the fact that the top and bottom hits obtained in the screening cluster together and follow similar trends in the Pearson correlation value, including *YTHDC1* and *ASH2L*, supports the idea that they are functionally related to the p53 pathway in the direction predicted by the screening. We agree that this analysis alone is not demonstrative, but provides evidence in support of the translatability of the screening results, strengthening the biological relevance of the hits.

Figure R1. p53 pathway alteration score between p53-downstream-effector misregulation (p53 pathway) compared to the expression of the *PRMT1* (left panel) and *YTHDC1* (right panel) in LUAD patients from TCGA with p53 wild-type genotype versus healthy patients.

To make this point clearer in the manuscript, we have expanded the text referring to this analysis in the section “Identification of RNA modifiers that affect p53 response through reporter-based CRISPR screening”.

3. The data presented in Fig. 2 were not convincing. siRNA targeting *YTHDC1* only showed a modest reduction in *YTHDC1* protein level. Similarly, Fig. 2E showed that siRNA targeting *YTHDC1* did not result in significant downregulation of *YTHDC1*

expression. The authors need to use YTHDC1 knockout cells and reconstituted cells to further validate their results.

Figure R2. Representative immunoblot and quantification of YTHDC1 and p53, with GAPDH as loading control. Cells were transfected with siRNA against *TP53*, *YTHDC1* or Scramble (SCR) as negative control. After silencing cells were treated to Nutlin-3a, DMSO was used for the untreated condition as negative control. The bar plot shows protein quantification relative to GAPDH level of $n = 5$ biologically independent experiments for p53 (**A**) and YTHDC1 (**B**).

Now we show a more representative WB picture in Figure R2A, which will replace the old Figure 2A and also, we have quantified the knockdown levels of YTHDC1 protein. It is reduced to 46.7% (Figure R2B, and new Figure EV2A). This is strong and significant downregulation of the protein, and results in a decrease of p53 expression to 64%, comparable to the downregulation of p53 obtained with p53 siRNA (56.8%) in Nutlin-3a-treated condition.

The mRNA knockdown levels for *YTHDC1*-siRNAs are shown in Figure EV1H, which are between 70% and 85% of knockdown efficacy depending on the use of siRNA-1 or siRNA-2, respectively.

However, we realised that there was a mistake in the labelling of Figure 2E, because the labels for "*TP53*" and "*YTHDC1*" were interchanged. We apologize for such mistake that we have corrected in the revised Figure 2E. The inhibition of *YTHDC1* observed in the RNA-seq experiment presented in this figure is highly significant (p adjusted value = $5.29E-59$), and results in downregulation of *TP53* mRNA levels ($\log_2FC = -0.704265974$, p adjusted value = $2.7E-26$). Most importantly, it is linked to a significant alteration of the p53 pathway (Fig 2F). The independent quantification of *YTHDC1* mRNA by qRT-PCR in these samples confirmed the knockdown efficiency (Figure R3).

Figure R3. RNA level quantification by RT-qPCR for *YTHDC1* in RNA-seq samples. Cells were transfected with two siRNA against *YTHDC1* (DC1 (1+2)), or Scramble (SCR) as negative control.

Altogether these data demonstrate that the downregulation of *YTHDC1* leads to a decrease of p53 levels with a general impact in the p53 pathway, in line with the expected based on the screening results.

Following the reviewer's suggestion, we spent several months trying to generate a *YTHDC1* KO cell line. After many rounds of selection, and after screening hundreds of candidate clones, our success rate was way below what we typically expect for this kind of experiment. We have extensive expertise in generating CRISPR-KO stable cell lines, for both long non-coding and protein coding genes we typically are able to obtain between 5 and 15% of screening clones harbouring an homozygous deletion of the targeted gene. Only recently, during our last attempt we screened 80 potential clones and we were finally able to find a single positive clone.

Sanger sequencing confirmed that this clone presents a frameshifting deletion of 1 nucleotide in both *YTHDC1* alleles (Figure R4A), resulting in a complete depletion of *YTHDC1* protein (Figure R4B-C).

Figure R4. (A) Comprehensive profile of insertions and deletions (indels) in YTHDC1-KO A549 clone compared to a control A549. **(B)** Representative picture of a Western blot of normal A549 cell line and the YTHDC1-KO A549 clone. **(C)** Western blot quantification of YTHDC1. **(D)** RNA level quantification of mature mRNA and pre-mRNA by RT-qPCR for *TP53* in A549 cells (grey) and YTHDC1-KO (dark blue). **(E)** Intron retention measurement for *ATR*, *BIRC6* and *SETX* by qRT-PCR, normalising signal from an upstream exon junction.

Although it could be misleading to generalise our results based on a single CRISPR-KO clone, the phenotype we observed in this cell line recapitulates our previous findings obtained by using siRNA silencing. This clone showed decrease in both mature and pre-mRNA of *TP53*, confirming that the lack of YTHDC1 results in defects on RNA polymerase II proper function on this specific gene (Figure R4D). Moreover, we were able to validate the increased intron retention of *ATR*, *BIRC6* and *SETX* (Figure R4E). Thus, we concluded that YTHDC1 knock-out results in the same phenotype as observed in siRNA YTHDC1 knockdown A549 cells.

Figures R4A-D has been included in the revised manuscript as Figure EV2C-E, and Figure R4E has been included as Figure EV4F. And the text has been updated accordingly.

Because it took us very long time to generate an homozygous YTHDC1 knock-out clone, we were not able to consider this clone for the most complex experiments. Even so, we included it in the manuscript as supporting evidence. Nevertheless, to address the reviewers' request, while working on KO generation, we used alternative approaches to provide independent lines of evidence of the role of YTHDC1. On the one hand, we have performed experiments with the METTL3 inhibitor (STM2457), and attempted to rescue the phenotypes by overexpressing the wild type or tryptophan-mutant YTHDC1 in a background where the endogenous protein had been depleted (please see response 3

to Referee #2). Altogether our results confirm the role of YTHDC1 in *TP53* transcription and intron retention in m6A-independent and dependent manner, respectively.

4. The effect on RNAPII pausing shown in Fig. 3 was modest and not specific for p53 gene.

We agree with the reviewer that the effect observed in RNAPII pausing may seem modest. However, due to the nature of the analysis to calculate RNAPII pausing, which considers differences between signal of RNAPII in TSS versus gene body, those are expected values.

However, to benchmark this result, we took public data of RNAPII ChIP-seq from a previous study where the phenotype of the knock-out of a factor centrally involved in RNAPII elongation was studied (Wu et al., 2022). To be able to compare the data, we performed the same analysis as for YTHDC1 knock-down cells by calculating the RNAPII pausing index of genes in wild type cells compared to knock-out cells for the negative regulator of elongation NELF (Figure R5).

Figure R5. (A) Violin plots of the distribution of global RNAPII Pausing Index ratio in NELF-KO (green) and YTHDC1-KD (blue) **(B)** Violin plots of the distribution of expression corrected Pausing Index ratio calculated on the respective ChIP-seq experiment for NELF (green) and YTHDC1 (blue). All the actively transcribed genes were divided into four different groups based on the presence or absence of NELF peaks in the TSS (green and light green, respectively) and based on the presence or absence of YTHDC1 peaks in the TSS (cyan and blue, respectively). Pausing index was calculated as \log_2 RNAPII promoter density/RNAPII gene body density for both cells transfected with scramble (SCR) siRNA and YTHDC1 siRNA. Data are presented as distribution of Pausing Index ratio for SCR versus YTHDC1 knockdown for each category. Data are presented as mean \pm s.d. ns, not significant ($P \geq 0.01$), * ($P \leq 0.01$), ** ($P \leq 0.001$), *** ($P \leq 0.0001$), **** ($P \leq 0.00001$), paired two-tailed student's t-test was performed.

In Figure R5A, we show the general effect in RNAPII pausing index in NELF WT vs KO cells and YTHDC1 KD vs SCR cells. Both resulted in seemingly modest changes of RNAPII pausing index values, however highly significant, even slightly higher in YTHDC1 knock-down condition. In Figure R5B, we show the measurement of the pausing index in genes with and without ChIP signal for NELF or YTHDC1 in their respective datasets. The differences between NELF-KO are higher than YTHDC1-KD, but it is something expected. We have to take into account that NELF was genetically knocked out (not knocked down) which should have more dramatic effect. Taking together these results, we conclude that YTHDC1 depletion causes a highly significant pausing of RNAPII. We thank the reviewer for this comment, which made us notice the significance of YTHDC1 function compared to other factors.

Considering these results, we strongly believe that the effect on RNAPII pausing is sufficient to result in reduced levels of p53 and affect the p53 pathway (Figure 2F).

Regarding the second point, the referee is correct that YTHDC1 not only affects p53 transcription, which we believe is an interesting observation, but this affection is enough to explain the significant reduction that we found of p53 at protein and RNA level upon YTHDC1 depletion (Figure R2A and Figure 2C). Moreover, since genes affected by RNAPII pausing upon YTHDC1 depletion are enriched in DNA damage/DNA repair genes (Figure EV3B), our results indicate that YTHDC1 acts as a master regulator of the DNA damage response, which has strong biological implications.

To show these new data, we show the comparison between YTHDC1 and NELF (Figure R5B) as revised Figure 3B and Figure EV3A, respectively, and edited the text accordingly

Referee #2 (Report for Author)

This is a very interesting manuscript where the authors have identified YTHDC1 as a critical mediator of p53 and its transcriptional response. A major driver for publication is the identification of a completely new regulator of p53, which is very surprising since this is such a highly studied protein. The other major driver is that this study links a very important m6A pathway protein, YTHDC1 with the p53 pathway.

This was a very interesting paper to read. The authors use a very good and high quality CRISPR screening approach. The connection between YTHDC1 and p53 is compelling.

We thank the referee for appreciating the impact of this work and the special interest of the CRISPR screening results.

However, there are problems with the mechanisms that need to be addressed. I'm not fully convinced of the mechanistic connections due to some technical problems and other issues with some of the experiments.

We hope that the multiple new analyses and revisions satisfy the referee's concerns.

1. First, I think most readers will be surprised that YTHDC1 has not come up in prior CRISPR or other screens for regulators of p53 function. Why is that? Perhaps YTHDC1 did come up, but it was never studied or addressed. The author should, in the supplement, list all of the screens that have been performed for p53 regulators, and indicate where YTHDC1 was ranked amongst the regulators. Perhaps, some of these earlier screens did not include YTHDC1. Alternatively, if there are screens that are very similar to the ones performed here, and YTHDC1 did not appear as a among the topics, the net is somewhat concerning. It should at least be mentioned.

The reviewer correctly pointed out that both YTHDC1, one of the main regulators of m6A biology, and p53 pathway were extensively characterised during the last decades. However, the interplay between RNA metabolism and transcriptional response, is a relatively recent field often overlooked by the scientific community. Indeed, our work was motivated by the lack in literature of CRISPR screenings specifically designed to unbiasedly detect p53 non-canonical regulators, which we believe is the main novelty of our study. Most of the publicly available screenings relied on proliferation or survival as readout in basal or drug treatment conditions (Bock et al., 2022; Bowden et al., 2020; Stolte et al., 2018; Zhang et al., 2023). Due to the very high throughput nature of this approach, it is challenging to validate all the hits that show positive or negative correlation with the phenotype under exam. The standard procedure in a CRISPR screening set up is to validate less than 1% of the putative hits, limiting further investigations typically to the top 10 or 20 candidates. It is likely that a very high number of candidate regulators were excluded from the analysis because they were not part of the top 1% ranked genes.

With that in mind, we extensively reviewed all the CRISPR Screenings performed in human cell lines deposited in the BioGrid ORCS repository (<https://orcs.thebiogrid.org/Gene/91746>). We found that sgRNAs targeting YTHDC1 have been included in a total of 1276 screenings, and it is shown as a significant hit in 485. 96.9% studied cell proliferation/viability phenotype in multiple contexts (Table R1, Figure R6). Among these 485 screening, the perturbation of YTHDC1 was significantly associated with defects in cell proliferation or survival 457 times, highlighting the fundamental role of this gene in maintaining cellular homeostasis, which could be in line with the here described role as a master regulator for DNA damage response.

Interestingly, we found that five CRISPR-KO screenings have been performed to study synthetic lethality in TP53-KO background. In two of them YTHDC1 was considered significant, but both studies did not consider YTHDC1 for further studies, probably because it did not rank among top candidates. YTHDC1 KO resulted in positive selection in hTERT-RPE1 cells (Drainas et al., 2020), where YTHDC1 ranked as 66 hit out of 20890 candidates by increasing proliferation in these cells. In addition, a negative selection was found in HEK293-A cell lines (Feng et al., 2022), where YTHDC1 ranked as 788 out of 18053, by reducing cell proliferation.

Figure R6. Representative proportion of CRISPR Screenings where YTHDC1 was considered as hit (blue), highlighting the kind of selection that has been performed (A) and the kind of phenotype that has been studied (B).

The Table R1 has been included in the Table S1 of the revised manuscript. Moreover, we discuss this point in the Discussion:

2. The authors bring up the very interesting finding that the DepMap shows a strong codependency between p53 and YTHDC1. However, there are some pretty serious problems with this line of reasoning. The authors claim that the mechanism of YTHDC1 is m6A independent. However, the authors may not realize that METTL3 shows very strong codependency with p53 as well. Other components of the m6A writer complex also show high codependency. This seems to argue against the core idea that the authors present which is that the actions of YTHDC1, at least on p53, are independent of m6A.

As suggested by the reviewer, we looked at other m6A regulators, such as *FTO* and *ALKBH5* m6A-erasers, and both of them show a positive correlation with *TP53* expression, although one would expect the opposite based on the correlation with *METTL3*. If *TP53* expression was simply m6A dependent, writers and erasers should show neutral or opposite trends, which is not the case.

Although the DepMap is in agreement with a relationship between *YTHDC1* and *TP53*, it is difficult to draw mechanistic conclusions from this type of correlative analysis, based on the correlations that we saw in *METTL3*, *FTO* and *ALKBH5* analysis.

However, all the experimental analyses that we have performed in this study go in support of the positive correlation between *YTHDC1* and *TP53* expression in A549 cell line. Which is not the case for the relationship between *METTL3* and *TP53* as we have shown for the *METTL3* knockdown and inhibition assays (please see next response and Figure 5E, 5G).

3. A major method that the authors used to demonstrate that the m6A deposition is not needed for the actions of YTHDC1 is that they use siRNA to knock down METTL3, and failed to see p53 regulation similar to what they see with YTHDC1 knockdown. The authors may not know this, but METTL3 knocked down often does not deplete m6A levels. This is because METTL3 is generally knocked off the rate limiting for m6A production. Schwartz was the first to show that knockdown of METTL3 by 90% causes almost no change in m6A levels. Indeed, the authors did not measure m6A levels after their METTL3 knocked down to the best of my understanding. Typically one would expect that the authors would knock out YTHDC1 and replace it with a tryptophan mutant, or similar, that cannot bind m6A.

The reviewer is right, we can't assume that METTL3 depletion by siRNA will result in a significant downregulation of m6A level on total mature RNA. Although, our work is framed around the observation that newly synthesised RNA could be affected by YTHDC1 depletion at multiple levels, such as transcription and splicing. While the mature RNA already produced and marked with m6A seems to be very stable, we speculate that in absence of METTL3, the newly transcribed molecules cannot be processed by the methylation machinery, therefore we believe we are severely underestimating the effect of METTL3 depletion on nascent RNA due to technical limitation.

In light of the reviewer's important comment, to further strengthen our hypothesis that YTHDC1 possesses both m6A dependent and non-dependent functions, we have used a parallel approach to modulate cellular m6A global status. Yankova and collaborators have designed a highly specific inhibitor (STM2457) of the catalytic activity of METTL3-METTL14, the main writer complex of m6A-RNA modification (Yankova et al., 2021). Xiao and colleagues (Xiao et al., 2023) have previously tested STM2457 activity in A549 cells achieving strong reduction of m6A level in a shorter period of time compared to METTL3 RNA depletion. We were able to replicate these results achieving close to 90% reduction of m6A with 2.5 μ M of STM2457 (Figure R7).

Figure R7. TLC quantification of the ratio between total amount of Adenosine nucleotides (A) and N6-methyladenosine nucleotides (m6A).

Therefore, we analysed *TP53* mRNA and pre-mRNA levels, as well as intron retention for *ATR*, *BIRC6* and *SETX*, under effective m6A depletion conditions to elucidate dependency or independency for m6A modification of the different mechanisms.

With this tool, we confirmed that *TP53* pre-mRNA remained unaffected by the increasing concentration of the METTL3 inhibitor (Figure R8A), unlike *YTHDC1* depletion, and validating the hypothesis that m6A does not play a key role in controlling RNA polymerase II stalling/elongation properties, at least for this gene. On the other hand, we observed a significant positive correlation between the intronic signal and m6A levels (Figure R8B). Thus, we concluded that the depletion of m6A affects splicing dynamics in the same manner previously observed by knocking down METTL3 mRNA (Figure 5C-D), further proving that the presence of m6A seems to be essential for the RNA maturation process of these genes.

Figure R8. (A) RNA level quantification of mature mRNA (orange) and pre-mRNA (light orange) by RT-qPCR for *TP53* in A549 cells treated with increasing concentrations of STM2457. **(B)** Intron retention measurement for *ATR*, *BIRC6* and *SETX* by qRT-PCR, normalising signal from their respective upstream exon junction in A549 cells treated with increasing concentrations of STM2457.

In addition, following the reviewer's suggestion, we performed a rescue experiment overexpressing the wild-type and mutant form of *YTHDC1*, which is not able to bind m6A, in cells depleted for the endogenous form of this protein.

We generated A549-stable cell lines harbouring the *YTHDC1-wt* version or *Trp-mutant* (W377A / W428A) under the control of a doxycycline inducible promoter. Therefore, upon *YTHDC1* knockdown with a siRNA targeting 3'-UTR of the endogenous *YTHDC1* (Figure R9B), recombinant *YTHDC1* expression was induced with different doxycycline doses.

First, we titrated doxycycline concentration in order to achieve equivalent expression levels of the *wild-type* and *mutant* variants, which were 20 and 200 ng/mL doxycycline, respectively (Figure R9A). To consider that observed effects are caused by *YTHDC1* overexpression and not doxycycline treatment, parallelly we treated normal A549 cells with concentrations previously mentioned. This confirmed that doxycycline alone does not affect either *TP53* transcription (Figure R9C) or intron retention events (Figure R9D).

Then, we measured *TP53* mRNA and pre-mRNA levels to test whether the re-introduction of *YTHDC1* is able to recover the decreased expression of *TP53* caused by *YTHDC1* KD. Surprisingly, neither the re-expression of WT *YTHDC1* nor MUT *YTHDC1* was able to rescue *TP53* transcription levels (Figure R9E). We speculate that a specific feature of the exogenously expressed protein could interfere specifically with the phenotype observed on the transcription elongation of *TP53* gene. Of note, both

exogenously expressed proteins were tagged at the N-terminal domain of the protein to facilitate the discrimination between endogenous and exogenous protein with biochemical assays. The addition of this small peptide could mask specific protein-protein interactions that are essential for YTHDC1 to exploit its function in regulating RNA elongation. Alternatively, it is possible that the expression of YTHDC1 alone is not sufficient to restore the correct chromatin environment needed for normal TP53 expression.

On the other hand, the overexpression of the *wild-type* YTHDC1 but not *mutant* was able to decrease the intron retention in *ATR*, *BIRC6* and *SETX* observed upon depletion of endogenous YTHDC1, confirming the hypothesis that splicing dynamics are regulated by YTHDC1 in m6A-dependent manner (Figure R9F).

In conclusion, the results of the experiments performed with the *wild-type* and *mutant* forms of YTHDC1, together with the observed effect of the highly efficient METTL3 inhibitor, confirm the YTHDC1 and m6A dependency of the *ATR*, *BIRC6* and *SETX* splicing events under study. While the expression of TP53 couldn't be rescued by the re-introduction of YTHDC1, our experiments performed in m6A depletion clearly disentangle the observed phenotype from the presence of m6A.

Figure R9. (A) Representative western blot of HA-YTHDC1 *wt* and *mutant* versions upon doxycycline treatment with 20 and 200 ng/mL, respectively. **(B)** RNA level quantification of endogenous YTHDC1 upon 3'-UTR designed siRNA transfection. **(C-D)** Parental A549 cells treated with concentrations of Doxycycline applied for YTHDC1 inducible system. **(E)** RNA level quantification of TP53 mRNA and pre-mRNA levels in A549 cells that has been transfected with siRNA for endogenous YTHDC1 and induced expression of *wt* and *mutant*. **(F)** RNA level quantification of intron retention for *ATR*, *BIRC6* and *SETX* genes in A549 cells that has been transfected with siRNA for endogenous YTHDC1 and induced expression of *wt* and *mutant* (E-F).

We want to thank the reviewer for suggesting this set of experiments, following her/his guidance, we were able to further strengthen our initial hypothesis, providing several lines of evidence supporting the presence of both m6A-dependent and m6A-independent functions of YTHDC1.

Due to the importance of the new found features related to the m6A dependence for both processes, we decided to generate a new section in the revised manuscript titled "m6A dependency on YTHDC1 regulation", where we integrated the data from this part together with the response to Point 3 of Referee#3. We included a new Figure 5 and Figure EV5 in the revised manuscript, where we present all these results.

4. Note that some studies show that YTHDC1 is a major component of speckles :<https://doi.org/10.1016/j.devcel.2021.01.015> - this activity involves its ability to bind m6A and the MALAT1 noncoding RNA, target transcripts do not necessarily need to contain m6A, similar to what the authors find here. I would suggest that they reconsider their proposed mechanism, at least in the context of m6A independent functions on p53.

Although in our paper we did not propose a specific mechanism, this is a very interesting observation. We thank the reviewer for his/her suggestion that YTHDC1 could be acting through or by the recruitment of *MALAT1*.

To address this point, we performed the single or the simultaneous knockdown of *YTHDC1* and *MALAT1* to test whether they have synergistic or antagonistic effect in the described phenotype for *TP53* expression and splicing deficiencies. We expect that if YTHDC1 is acting on these processes via *MALAT1*, the knockdown of this long non coding RNA should phenocopy the effects we previously observed by knocking down YTHDC1, and the double knock down should affect transcriptional behaviours with even stronger magnitude.

First, we confirmed the knockdown efficacy for LNA of *MALAT1* and siRNAs of *YTHDC1* (Figure R10A), achieving reduction in expression of 95% and 85%, respectively.

Focusing on *TP53* expression (Figure R10A), we confirmed that *YTHDC1* knockdown alone resulted in the reduction in both mature and pre-mRNA of *TP53*. However, we did not observe significant effects by altering *MALAT1* levels, suggesting that *MALAT1* is not involved in *TP53* transcriptional regulation by YTHDC1. By knocking down simultaneously both RNAs, we obtained the same level of reduction of mature or pre-*p53* mRNA observed with the single YTHDC1 perturbation, confirming that the phenotype observed is exclusively ascribable to this protein.

Having in mind that splicing is a process that happens co-transcriptionally, we thought that *MALAT1* could also facilitate YTHDC1 localization into splicing machinery. However, when analysing the splicing phenotype, we found again that *YTHDC1* knockdown alone resulted in unproductive splicing for the studied introns of *ATR*, *BIRC6* and *SETX* introns, but we did not find significant changes when we reduced *MALAT1* (Figure R10B), meaning that this process is also controlled for YTHDC1 protein alone.

Taken together, these results show that *MALAT1* does not seem to play a role in the YTHDC1 regulations under study.

Figure R10. (A) RNA level quantification of *MALAT1*, *YTHDC1*, *ATR* and *TP53* mRNA and pre-mRNA levels in A549 cells that has been transfected with siRNA for *YTHDC1* or LNA for *MALAT1* or both. **(B)** Exonic and intronic RNA quantification for *ATR*, *BIRC6* and *SETX* in A549 cells that has been transfected with siRNA for *YTHDC1* or LNA for *MALAT1* or both.

5. I'm somewhat suspicious of the proposed mechanism of *YTHDC1* regulating splicing. The authors should look carefully at the work of Brockdorff (<https://www.ncbi.nlm.nih.gov/pmc/articles/PMC8327914/>) these authors did a very careful analysis of m6A and its role in splicing using acute depletion. This is important because m6A knock down and other m6A inhibition mechanisms can result in secondary and tertiary effects which ultimately affect splicing. However, this work using acute m6A depletion, I believe, did not show any overlap with the proposed alternative splicing mechanisms shown here. I think that the mechanism that they've identified is an indirect effect - m6A probably affects a certain regulator, which ultimately affects other pathways that lead to splicing defects that are ascribed to *YTHDC1* in this study. More documentation of the direct effect of *YTHDC1*, in light of Brockdorff needs to be described, or the authors need to change their story so that they encompass secondary effects of m6A/*YTHDC1* depletion.

We thank the reviewer for bringing to our attention this important piece of literature. The authors of this work claim that the vast majority of splicing aberrations observed upon long-term depletion of *METTL3* are due to secondary effects, such as deregulation of

factors involved in this process. To disentangle direct and indirect effects, they establish a protocol where METTL3 is rapidly depleted through a degron system, allowing them to quickly abolish m6A deposition in hours instead of days, in order to easily isolate only the direct effects resulting from the lack of this epitranscriptomic mark. After carefully analysing their data, the authors conclude that, even if long-term phenotype is often confounded by secondary processes, all the splicing changes observed are traceable to the presence of m6A, which is in line with the observations we made in our work. The author specifically says:

“Our analysis extends previous models that either only covered exon skipping (Xiao et al. 2016) or focused on intron retention (Fish et al. 2019). Although we observed some overlap with splicing changes seen in prior studies our analysis detected many more instances of m6A-mediated differential splicing. The fact that our observations followed acute depletion of METTL3 suggests that the changes are directly linked to METTL3 function rather than secondary long-term effects from perturbing the m6A system.”

Moreover, to provide mechanistic insight of their observations, they explore the role of the RNA binding protein RBM15 in this process by intersecting chromatin meRIP data with irCLIP-seq data of RBM15. Despite they found a correlation between RBM15 and m6A, the presence of this RNA binding protein was reported only in 50% of the m6A sites in nascent RNA, leaving open the possibility that alternative mechanisms could co-exist to regulate this nascent RNA through m6A reading.

Interestingly, they also explore the possibility that YTHDC1 could have a direct role in regulating splicing dynamics, concluding that:

“the splicing changes of these m6A-bearing alternative 5'-splicing events observed in dTAG METTL3 phenocopy those seen in Ythdc1 conditional knockout”.

Here we report supplementary figure 10 of Brockdorff's paper, where they clearly see almost perfect overlap between m6A depletion direct effects and knock-out of YTHDC1.

This finding recapitulates perfectly what was observed by us in the present study. Although in different systems, we are confident that Brockdorff and our work reveal the presence of m6A/YTHDC1 axis in regulating splicing dynamics.

Brockdorff demonstrates a crosstalk between the m6A machinery and the regulation of RNA splicing, which is generally in agreement with our observations, although, Brockdorff's study does not directly investigate how YTHDC1 may be involved in these regulations, where other m6A readers, such as RBM15 are involved. Although we don't fully understand the mechanism behind YTHDC1 splicing regulation, our data strongly points to the direct location of YTHDC1 on the genes under study, where we focus on specific examples of intron retention where both m6A and YTHDC1 are involved. We therefore believe that our conclusions are compatible with the work of Brockdorff and colleagues, although we lack the high resolution achieved with an acute depletion of the protein.

We also considered the possibility that intron retention events could be indirect of *YTHDC1* knockdown. However, *YTHDC1* is generally bound to the introns affected (Figure EV4D), and in particular to the 3 DDR factors that we focused (Figure EV5B), as

indicated by YTHDC1-CLIP-seq data (see response 1 to Referee #3). In addition, wt YTHDC1 but not the mutant protein is able to rescue the splicing defects, further supporting the direct role of the protein. Our data also supports that this regulation is dependent on m6A, as shown by ChrMeRIP transcriptome-wide (Figure 5B) and specifically in *ATR*, *BIRC6* and *SETX* (Figure EV5B). Moreover, as discussed in response 3, we took the reviewer's suggestion and we went through using METTL3 inhibitor approach, which confirmed the m6A-dependency of the intron retention events observed upon YTHDC1 depletion.

6. Attardi proposed a mechanism of p53 stabilization via Mettl3 which seems to provide an alternative model for m53 regulation (<https://doi.org/10.1016/j.molcel.2022.04.010>). I cannot reproduce this study but nevertheless, the authors should address whether this can account for p53 expression in their hands or rebut this study, especially if they find Mettl3 dependence for their pathway

We thank the reviewer for this comment. We were indeed aware of this study. However, as pointed out, Attardi's work was focused on p53 regulation at the protein level, different to what we are studying. Although both mechanisms may be superimposed in the cells, the effects that we report in our manuscript are due to changes in *TP53* mRNA transcription.

As above described, we extensively proved that p53 RNA level is not affected by METTL3 (Figure 5C and Figure R8A). Silencing the enzyme through siRNA, or abolishing its function with specific small molecules do not alter the overall level of p53 transcript.

Moreover, METTL3 was one of the targets of our CRISPR screening. Therefore, we decided to go through our screening results looking for evidence in line with Attardi's previous observation. We realised that METTL3-KO leads to a slight increase of p21 reporter signal (Figure R11), opposite to the expected based on Attardi's work. METTL3 ranked as 248 and 376 candidate in p53-enhanced- and p53-attenuated-response populations, respectively. In conclusion, we can assume that, at least in our system, we were not able to validate the existence of a direct correlation between the activity of METTL3 and overall level of p53 protein.

Figure R11. Scatter plot of Log₂ Fold Change for sgRNA enrichment in p53-enhanced-response (Y-axis) and p53-attenuated-response (X-axis) populations. METTL3 and YTHDC1 has been marked.

Referee #3 (Report for Author)

Through a targeted CRISPR-knockout screen, these authors identified a number of genes involved in p53 regulation. In this list a nuclear m6A-binding protein YTHDC1 appears to be a master regulator of p53 expression. The authors showed two pathways underlying this regulation. YTHDC1 can bind to the transcription start sites of TP53 and other genes involved in DNA damage response, promoting their transcriptional elongation. This protein also recognizes m6A and mediates correct splicing. Depletion of YTHDC1 leads to intron retention and reduced TP53 expression. These results are very interesting, particularly the splicing regulation part, which to my knowledge is novel and a rare example showing connection of intron methylation with splicing and functional outcome. I do have several questions that would need to be addressed.

We thank the referee for appreciating the interest of our study.

Main:

1. The authors uncovered two effects of YTHDC1, transcription regulation and splicing regulation. The authors will really need to separate these two effects on selected genes. They showed that YTHDC1 occupies promoter site through ChIP. Can the authors perform CLIP-seq and analyze proportions of YTHDC1 on promoter RNA or nascent RNA versus across splicing junctions. This is critical.

To address this point, we have performed YTHDC1 CLIP-seq experiments on chromatin extracts, since it is highly enriched in RNA nascent. To that end, we have applied formaldehyde crosslinking and a chromatin fractionation prior immunoprecipitation with the specific anti-YTHDC1 antibody (Abcam #ab264375). The immunoprecipitated RNA was recovered and used for library preparations using a protocol for total ribo-depleted RNA and sequenced.

As the reviewer proposed, we have analysed peak distribution proportions for YTHDC1 CLIP-seq and we compared with peak distribution for YTHDC1 ChIP-seq. We divided localization in 6 different categories: Upstream TSS (considered as peaks located 500 bp upstream from TSS), TSS (considered as peaks located 500 bp downstream from TSS), Splicing sites (taking a window of ± 500 bp for 5'- or 3'-splice sites), exon, intron and not annotated regions (Figure R12A). As expected, there is a clear enrichment for ChIP peaks around TSS, suggesting that YTHDC1 chromatin localization at TSS is critical for transcriptional regulation. Regarding YTHDC1-CLIP distribution, we see that there is enrichment around splice sites and exons. This differential distribution between YTHDC1 ChIP and CLIP peaks highlights that both processes, transcriptional and splicing regulation, are two separate functions for YTHDC1 that might involve different cofactors.

Moreover, when we analysed genes that suffered differential pausing index and genes with increased intron retention (please see Response 5), we did not observe significant overlap between genes with each phenotype, supporting additional evidence of the distinct nature of both processes.

Figure R12. (A) Distribution for YTHDC1 ChIP and CLIP peaks defined as upstream TSS (-500 bp to TSS), TSS (TSS to +500bp), Splicing sites (± 50 bp from 5'- and 3'-splice site), exons, introns and non-annotated (non-genic) regions.

We added this analysis in Figure EV4E, representing the relative YTHDC1 localizaiton at TSS and splicing sites, and modified the text accordingly.

2. A previous work from Yang Shi and coworkers has shown a mechanism of m⁶A to promote nascent RNA synthesis ([https://www.cell.com/molecular-cell/pdfExtended/S1097-2765\(22\)00111-3](https://www.cell.com/molecular-cell/pdfExtended/S1097-2765(22)00111-3)). It looks to me this pathway might be the main pathway functioning here instead of splicing regulation. Again, the authors need to carefully analyze both pathways.

We thank the reviewer for this comment. We were also aware of this work, so we also considered the possibility that both pathways could be related. Based on Yang Shi's work, YTHDC1 protects m⁶A nascent RNA from degradation by Integrator complex. If this pathway was controlling the splicing effects, we should also detect changes of the transcript levels of the genes suffering splicing changes. However, RNA-seq shows no changes in RNA levels of the genes affected by intron retention. The mRNA levels for *ATR*, *BIRC6* and *SETX* are not affected by YTHDC1 depletion (Figure EV4A). Only *BIRC6* presented a sharp decrease of RNA-seq reads downstream of its retained intron upon YTHDC1 knockdown. We hypothesised that this sharp reduction is caused by the emergence of a cryptic polyadenylation signal (Figure 4C, middle panel), but does not seem consistent with general RNA control by the exosome coupled to the integrator complex.

Another point to take into consideration, and in line with comment 3 by Referee #2, is that in this study by Yang Shi, the authors assume that METTL3 knockdown leads to m⁶A downregulation, however they don't check m⁶A levels. To be precise, the effects reported in that paper are METTL3-dependent, but it is not clear whether they are m⁶A-dependent.

3. “to investigate whether m⁶A potentially mediates the differences in RNAPII pausing that we observed in YTHDC1-knockdown cells, we divided the transcriptome into m⁶A genes and non-m⁶A genes,... suggesting that it is the presence of YTHDC1, rather than the presence of m⁶A, is the most critical for the control of elongation dynamics in our system” makes no sense. The authors are studying nascent RNA and introns, yet they were trying to correlate m⁶A on mature mRNA with transcription and splicing changes. They really need to map m⁶A (MeRIP) on nascent RNA or chromatin RNA. This is key to the current work.

We thank the reviewer for this important comment. To address this point, we analysed publicly available MeRIP data performed on chromatin-associated RNA (ChrMeRIP) (Xu et al., 2022) (Figure R13).

We also used the newly generated YTHDC1-CLIP on chromatin RNA to classify genes in two groups: bound or not by YTHDC1 at the RNA level (Figure R13, blue bondi violin plots). The increasing RNAPII pausing index ratio of transcripts without YTHDC1-CLIP binding is even higher than those where YTHDC1 bound. This indicates that chromatin localization of YTHDC1 at the TSS of a gene is more determinant for the RNAPII pausing caused by YTHDC1 depletion. Moreover, the classification of genes in m⁶A and non-m⁶A, based on ChrMeRIP-seq signal, showed no significant differences in RNAPII pausing between both groups, supporting the m⁶A-independency of this YTHDC1-dependent phenomenon.

Thus, we concluded that YTHDC1-dependent RNAPII pausing is an m⁶A-independent mechanism that occurs in correlation with the chromatin localization of YTHDC1 at their TSSs. This conclusion is also supported by experiments where m⁶A levels are dramatically reduced by METTL3-inhibitor treatment (see Response 3, Referee #2), where *TP53* pre-mRNA levels are not affected. Taking together these results, we conclude that YTHDC1 regulates *TP53* transcription in a m⁶A-independent manner.

Interestingly, there is a ChrMeRIP m⁶A peak in the intron 1 of *TP53* transcript (Figure R14A), overlapping with the intronic lncRNA ENST00000571370.1. This lncRNA is a transcript of ~1700 nucleotides previously described but that has not been related with *TP53* expression (Pang et al., 2019; Zhang et al., 2022).

Regarding the intron retention events of *ATR*, *BIRC6* and *SETX*, there are significant m⁶A peaks in the introns and flanking exons of the introns that are retained upon YTHDC1 depletion (Figure R14B). Moreover, ChrMeRIP showed highly significant m⁶A peaks in the retained introns, emphasizing the m⁶A dependency of this mechanism (Figure R14C).

Figure R13. Violin plots of the distribution of expression corrected RNAPII Pausing Index ratio calculated on the respective YTHDC1-CLIP-seq (blue bondi), YTHDC1 (light blue) and ChrMerIP-seq (red) experiments. All the actively transcribed genes were divided into four different groups based on the presence or absence of YTHDC1-CLIP peaks (dark and light blue bondi, respectively), based on the presence or absence of YTHDC1-ChIP peaks in the TSS (cyan and blue, respectively) or defined as m⁶A or non-m⁶A genes by ChrMerIP-seq peak calling (dark and light red, respectively). Pausing index was calculated as \log_2 RNAPII promoter density/RNAPII gene body density for both cells transfected with YTHDC1 siRNA (DC1-KD) and scramble (SCR) siRNA. Data are presented as distribution of Pausing Index ratio for YTHDC1 knockdown versus SCR for each category. Data are presented as mean \pm s.d. ns, not significant ($P \geq 0.01$), * ($P \leq 0.01$), ** ($P \leq 0.001$), *** ($P \leq 0.0001$), **** ($P \leq 0.00001$), paired two-tailed student's t-test was performed.

Figure R14. Genome browser tracks for total RNA-seq of cell transfected with scramble siRNA (light grey) or siRNA against YTHDC1 (cyan), ChrMeRIP input (orange) and m6A specific peaks (dark red), and YTHDC1 CLIP (green) from public data of HEK293T and our data in A549 showing reads coverage over *TP53* (**A**) *BIRC6*, *SETX* and *ATR* retained introns (**B**). Sequencing data were normalized as Fragments Per Kilobase of transcript per Million mapped reads (FPKM). (**C**) Plot showing the number of differentially retained introns (dark bars) and 1000 random selections of GENCODE v19 introns (light bars), classified in 5 different groups depending on the proximity to a ChrMeRIP peak (group 1: peak inside the intron; group 2: peak in adjacent exons; group 3: peak in adjacent introns; group 4: peak anywhere inside the gene; group 5: no peak inside the gene). Empirical p-value was obtained by randomizing 100 times the selected introns to be compared.

In the revised manuscript we generated a new section titled “m6A dependency on YTHDC1 regulation”, where we summarize these results together with results obtained from Respond 3 to Referee #2.

4. The novelty of the work lies on splicing regulation. The authors can deliver dCas13-FTO or ALKBH5 to intron m6A sites or nascent RNA m6A sites and monitor productive mRNA synthesis.

We kindly disagree with the reviewer statement. We do not believe the novelty of our work exclusively lies on the observations we made on splicing regulation. Indeed, several other published works proved that m6A deposition and splicing are somehow related (Uzonyi et al., 2023; Xiao et al., 2016; Yang et al., 2022). We believe that the novelty here is broader. We are the first to prove that YTHDC1 has both m6A dependent and independent functions, which was previously proposed in the field, but not validated. Moreover, we are the first to link m6A-dependent aberrant splicing events to the DNA damage response in the context of lung cancer, providing evidence both in vitro and vivo.

Despite that, we took all the reviewers' comments seriously, and we thank the reviewer for suggesting this experiment.

By using the plasmids kindly donated by Mingyi Xie (Wang et al., 2023), we generated stable cell lines expressing dCasRX protein fused to both wild type and mutant version of ALKBH5 (Figure R15A). Since the success of the experiment heavily relies on the correct sgRNA design to direct dCasRX-ALKBH5 fusion protein in the close proximity to the m6A site under exam, we took advantage of published data from Samie R. Jaffrey's lab (Linder et al., 2015), where they mapped with two different techniques m6A sites at single-nucleotide-resolution.

By integrating these data with the experiments performed in this work we identified 2 m6A sites around the intron 3 and 4 of *ATR* (Figure R15B) as promising potential targets of YTHDC1 responsible for promoting the correct splicing.

Following Xie guidelines, to increase the chance of delivering ALKBH5 in the correct position to catalyse the removal of the methyl group, we designed and cloned 3 sgRNAs for each m6A site: at position -2, 0, and +2 relative to the m6A site. To further validate our design, we analysed each guide RNA sequence with cas13design software (<https://cas13design.nygenome.org/>).

Figure Rebuttal 15. (A) Representative western blot of HA-tag and GAPDH. Arrow highlights size of the dCasRX-ALKBH5 protein. **(B)** Genome browser tracks for total RNA-seq of cell transfected with scramble siRNA (light grey) or siRNA against YTHDC1 (cyan), YTHDC1 CLIP (green) from public data of HEK293T and our data in A549 ChrMeRIP input (orange) and m6A specific peaks for ChrMeRIP-seq, CIMS- and CITS m6A-seq (dark red) showing reads coverage over *ATR* retained intron. Sequencing data were normalized as Fragments Per Kilobase of transcript per Million mapped reads (FPKM). Right panel zooms in representative CIMS-m6A-seq peak of *ATR*.

Figure R16. Exonic and intronic signal quantification for *ATR*, in A549 cells that has been transduced with lentivirus carrying dCasRX-ALKBH5 or dCasRX-ALKBH5-H402A constructs. Later the cell lines were transfected with plasmids **(A)** or transduced with lentivirus **(B)** that carry CRISPR-sgRNAs to direct dCasRX protein to m6A sites localized at exon 3 or exon 4 of *ATR* transcript. For each m6A peak were designed 3 sgRNAs. Results are relative to effects on NT-sgRNA (non-target) as negative control.

dCasRX-ALKBH5 stable cell lines were transduced with the sgRNA and selected with

blasticidin for 3 days to ensure the homogenous expression of each guide RNA. 5 days after selection, cells were collected and analysed by real time quantitative PCR looking for differences in intron retention.

Unfortunately, after performing several replicates of the experiments, we were not able to draw any conclusion (Figure R16). All the conditions tested seem to indicate that the experiment is not working properly. Although very exciting, this is a very new technology, further validations of the feasibility of this approach need to be provided before it will become widely adopted by the m6A scientific community. We believe this is a very powerful tool, but each target requires laborious and extensive optimization of the technology that goes beyond the scope of our work. We hope we provided enough data (WT/Mutant-YTHDC1 overexpression together with response 3 to Referee #2) to convince the reviewer that the splicing phenotype observed in our work depends on the presence of m6A in the proximity of retained intron.

5. Again, effects of splicing changes versus transcription are critical. A few clear examples that splicing but not transcription affects outcome of gene expression and DNA damage response are required.

Yes, the reviewer is absolutely right, the effects observed in splicing versus RNAPII elongation are critical, and the main point of our paper. That is the reason why we have performed multiple approaches to evaluate them in a parallel manner. We provided evidence that genes, such as TP53, are affected at transcription level but not at splicing level. More importantly, we provided evidence that *ATR*, *BIRC6* and *SETX* are aberrantly spliced upon depletion of YTHDC1, but they do not show significant differences in mRNA level (Figure EV4A). We included in the discussion the following sentence to stress this point:

“Moreover, transcriptional elongation and splicing are functionally coupled (Caizzi et al., 2021), although there is increasing awareness of the existence of distinct splicing events that are linked to different subnuclear localizations, and different types of aberrant splicing (Tammer et al., 2022). Here we found differential distribution in the binding of YTHDC1 to the genome and chromatin-associated RNA (Fig EV4E), which could be a clue of a spatial separation of both processes, at least in some of the splicing events. The full elucidation of such interplay will shed light on the function of YTHDC1 in transcriptional control.”

Furthermore, we cross-compared data from genes with increasing RNAPII pausing index and genes with aberrant splicing events (Figure 4A) upon YTHDC1 knockdown, and found that 75% of genes with aberrant splicing events has no increasing RNAPII pausing index, and the other way, 99.8% genes with increasing RNAPII pausing index has no aberrant splicing events in their transcripts. Suggesting that these 2 layers of regulation are independent and mutually exclusive, at least in our system.

Minor:

Fig. 5F not clear.

We have obtained better pictures, so the quantified differences are more clear to the readers. We include this picture in the revised Figure 5F, thanks for the suggestion

“However, several YTH family proteins have been reported to control m6A-marked RNA, modifying their stability, resulting in dramatic changes to RNA half-life (Zaccara & Jaffrey, 2020; Z. Zhang et al., 2020)”

Wrong citation. First paper was published in 2013 by Wang et al. Important to cite original literature. Please go through the manuscript and cite original literature.

We apologize for this mistake, we modified the citation.

BIBLIOGRAPHY

Bock, C., Datlinger, P., Chardon, F., Coelho, M. A., Dong, M. B., Lawson, K. A., Lu, T., Maroc, L., Norman, T. M., Song, B., Stanley, G., Chen, S., Garnett, M., Li, W., Moffat, J., Qi, L. S., Shapiro, R. S., Shendure, J., Weissman, J. S., & Zhuang, X. (2022). High-content CRISPR screening. *Nature Reviews Methods Primers* 2:1, 2(1), 1–23. <https://doi.org/10.1038/s43586-021-00093-4>

Bowden, A. R., Morales-Juarez, D. A., Sczaniecka-Clift, M., Agudo, M. M., Lukashchuk, N., Thomas, J. C., & Jackson, S. P. (2020). Parallel crispr-cas9 screens clarify impacts of p53 on screen performance. *ELife*, 9, 1. <https://doi.org/10.7554/ELIFE.55325>

Drainas, A. P., Lambuta, R. A., Ivanova, I., Serçin, Ö., Sarropoulos, I., Smith, M. L., Efthymiopoulos, T., Raeder, B., Stütz, A. M., Waszak, S. M., Mardin, B. R., & Korbel, J. O. (2020). Genome-wide Screens Implicate Loss of Cullin Ring Ligase 3 in Persistent Proliferation and Genome Instability in TP53-Deficient Cells. *Cell Reports*, 31(1). <https://doi.org/10.1016/j.celrep.2020.03.029>

Feng, X., Tang, M., Dede, M., Su, D., Pei, G., Jiang, D., Wang, C., Chen, Z., Li, M., Nie, L., Xiong, Y., Li, S., Park, J. M., Zhang, H., Huang, M., Szymonowicz, K., Zhao, Z., Hart, T., & Chen, J. (2022). Genome-wide CRISPR screens using isogenic cells reveal vulnerabilities conferred by loss of tumor suppressors. *Science Advances*, 8(19), 6638. https://doi.org/10.1126/SCIADV.ABM6638/SUPPL_FILE/SCIADV.ABM6638_TABLES_S1_TO_S7.ZIP

Linder, B., Grozhik, A. V., Olarerin-George, A. O., Meydan, C., Mason, C. E., & Jaffrey, S. R. (2015). Single-nucleotide-resolution mapping of m6A and m6Am

throughout the transcriptome. *Nature Methods* 2015 12:8, 12(8), 767–772.

<https://doi.org/10.1038/nmeth.3453>

Pang, S., Lv, J., Wang, S., Yang, G., Ding, X., & Zhang, J. (2019). Differential expression of long non-coding RNA and mRNA in children with Henoch-Schönlein purpura nephritis. *Experimental and Therapeutic Medicine*, 17(1), 621.

<https://doi.org/10.3892/ETM.2018.7038>

Stolte, B., Iniguez, A. B., Dharia, N. V., Robichaud, A. L., Conway, A. S., Morgan, A. M., Alexe, G., Schauer, N. J., Liu, X., Bird, G. H., Tsherniak, A., Vazquez, F., Buhrlage, S. J., Walensky, L. D., & Stegmaier, K. (2018). Genome-scale CRISPR-Cas9 screen identifies druggable dependencies in TP53 wild-type Ewing sarcoma. *The Journal of Experimental Medicine*, 215(8), 2137. <https://doi.org/10.1084/JEM.20171066>

Tammer L, Hameiri O, Keydar I, Roy VR, Ashkenazy-Titelman A, Custódio N, Sason I, Shayevitch R, Rodríguez-Vaello V, Rino J, Lev Maor G, Leader Y, Khair D, Aiden EL, Elkon R, Irimia M, Sharan R, Shav-Tal Y, Carmo-Fonseca M, Ast G. Gene architecture directs splicing outcome in separate nuclear spatial regions. *Mol Cell*. 2022 Mar 3;82(5):1021-1034.e8. doi: 10.1016/j.molcel.2022.02.001. Epub 2022 Feb 18. PMID: 35182478.

Uzonyi, A., Dierks, D., Le Hir, H., Slobodin, B., Schwartz, S., Nir, R., Kwon, O. S., Toth, U., Barbosa, I., Burel, C., Brandis, A., & Rossmanith, W. (2023). Exclusion of m6A from splice-site proximal regions by the exon junction complex dictates m6A topologies and mRNA stability. *Molecular Cell*, 83, 237–251.

<https://doi.org/10.1016/j.molcel.2022.12.026>

Wang, Y., Traugot, C. M., Bubenik, J. L., Li, T., Sheng, P., Hiers, N. M., Fernandez, P., Li, L., Bian, J., Swanson, M. S., & Xie, M. (2023). N6-methyladenosine in 7SK small nuclear RNA underlies RNA polymerase II transcription regulation.

Molecular Cell, 83(21), 3818-3834.e7. <https://doi.org/10.1016/J.MOLCEL.2023.09.020>

Wu, B., Zhang, X., Chiang, H. C., Pan, H., Yuan, B., Mitra, P., Qi, L., Simonyan, H., Young, C. N., Yvon, E., Hu, Y., Zhang, N., & Li, R. (2022). RNA polymerase II pausing factor NELF in CD8+ T cells promotes antitumor immunity. *Nature Communications* 2022 13:1, 13(1), 1–14. <https://doi.org/10.1038/s41467-022-29869-2>

Xiao, W., Adhikari, S., Dahal, U., Chen, Y. S., Hao, Y. J., Sun, B. F., Sun, H. Y., Li, A., Ping, X. L., Lai, W. Y., Wang, X., Ma, H. L., Huang, C. M., Yang, Y., Huang, N., Jiang, G. Bin, Wang, H. L., Zhou, Q., Wang, X. J., ... Yang, Y. G. (2016). Nuclear m6A Reader YTHDC1 Regulates mRNA Splicing. *Molecular Cell*, 61(4), 507–519.

<https://doi.org/10.1016/J.MOLCEL.2016.01.012>

Xiao, H., Zhao, R., Meng, W., & Liao, Y. (2023). Effects and translational characteristics of a small-molecule inhibitor of METTL3 against non-small cell lung cancer. *Journal of Pharmaceutical Analysis*, 13(6), 625–639.

<https://doi.org/10.1016/J.JPHA.2023.04.009>

Xu, W., He, C., Kaye, E. G., Lan, F., Shi, Y., Shen, H., Li, J., Mu, M., Nelson, G. M., Dong, L., Wang, J., Wu, F., Shi, Y. G., & Adelman, K. (2022). Dynamic control of chromatin-associated m6A methylation regulates nascent RNA synthesis. *Molecular Cell*, 82, 1156-1168.e7. <https://doi.org/10.1016/j.molcel.2022.02.006>.

Yang, X., Triboulet, R., Liu, Q., Sendinc, E., & Gregory, R. I. (2022). Exon junction complex shapes the m6A epitranscriptome. *Nature Communications* 2022 13:1, 13(1), 1–12. <https://doi.org/10.1038/s41467-022-35643-1>

Yankova, E., Blackaby, W., Albertella, M., Rak, J., De Braekeleer, E., Tsagkogeorga, G., Pilka, E. S., Aspris, D., Leggate, D., Hendrick, A. G., Webster, N. A., Andrews, B., Fosbeary, R., Guest, P., Irigoyen, N., Eleftheriou, M., Gozdecka, M., Dias, J. M. L., Bannister, A. J., ... Kouzarides, T. (2021). Small-molecule inhibition of METTL3 as a strategy against myeloid leukaemia. *Nature* 2021 593:7860, 593(7860), 597–601. <https://doi.org/10.1038/s41586-021-03536-w>

Zhang, Y., Weh, K. M., Howard, C. L., Riethoven, J.-J., Clarke, J. L., Lagisetty, K. H., Lin, J., Reddy, R. M., Chang, A. C., Beer, D. G., & Kresty, L. A. (2022). Characterizing isoform switching events in esophageal adenocarcinoma. <https://doi.org/10.1016/j.omtn.2022.08.018>

Zhang, Z., Wang, H., Yan, Q., Cui, J., Chen, Y., Ruan, S., Yang, J., Wu, Z., Han, M., Huang, S., Zhou, Q., Zhang, C., & Hou, B. (2023). Genome-wide CRISPR/Cas9 screening for drug resistance in tumors. *Frontiers in Pharmacology*, 14, 1284610. <https://doi.org/10.3389/FPHAR.2023.1284610/BIBTEX>

Dear Dr Huarte,

Thank you for submitting a revised version of your manuscript. Your study has now been seen by the original referees #2 and #3, who find that their previous concerns have been addressed and now recommend publication of the manuscript. Referee #2 requested additional explanations/changes in the manuscript text which I think are reasonable and helpful (see below).

In addition, please find of a few mainly editorial points raised by our data editors that have to be addressed before I can extend formal acceptance of the manuscript:

- KW: missing
- REFERENCE FORMAT: ok, alphabetical, none et al., but DOI numbers are present
- COI: title needs renaming to "DISCLOSURE AND COMPETING INTERESTS STATEMENT"
- AC/CRedit: section needs to be removed
- The journal does not permit citation of "Data not shown". All data referred to in the paper should be displayed in the main or Expanded View figures: (on pages 17 and 21)
- Figures need to be uploaded as individual, high-res figure files
- DATASET EV LEGENDS: Supplemental Tables 1-4 should be renamed to Dataset EV1-EV4 with the corresponding callouts, and the correct label should be included in the legends (summary). Supplemental Table 4 is missing the legend.
- SOURCE DATA: Source data files need to be saved in a scheme one figure/folder and then uploaded as .zip files. E.g. all the Source data files for figure 1 need to be saved in a single folder and this needs to be zipped and then uploaded as "SD figure 1.zip" file.
- Papers published in The EMBO Journal are accompanied online by a 'Synopsis' to enhance discoverability of the manuscript. It consists of A) a short (1-2 sentences) summary of the findings and their significance, B) 3-4 bullet points highlighting key results and C) a synopsis image that is 550x300-600 pixels large (width x height, jpeg or png format). You can either show a model or key data in the synopsis image. Please note that the image size is rather small, and that text needs to be readable at the final size. Please send us this information together with the revised manuscript.
- The specific URL for GSE239333 dataset is not provided in the data availability statement.
- Figure Legends (main + EV): Please note that the legend for figure EV 4d is incorrectly labelled as EV 4f in the data information section for statistical test. This needs to be rectified.
- Figure legends should be reorganized to 2 sections: Figure legends (for main figures) and EV figure legends (for EV figures).
- Figure legends:
 1. Please indicate the statistical test used for data analysis in the legends of figures 2e-f; 4b; 5b; EV 3c; EV 4a.
 2. Please note that in figures 3b; 4g-i; 5a-f; EV 3a-b; EV 4d; EV 5c-d, g; there is a mismatch between the annotated p values in the figure legend and the annotated p values in the figure file that should be corrected."
 3. Please note that the box plots need to be defined in terms of minima, maxima, centre, bounds of box and whiskers, and percentile in the legends of figures 6b-c.
 4. Please note that information related to n is missing in the legends of figures 3d; EV 5f.
 5. Please note that the error bars are not defined in the legend of figure 5b."
 6. Please note that scale bar and its definition are missing for figure 6a.

Please let me know if you have any questions regarding any of these points. You can use the link below to upload the revised

files.

With best regards,

Cornelius Schneider

Cornelius Schneider, PhD
Editor | The EMBO Journal
c.schneider@embojournal.org

Referee #2:

The revision is highly improved.

I have two minor comments:

The authors say "Notably, proteins involved in m6A-mRNA metabolism emerged as key targets, with YTHDC1 as the top candidate. " People in the m6A field will want to know which proteins (the figure does not label any besides YTHDC1). So please rewrite this sentence as follows: "Notably, proteins involved in m6A-mRNA metabolism emerged as key targets, such as XX, YY, and ZZ, with YTHDC1 as the top candidate," and indicate other top m6A metabolism genes where it says XX, YY, and ZZ. Also please add/label them in the figure.

Related to the point above, the Discussion needs to explain how other -non-nuclear- m6A pathway proteins (if these are the m6A mRNA metabolism proteins mentioned above) would affect the p53 reporter since these proteins would not affect transcription. If these are proteins involved in m6A writing, then the question is why would they come up when they presumably do not interact with YTHDC1, which is proposed to regulate transcriptional pathways linked to p53. Some discussion is needed on this point.

Referee #3:

The authors have done extensive new experiments. The manuscript contains a large amount on information. The authors also proposed m6A-independent function of YTHDC1, which is indeed likely. I would suggest emphasize context dependent regulation. In different context p53 is regulated differently, primary cells versus cancer cells, different types of cancer cells with different genetic backgrounds.

The authors have addressed all minor editorial requests.

Dear Dr. Huarte,

I am pleased to inform you that your manuscript has been accepted for publication in the EMBO Journal.

Yours sincerely,

Cornelius Schneider, PhD
Editor
The EMBO Journal
c.schneider@embojournal.org
